# AQER: A Scalable and Efficient Data Loader for Digital Quantum Computers

**Kaining Zhang**[1], **Xinbiao Wang**[1], **Yuxuan Du**[1,2]*, **Min-Hsiu Hsieh**[3]*, **Dacheng Tao**[1]*

[1]Generative AI Lab, College of Computing and Data Science,
Nanyang Technological University, Singapore 639798, Singapore
[2]School of Physical and Mathematical Sciences,
Nanyang Technological University, Singapore 639798, Singapore
[3]Hon Hai (Foxconn) Research Institute, Taipei, Taiwan
`yuxuan.du@ntu.edu.sg,min-hsiu.hsieh@foxconn.com,dacheng.tao@ntu.edu.sg`

## Abstract

Digital quantum computing promises to offer computational capabilities beyond the reach of classical systems, yet its capabilities are often challenged by scarce quantum resources. A critical bottleneck in this context is how to load classical or quantum data into quantum circuits efficiently. Approximate quantum loaders (AQLs) provide a viable solution to this problem by balancing fidelity and circuit complexity. However, most existing AQL methods are either heuristic or provide guarantees only for specific input types, and a general theoretical framework is still lacking. To address this gap, here we reformulate most AQL methods into a unified framework and establish information-theoretic bounds on their approximation error. Our analysis reveals that the achievable infidelity between the prepared state and target state scales linearly with the total entanglement entropy across subsystems when the loading circuit is applied to the target state. In light of this, we develop AQER, a scalable AQL method that constructs the loading circuit by systematically reducing entanglement in target states. We conduct systematic experiments to evaluate the effectiveness of AQER, using synthetic datasets, classical image and language datasets, and a quantum many-body state datasets with up to 50 qubits. The results show that AQER consistently outperforms existing methods in both accuracy and gate efficiency. Our work paves the way for scalable quantum data processing and real-world quantum computing applications.

## 1 Introduction

Digital quantum computers (Feynman, 1982) promise to deliver computational capabilities that surpass those of classical systems in diverse fields (Shor, 1994; Harrow et al., 2009; Cao et al., 2019; Liu et al., 2021; Huang et al., 2022; Liu et al., 2024). After four decades of exploration (Martinis et al., 1985; Jaksch et al., 2000; Makhlin et al., 2001; Clarke and Wilhelm, 2008; DiCarlo et al., 2009; Isenhower et al., 2010; Arute et al., 2019; Wu et al., 2021; Graham et al., 2022), substantial progress has been achieved, with superconducting (Acharya et al., 2025) and neutral atom platforms (Xu et al., 2025) demonstrating quantum advantages on synthetic benchmarks. Nevertheless, the available quantum resources, such as the number of high-quality qubits and coherence time, would remain severely constrained in the foreseeable future (Jiang et al., 2025). This limitation underscores the need to consistently improve the efficiency of the three foundational modules of quantum computing: quantum state preparation (Girolami, 2019; Gonzalez-Conde et al., 2024), quantum processing (Bharti et al., 2022; Cardama et al., 2025), and readout (Zhang et al., 2021; Schreiber et al., 2025). Maximizing resource utilization across these stages is essential to advancing the practical utility of quantum computers. In this context, efficient quantum state preparation, which amounts to constructing quantum gate sequences that encode classical or quantum inputs into quantum states, emerges as a critical prerequisite (Ranga et al., 2024). This step remains notoriously challenging, as theoretical results indicate that in the worst case, preparing an arbitrary quantum state

---

*Corresponding authors.

within a provable error tolerance may require an exponential number of quantum gates or ancillary qubits (Zhang et al., 2022c; Gui et al., 2024).

Recently, a new concept known as the *approximate quantum loader* (AQL) (Iaconis and Johri, 2023) has emerged, offering a promising direction for efficient quantum state preparation. Unlike earlier approaches that aimed to achieve provable accuracy guarantees, AQL embraces a trade-off between preparation fidelity and circuit complexity. The central insight driving this paradigm is that many quantum algorithms can tolerate some imprecision in the input state, which allows substantial reductions in gate count and resource overhead. For instance, in quantum machine learning, small perturbations in input features often have negligible impact on classification accuracy (Nguyen et al., 2020) and do not compromise the demonstration of quantum advantage, particularly in terms of sample complexity (Huang et al., 2023). Motivated by this observation, considerable effort has been devoted to designing efficient AQL schemes for various data types. These methods broadly fall into two categories: tensor network (TN)–based approaches (Jobst et al., 2024; Iaconis et al., 2024), which employ matrix product state (MPS) and other TN representations, and circuit-based approaches (Mitsuda et al., 2024; Shirakawa et al., 2024), which directly optimize quantum gate sequences (see Fig. 1 for a general framework and Sec. 3.1 for details). Despite this progress, most existing techniques are either heuristic or provide theoretical guarantees only under restrictive scenarios. Consequently, *a systematic understanding of the fundamental limits on approximation error achievable by AQL remains elusive*. Addressing this question would provide critical insights for the principled development of AQL algorithms and for the resource-efficient use of digital quantum computers.

Here we narrow the above knowledge gap by first reformulating a wide range of AQL methods into a unified framework. An intuition is illustrated in Fig. 1, where the AQL construction amounts to identifying a sequence of (tunable) quantum gates that minimizes the distance between the state evolved by this sequence and the target quantum state. Building on this framework, we derive **information-theoretic lower and upper bounds** for the approximation error of AQL (Theorem 3.1), which are independent of specific AQL strategies. Specifically, we demonstrate that the approximation error decreases linearly with the newly proposed entanglement measure, which amounts to the summation of the single-qubit entanglement entropy of the target quantum states after the inverse of the AQL gate sequence. This result provides the key insight that AQL performance can be

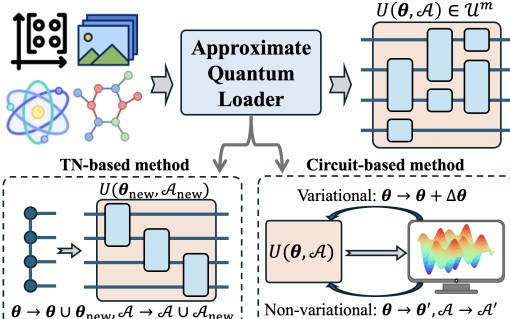

Figure 1: The general framework of AQL. Typical AQLs are separated into two categories: TN–based methods and circuit-based methods, both of which aim to construct quantum circuits from a given gate set $\mathcal{U}$ that approximately prepare the target state.

fundamentally characterized by the degree of entanglement reduction achievable during the AQL, with a larger entanglement reduction leading to more accurate loading.

Motivated by the theoretical importance of entanglement in determining the performance of AQL, we develop AQER, a scalable and efficient AQL method that constructs the gate sequence guided by the principle of maximal Entanglement Reduction. In particular, AQER leverages this principle to reduce the entanglement of target states by progressively adding parameterized single- and two-qubit gates, and employs an explicitly constructed single-qubit gate sequence to further reduce the approximation error. Compared to existing AQL methods, AQER offers **two key advantages**. First, AQER is flexible and universal, supporting efficient approximate loading of both classical data and unknown quantum states. Second, AQER is robust and easy to optimize. Guided by entanglement-reduction optimization, it not only achieves low approximation error but also mitigates vanishing gradient problems during the parameter training (Larocca et al., 2025), thereby ensuring scalability to large-qubit systems. To validate the effectiveness of AQER, we conduct systematic experiments to benchmark its performance on different quantum state loading tasks, ranging from synthetic quantum states, real-world image and language datasets, to quantum many-body systems with up to 50 qubits. The achieved results reveal that AQER consistently outperforms existing methods. Our work paves the way for scalable quantum data processing and practical applications of quantum computing in real-world tasks.

In summary, our contributions are threefold. (i) We propose a unified framework for a wide range of AQL methods and derive two information-theoretic bounds on the approximation error of AQL with respect to the entanglement measure. To the best of our knowledge, this is the first study to establish theoretical limits for AQL from an information-theoretic perspective. (ii) Motivated by these theoretical results, we develop AQER, a scalable and efficient AQL method that features principled entanglement reduction optimization to efficiently utilize gate resources. (iii) We conduct extensive numerical simulations across diverse datasets with up to 50 qubits, and compare AQER against reference AQL methods. The results validate the effectiveness of AQER, demonstrating superior performance with lower approximation error and reduced gate count. The corresponding code is available at GitHub for reproducibility and benchmarking purposes.

## 2 PRELIMINARY

Here, we introduce the basics of quantum computing, quantum entanglement, and variational quantum algorithms, followed by reviewing typical AQL methods. Refer to Appendix A for more details.

**Basics of quantum computing.** The pure *state* of a qubit can be written as $|\psi\rangle = a|0\rangle + b|1\rangle$, where $a, b \in \mathbb{C}$ satisfy $|a|^2 + |b|^2 = 1$, and $|0\rangle = (1, 0)^T, |1\rangle = (0, 1)^T$. The $N$-qubit space is formed by the tensor product of $N$ single-qubit spaces. For a pure state $|\psi\rangle$, the corresponding density matrix is defined as $\rho = |\psi\rangle\langle\psi|$, where $\langle\psi| = (|\psi\rangle)^\dagger$. Mixed states are represented by density matrices of the form $\rho = \sum_k c_k |\psi_k\rangle\langle\psi_k|$, where $c_k \geq 0$ and $\sum_k c_k = 1$. For an $N$-qubit state $|\psi\rangle$, the reduced density matrix of a subsystem $A \subseteq [N]$ is obtained by the partial trace $\rho_A = \mathrm{Tr}_{[N]\setminus A}[|\psi\rangle\langle\psi|]$. A quantum *gate* is represented by a unitary matrix acting on the state, and can be depicted in the circuit model as —□— in quantum circuit notation. Typical quantum operations include fixed gates such as $CZ := \mathrm{diag}(1, 1, 1, -1)$, and tunable single-qubit rotation gates given by $R_\sigma(\theta) = e^{-i\theta\sigma/2}$ with $\sigma \in \{X, Y, Z\}$ and $X = \begin{pmatrix} 0 & 1 \\ 1 & 0 \end{pmatrix}$, $Y = \begin{pmatrix} 0 & -i \\ i & 0 \end{pmatrix}$, $Z = \begin{pmatrix} 1 & 0 \\ 0 & -1 \end{pmatrix}$ being Pauli-X, -Y, -Z operators. The rotation gate could be generalized to the two-qubit case, for example $R_{ZZ}$ with $ZZ = Z \otimes Z$. The quantum *measurement* refers to the procedure of extracting classical information from a quantum state. It is mathematically specified by a Hermitian matrix $H$ called the *observable*. Measuring the state $|\psi\rangle$ or $\rho$ with the observable $H$ yields a random variable whose expectation value is $\langle\psi| H |\psi\rangle$ or $\mathrm{Tr}[H\rho]$, respectively. To quantify the similarity between two quantum states, we use *fidelity* defined as $F = |\langle\psi_1|\psi_2\rangle|^2$ or $F = \mathrm{Tr}[\rho_1\rho_2]$, with the corresponding *infidelity* given by $1 - F$ (Nielsen and Chuang, 2010). Both of them can be evaluated by taking the density matrix of either $|\psi_1\rangle$ or $|\psi_2\rangle$ as the observable.

**Quantum entanglement.** Quantum entanglement is a unique property of quantum systems that distinguishes them from classical systems and serves as a pivotal quantum resource for achieving quantum computational advantages. Specifically, an $N$-qubit quantum state $|\psi\rangle$ of a composite system is called entangled across two subsystems $A, B \subset [N]$ if it cannot be written as a tensor product of states of the subsystems, i.e., $|\psi\rangle \neq |\psi_A\rangle \otimes |\psi_B\rangle$ for any states $|\psi_A\rangle, |\psi_B\rangle$ in the respective subsystems. Various metrics have been developed to quantify the degree of entanglement in quantum states. One commonly used measure is the Renyi-2 entropy, defined as $\mathcal{S}_A(|\psi\rangle) = -\log_2 \mathrm{Tr}[\rho_A^2]$, where $\rho_A$ is the reduced density matrix of $|\psi\rangle$ on the subsystem $A$. When $\mathcal{S}_A(|\psi\rangle) = 0$, the quantum state $|\psi\rangle$ is separable and can be written as a product state $|\psi\rangle = |\psi_A\rangle \otimes |\psi_B\rangle$.

**Variational quantum algorithm (VQA).** VQA (Cerezo et al., 2021a) is a hybrid quantum–classical paradigm in which a parameterized quantum circuit (PQC) is trained by a classical optimizer. It has become a core framework for quantum machine learning (Schuld and Killoran, 2019; Du et al., 2025), with applications spanning discriminative learning, generative learning (Lloyd and Weedbrook, 2018; Gao et al., 2018; Tian et al., 2023), and reinforcement learning (Chen et al., 2020; Cheng et al., 2023; 2024). Concretely, a PQC is written as $V(\boldsymbol{\theta}) = V_m(\boldsymbol{\theta}_m) \cdots V_1(\boldsymbol{\theta}_1)$, and the parameters $\boldsymbol{\theta}$ are optimized to minimize a cost function defined with respect to an input state $\rho_{\mathrm{in}}$ and an observable $O$, i.e., $f(\boldsymbol{\theta}) = \mathrm{Tr}[O V(\boldsymbol{\theta})\rho_{\mathrm{in}} V(\boldsymbol{\theta})^\dagger]$. The parameters are updated by a classical optimizer such as gradient descent or Adam, which requires accessing (exact or estimated) gradients. When the variational circuit is simulated on a classical computer, gradients can be computed directly via automatic differentiation (Bergholm et al., 2018). When executed on a quantum device, gradients are typically estimated by the parameter-shift rule (Wierichs et al., 2022): $\frac{\partial f}{\partial \theta_j} = \frac{1}{2}f(\boldsymbol{\theta}_+) - \frac{1}{2}f(\boldsymbol{\theta}_-)$, where $\boldsymbol{\theta}_\pm$ differ from $\boldsymbol{\theta}$ by $\pm\pi/2$ on the $j$-th parameter.

## 2.1 RELATED WORK

Existing approaches to AQL can be broadly categorized into TN-based and circuit-based methods.

**TN-based methods.** TN-based methods exploit tensor networks to efficiently represent and prepare low-entanglement quantum states. For example, matrix product states (MPS) with bond dimension $k$ can be prepared exactly with $\mathcal{O}(Nk^2)$ two-qubit gates (Schön et al., 2005), and approximate encoding is feasible for states with compact MPS representations (Ran, 2020). In light of this, TN-based strategies have been applied to construct approximate quantum state encodings for **classical data** (Holmes and Matsuura, 2020; Iaconis and Johri, 2023; Jobst et al., 2024; Iaconis et al., 2024). These approaches provide a principled framework with controlled approximation error for low-entanglement inputs, but their utility to quantum data or highly entangled classical data is limited.

**Circuit-based methods.** Circuit-based methods directly optimize the quantum gates that generate the target state, often without relying on an explicit low-entanglement representation. These methods can be broadly categorized into variational and non-variational strategies. Variational approaches train parameterized circuits to minimize infidelity with respect to the target state (Nakaji et al., 2022; Rudolph et al., 2023b; Mitsuda et al., 2024), while non-variational methods iteratively optimize local two-qubit gates along the circuit, progressively improving accuracy without explicit parameter training (Rudolph et al., 2023a; Shirakawa et al., 2024). Circuit-based methods are flexible and broadly applicable, but lack rigorous theoretical guarantees and suffer from barren plateaus.

## 3 APPROXIMATE QUANTUM LOADER

In this section, we first reformulate the main AQL approaches within a unified optimization framework and derive two information-theoretical bounds of the approximation error of AQLs in Sec. 3.1. We then present the implementation details of AQER in Sec. 3.2.

### 3.1 UNIFIED FRAMEWORK OF AQL AND THEORETICAL ANALYSIS

Existing AQL methods can be broadly categorized into *TN-based* methods and *circuit-based* methods. For clarity, we first elucidate a unified framework for AQL and explain how different methods can be formulated within this framework. Building on this framework, we then provide an information-theoretic analysis of the approximation error achievable by AQL.

**A unified framework of AQL.** We begin by recalling the fundamental definition of *approximate quantum loader* (AQL). An AQL amounts to preparing the target state with controllable accuracy by generating quantum gate sequences from a given gate set $\mathcal{U}$, as shown in Fig. 1. Specifically, let $U(\boldsymbol{\theta}; \mathcal{A}) \in \mathcal{U}^m$ denote a circuit composed of $m$ gates, where $\boldsymbol{\theta}$ and $\mathcal{A}$ represent tunable parameters and the circuit architecture, respectively. With the aim of identifying the optimal $\boldsymbol{\theta}$ and $\mathcal{A}$, AQL can be reformulated as *a unified framework* for solving the following optimization problem

$$\arg\min_{\boldsymbol{\theta}, \mathcal{A}} \left[ 1 - |\langle \boldsymbol{v}_{\text{target}} | U(\boldsymbol{\theta}; \mathcal{A}) | \psi_{\text{product}} \rangle|^2 \right], \tag{1}$$

where $|\boldsymbol{v}_{\text{target}}\rangle$ is the target quantum state and $|\psi_{\text{product}}\rangle$ is an easily preparable product state. Within this framework, various AQL methods differ in how they construct the circuit $U(\boldsymbol{\theta}; \mathcal{A})$, either by the way of updating $\boldsymbol{\theta}$, the design of $\mathcal{A}$, or adjusting both. A brief overview of how TN- and circuit-based methods fall into Eq. (1) is provided below, with further details given in Appendix A.3.

*TN-based methods.* For these methods (Ran, 2020), the circuit $U(\boldsymbol{\theta}, \mathcal{A})$ is constructed incrementally by sequentially appending local unitaries obtained from the TN representation. Each unitary is further decomposed into hardware-available gates using the KAK decomposition (Tucci, 2005), which yields the final circuit. In terms of Eq. (1), the optimization proceeds by extending the circuit as $U(\boldsymbol{\theta}; \mathcal{A}) \to U(\boldsymbol{\theta} \cup \boldsymbol{\theta}_{\text{new}}; \mathcal{A} \cup \mathcal{A}_{\text{new}})$, while keeping the previous parameters and architecture fixed.

*Circuit-based methods.* Circuit-based AQL methods can be divided into variational and non-variational approaches. In variational approaches (Nakaji et al., 2022; Rudolph et al., 2023b; Mitsuda et al., 2024), the circuit $U(\boldsymbol{\theta}; \mathcal{A})$ is constructed by optimizing a variational quantum circuit with cost function $\ell(\boldsymbol{\theta}) = 1 - |\langle \boldsymbol{v}_{\text{target}} | U(\boldsymbol{\theta}; \mathcal{A}) | \psi_{\text{product}} \rangle|^2$, where $\boldsymbol{\theta}$ are tunable and $\mathcal{A}$ is fixed. By contrast, non-variational approaches (Rudolph et al., 2023a; Shirakawa et al., 2024) gradually update all two-qubit unitaries in a prescribed circuit following a zigzag schedule. This involves sequentially

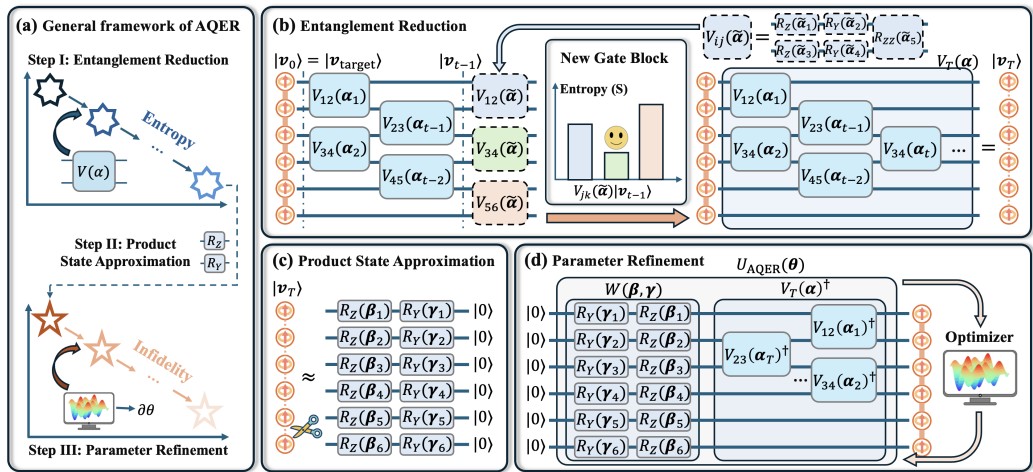

Figure 2: The workflow of the AQER algorithm. (a) An overview of AQER, which consists of three steps. (b) *Step I: entanglement reduction*. This step iteratively appends two-qubit gate blocks to progressively reduce the entanglement of the input $|v_{\text{target}}\rangle$ with the circuit $V_T(\boldsymbol{\alpha})$. (c) *Step II: product state approximation*. This step approximates the low-entanglement state $|v_T\rangle$ by applying single-qubit rotations $\{R_Z(\boldsymbol{\beta}_n)\}_{n=1}^N$ and $\{R_Y(\boldsymbol{\gamma}_n)\}_{n=1}^N$ to the initial state $|0\rangle^{\otimes N}$. (d) *Step III: parameter refinement*. This step finetunes all circuit parameters in $\boldsymbol{\theta} = (\boldsymbol{\alpha}, \boldsymbol{\beta}, \boldsymbol{\gamma})$ to minimize the infidelity and obtain the final AQL $U_{\text{AQER}}(\boldsymbol{\theta}^*)$.

adjusting both the parameters $\boldsymbol{\theta}$ and the architecture $\mathcal{A}$ of the circuit $U(\boldsymbol{\theta}; \mathcal{A})$, and thus fits within the general optimization framework in Eq. (1).

**Information-theoretic analysis.** A benefit of the unified framework in Eq. (1) is that it enables an algorithm-independent theoretical analysis of AQL. Here, we establish two information-theoretical bounds of the approximation error achievable by AQL using an entanglement measure, as stated in the following theorem with the proof deferred to Appendix B.2.

**Theorem 3.1.** *Denote the entanglement measure for an $N$-qubit state $|\psi\rangle$ as $\mathcal{S}(|\psi\rangle) = \sum_{i=1}^N \mathcal{S}_{\{i\}}(|\psi\rangle)$. Then, for the state $|v_{\text{target}}\rangle$ and a circuit $U$ with $\mathcal{S}(U^\dagger|v_{\text{target}}\rangle) = S$, the infidelity between $|v_{\text{target}}\rangle$ and the state generated from $U$ on any product state $|\psi_{\text{product}}\rangle$ is lower bounded as $1 - |\langle v_{\text{target}}|U|\psi_{\text{product}}\rangle|^2 \geq f_1(S) := \frac{1}{2}\left(1 - \sqrt{2^{1-S/N} - 1}\right)$. Moreover, given access to $\rho$, we can construct a product state $|\psi'_{\text{product}}\rangle$ such that the infidelity is upper bounded as $1 - |\langle v_{\text{target}}|U|\psi'_{\text{product}}\rangle|^2 \leq f_2(S) := \frac{1}{2}\left(1 - \sqrt{2^{1-S+\lfloor S\rfloor} - 1} + \lfloor S\rfloor\right)$. When $S \to 0$, $f_1(S) \to \frac{\ln 2}{2N}S + \mathcal{O}(S^3)$ and $f_2(S) \to \frac{\ln 2}{2}S + \mathcal{O}(S^3)$.*

The above results indicate that the approximation error of AQL is fundamentally governed by the entanglement measure $\mathcal{S}$ of the evolved states $U^\dagger|v_{\text{target}}\rangle$. In particular, the infidelity between the target state $|v_{\text{target}}\rangle$ and the prepared state $U|\psi_{\text{product}}\rangle$ scales linearly with the entanglement measure value $S$. Hence, a smaller $S$ guarantees lower infidelity and smaller approximation error. Since $S$ depends on both $|v_{\text{target}}\rangle$ and the circuit $U$, reducing infidelity through parameter and architecture optimization in AQL is equivalent to minimizing the entanglement measure $\mathcal{S}$.

**Remark.** Theorem 3.1 can be generalized to noisy channel cases as provided in Appendix C.

## 3.2 AQER: AN EFFICIENT AND SCALABLE AQL

The theoretical role of the entanglement measure $\mathcal{S}$ in determining the approximation error suggests that it can serve as a practical indicator of the quality of an AQL. Driven by this insight, we propose AQER, an efficient and scalable AQL that solves the optimization problem in Eq. (1) by constructing the gate sequence guided by the principle of maximal entanglement reduction. In the remainder of this subsection, we present the implementation of AQER.

**Overview of AQER:** As illustrated in Fig. 2(a), AQER consists of three key components: (I) suppressing the entanglement measure $\mathcal{S}$ in the input data by incrementally adding quantum gates; (II) applying single-qubit rotation gates for correcting leading errors; and (III) optimizing the circuit parameters for further refinement. Next, we explain these procedures separately.

*Step I: Entanglement Reduction.* The goal of this step is to construct a gate sequence that reduces the entanglement of the target input by monitoring the entanglement measure $\mathcal{S}$. As illustrated in Fig. 2(b), this is achieved by **iteratively** appending two-qubit gate blocks $V_{\mathcal{I}_t}(\boldsymbol{\alpha}_t)$, where the tunable variables include the acting qubit pair $\mathcal{I}_t = (j_t, k_t)$ with $1 \leq j_t \neq k_t \leq N$ and the gate parameters $\boldsymbol{\alpha}_t$. Specifically, the blocks $\{V_{\mathcal{I}_t}(\boldsymbol{\alpha}_t)\}$ have the identical structure $R_{ZZ}R_Y R_Z$ with single-qubit rotation gates applied to both qubits. Let $V_{t-1}(\boldsymbol{\alpha}_{1:t-1}) = V_{\mathcal{I}_{t-1}}(\boldsymbol{\alpha}_{t-1}) \cdots V_{\mathcal{I}_1}(\boldsymbol{\alpha}_1)$ be the gate sequence generated after $(t-1)$ iterations. At the $t$-th iteration, the acting qubit pair $\mathcal{I}_t$ and gate parameters $\boldsymbol{\alpha}_t$ of the block $V_{\mathcal{I}_t}(\boldsymbol{\alpha}_t)$ are obtained by solving the following optimization problem

$$\mathcal{I}_t, \boldsymbol{\alpha_t} = \arg \min_{\tilde{\mathcal{I}}, \tilde{\boldsymbol{\alpha}}} \mathcal{S}\left(V_{\tilde{\mathcal{I}}}(\tilde{\boldsymbol{\alpha}})|\boldsymbol{v}_{t-1}\rangle\right), \tag{2}$$

where $\mathcal{S}$ is the entanglement measure defined in Theorem 3.1 and $|\boldsymbol{v}_{t-1}\rangle := V_{t-1}(\boldsymbol{\alpha}_{1:t-1})|\boldsymbol{v}_{\text{target}}\rangle$. This iterative process is repeated for $T$ times such that the entanglement of the state $|\boldsymbol{v}_T\rangle = V_T(\boldsymbol{\alpha})|\boldsymbol{v}_{\text{target}}\rangle$ is sufficiently small, where $\boldsymbol{\alpha} := \boldsymbol{\alpha}_{1:T} = (\boldsymbol{\alpha}_1, \cdots, \boldsymbol{\alpha}_T)$.

*Step II: Product State Approximation.* After Step I, AQER aims to further suppress the approximation error by applying single-qubit rotation gates, as illustrated in Fig. 2(c). The motivation for this step is that, as suggested in Theorem 3.1, the state $|\boldsymbol{v}_T\rangle$ with low entanglement can be well approximated by a product state. To prepare such a product state from a standard initial state $|0\rangle^{\otimes N}$, we apply additional single-qubit rotations $W(\boldsymbol{\beta}, \boldsymbol{\gamma}) = \otimes_{i=1}^{N}(R_Z(\beta_i)R_Y(\gamma_i))$. Note that the optimal parameters $\boldsymbol{\beta} = (\beta_1, \cdots, \beta_N)$ and $\boldsymbol{\gamma} = (\gamma_1, \cdots, \gamma_N)$ can be **explicitly derived** without numerical optimization, as indicated in the corollary below. Refer to Appendix B.1 for the explicit form and derivation.

**Corollary 3.2** (informal). *Given constant access to the quantum state $|\boldsymbol{v}_T\rangle$ with reduced entanglement, the explicit form of each parameter in $(\boldsymbol{\beta}, \boldsymbol{\gamma})$ can be derived without optimization.*

*Step III: Parameter Refinement.* The final step of AQER further enhances the accuracy by fine-tuning the parameters of the gate sequences constructed in Steps I and II, as illustrated in Fig. 2(d). For clarity, we denote the combined circuit as $U_{\text{AQER}}(\boldsymbol{\theta}) = V_T(\boldsymbol{\alpha})^{\dagger}W(\boldsymbol{\beta}, \boldsymbol{\gamma})$, where $\boldsymbol{\theta} = (\boldsymbol{\alpha}, \boldsymbol{\beta}, \boldsymbol{\gamma})$ collects all tunable parameters. Parameter refinement then amounts to minimizing the infidelity between the target state $|\boldsymbol{v}_{\text{target}}\rangle$ and the state $U_{\text{AQER}}(\boldsymbol{\theta})|0\rangle^{\otimes N}$, i.e.

$$\boldsymbol{\theta}^* = \arg \min_{\boldsymbol{\theta}} \left(1 - |\langle \boldsymbol{v}_{\text{target}}|U_{\text{AQER}}(\boldsymbol{\theta})|0\rangle^{\otimes N}|^2\right). \tag{3}$$

After the optimization of Eq. (3), we obtain the AQL: $|\boldsymbol{v}_{\text{load}}\rangle = e^{-ig}U_{\text{AQER}}(\boldsymbol{\theta}^*)|0\rangle^{\otimes N}$, where $g$ is a global phase that does not affect any measurement outcomes.

**Remark.** (i) An important feature of AQER is the usage of the entanglement measure $\mathcal{S}$ as a proxy for the approximation error. For quantum data, evaluating and optimizing $\mathcal{S}$ is efficient since it involves only local measurements. For classical data, AQER can be simulated classically to construct $U_{\text{AQER}}$. (ii) By first suppressing the entanglement measure $\mathcal{S}$, AQER not only reduces the approximation error but also distinguishes itself from prior circuit-based methods by mitigating barren plateau issues, thereby enhancing trainability and scalability. See Appendices D and G for additional discussion and the time-complexity analysis of AQER. (iii) In general, AQER is a heuristic algorithm. However, for certain structured vector or state families, AQER admits efficient performance guarantees. In particular, as shown in Appendix H, AQER provably generates an optimal loading circuit for IQP states with polynomial resource cost.

## 4 EXPERIMENTS

We conduct extensive numerical simulations to evaluate the performance of AQER in loading both classical and quantum datasets. Further implementation details and additional results are provided in Appendices E and F, respectively.

### 4.1 DATASET CONSTRUCTION FOR AQL

We briefly introduce datasets used in this work with more details in Appendix E.1.

Table 1: Infidelity (↓) of different AQL methods on MNIST, CIFAR-10, SST-2, S-RQC, and GS-TFIM datasets. We compare AQER with $G \in \{20, 40, 80\}$ against reference methods, where the latter use equal or slightly larger $G$ due to feasibility constraints detailed in Appendix E.2. Values are reported with the mean (and standard deviation) over $M$ samples. The best and second-best results are highlighted in blue and orange, respectively.

| | MNIST | | | CIFAR-10 | | | SST-2 | | | S-RQC | | | GS-TFIM | | |
|---|---|---|---|---|---|---|---|---|---|---|---|---|---|---|---|
| MPS | $G$ 36 | 54 | 90 | $G$ 30 | 60 | 90 | $G$ 36 | 54 | 90 | $G$ 27 | 54 | 81 | $G$ 36 | 72 | 90 |
| | 0.330 (0.101) | 0.287 (0.089) | 0.237 (0.076) | 0.068 (0.038) | 0.056 (0.031) | 0.049 (0.028) | 0.901 (0.022) | 0.870 (0.022) | 0.814 (0.022) | 0.746 (0.100) | 0.663 (0.118) | 0.605 (0.127) | 0.056 (0.004) | 0.039 (0.003) | 0.041 (0.002) |
| HEC | $G$ 20 | 40 | 80 | $G$ 22 | 44 | 88 | $G$ 20 | 40 | 80 | $G$ 20 | 40 | 80 | $G$ 20 | 40 | 80 |
| | 0.430 (0.089) | 0.234 (0.079) | 0.103 (0.042) | 0.081 (0.039) | 0.050 (0.026) | 0.030 (0.016) | 0.892 (0.016) | 0.759 (0.012) | 0.546 (0.007) | 0.731 (0.097) | 0.484 (0.133) | 0.367 (0.161) | 0.168 (0.090) | 0.020 (0.013) | 0.007 (0.002) |
| AQCE | $G$ 20 | 40 | 80 | $G$ 30 | 45 | 90 | $G$ 20 | 40 | 80 | $G$ 30 | 45 | 90 | $G$ 20 | 40 | 80 |
| | 0.296 (0.083) | 0.145 (0.053) | 0.051 (0.023) | 0.068 (0.036) | 0.048 (0.026) | 0.024 (0.014) | 0.891 (0.017) | 0.761 (0.018) | 0.518 (0.013) | 0.534 (0.149) | 0.363 (0.156) | 0.267 (0.110) | 0.108 (0.026) | 0.068 (0.024) | 0.056 (0.015) |
| AQER (Ours) | $G$ 20 | 40 | 80 | $G$ 20 | 40 | 80 | $G$ 20 | 40 | 80 | $G$ 20 | 40 | 80 | $G$ 20 | 40 | 80 |
| | 0.195 (0.060) | 0.090 (0.034) | 0.034 (0.015) | 0.043 (0.023) | 0.029 (0.016) | 0.018 (0.010) | 0.819 (0.017) | 0.652 (0.013) | 0.406 (0.008) | 0.285 (0.152) | 0.128 (0.106) | 0.067 (0.069) | 0.028 (0.011) | 0.009 (0.006) | 0.003 (0.001) |

**Classical data.** We use three standard classical datasets: MNIST (Lecun et al., 1998), CIFAR-10 (Krizhevsky et al., 2009), and SST-2 (Socher et al., 2013). These datasets have been widely employed to benchmark AQL methods on diverse data types across vision and language tasks. Specifically, MNIST contains $28 \times 28$ grayscale handwritten digits; CIFAR-10 includes $32 \times 32$ RGB images of natural objects such as cats and cars; and SST-2 is a sentiment classification dataset, for which we use a pretrained Sentence-BERT model (Reimers and Gurevych, 2019) to obtain 1024-dimensional sentence embeddings. Each dataset is preprocessed into $M = 50$ normalized vectors, which are then encoded as target states using either amplitude encoding $|\boldsymbol{v}\rangle = \sum_{j=1}^{2^N} \boldsymbol{v}_j |j\rangle$ or compact encoding (Blank et al., 2022) $|\boldsymbol{v}\rangle = \sum_{j=1}^{2^N} (\boldsymbol{v}_j + \imath \boldsymbol{v}_{j+2^N}) |j\rangle$ with $N \in \{10, 11\}$ qubits.

**Quantum data.** We construct two types of quantum datasets: (i) synthetic states generated from random quantum circuits (RQCs); and (ii) ground states of the one-dimensional transverse-field Ising model (1D TFIM) (Pfeuty, 1970), denoted as S-RQC and GS-TFIM, respectively. These datasets represent typical examples of states arising from quantum circuit evolutions and many-body quantum systems. S-RQC contains $M = 50$ states generated by applying different RQCs to the state $|0\rangle^{\otimes N}$. Each RQC is sampled from the set $\textit{RandomShuffle} \left( \{CZ_{p_k,q_k}\}_{k=1}^{W} \cup \{R_{p_k}\}_{k=W+1}^{4W} \right)$, which includes $W$ $CZ$ gates and $3W$ single-qubit rotations on randomly chosen qubits $\{p_k, q_k\}$. Here, we set $N = 10$ and $N_{\text{RQC}} = 40$. GS-TFIM contains ground states of the 1D TFIM Hamiltonian $H = -\sum_{i=1}^{N-1} J Z_i Z_{i+1} - \sum_{i=1}^{N} g X_i$. We consider coefficients $g = 1$ and $J \in \{0.8, 0.9, 1, 1.1, 1.2\}$ to construct datasets of size $M = 5$ for each $N \in \{10, 20, 30, 40, 50\}$.

## 4.2 EXPERIMENTAL SETTINGS

**Reference AQL methods.** We select three typical reference AQL methods to provide a comprehensive comparison with AQER. The three reference methods are: (i) TN method based on 1D MPS (Iaconis and Johri, 2023), which represents approaches with guarantees on low-entanglement data; (ii) hardware-efficient circuit (HEC)-based method (Nakaji et al., 2022), where the circuit is commonly used in VQAs; (iii) automatic quantum circuit encoding (AQCE) (Shirakawa et al., 2024), which illustrates recent advances in non-variational AQL.

**Evaluation metrics.** To quantify the accuracy of an AQL, we use infidelity $1 - |\langle \boldsymbol{v}_{\text{load}} | \boldsymbol{v}_{\text{target}} \rangle|^2$ as the measure of approximation error. The infidelity ranges from 0 to 1, with *smaller values indicating lower approximation error and higher accuracy*. To evaluate the efficiency of an AQL, we consider the quantum resources to prepare the state from the encoding circuit. In particular, we record the number of employed two-qubit gates (e.g., $CZ$ and $R_{ZZ}$) in the encoding circuit. This quantity, denoted by $G$, serves as the measure of quantum resource consumption, since it dominates the circuit runtime (Ma and Li, 2024). A smaller $G$ corresponds to a more efficient AQL.

**Hyperparameter settings of AQER.** The number of iterations in Step I of AQER is set to $T \in \{5, 10, 20, 40, 60, 80, 100\}$ by default. For GS-TFIM with large qubit numbers ($N \geq 20$), we use

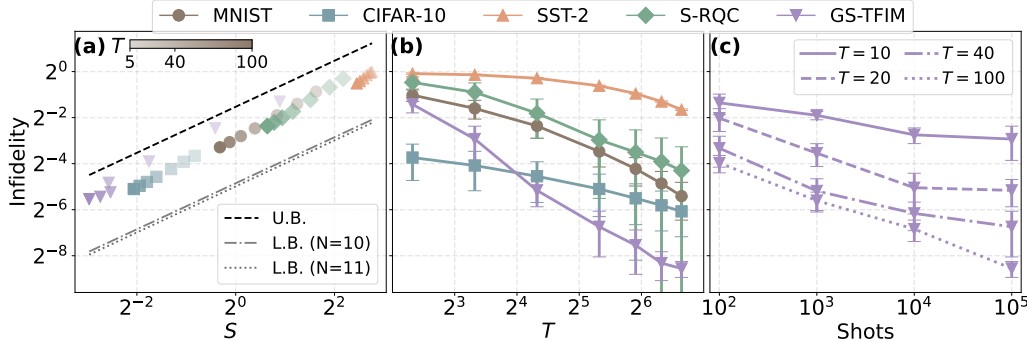

Figure 3: Performance of AQER across MNIST, CIFAR-10, SST-2, S-RQC, and GS-TFIM datasets, distinguished by different colors and markers. (a) Infidelity versus the entanglement measure value $S$ after Step II of AQER, averaged over $M$ samples. Color bars from light to dark indicate increasing $T$. Dashed lines indicate the linearized upper (U.B.) and lower (L.B.) bounds in Theorem 3.1, which neglect higher-order terms. (b) Infidelity versus different $T$ values after Step III of AQER across all datasets. (c) Infidelity versus different measurement shots for the GS-TFIM dataset, with different $T \in \{10, 20, 40, 100\}$.

$T \in \{20, 40, 60, 80, 100, 120, 160, 200\}$, since larger systems generally require more gates to capture quantum correlations. In AQER, the iteration count $T$ controls the two-qubit gate count $G$, with one iteration introducing one two-qubit gate. In the $t$-th iteration of Step I, the parameters $\boldsymbol{\alpha}_t$ in Eq. (2) are first initialized to zero and then optimized using the Nelder–Mead method with a convergence tolerance of $10^{-4}$. The qubit index set $\mathcal{I}_t$ is optimized by selecting the qubit pair that minimizes $S$ through adjusting $\boldsymbol{\alpha}_t$. Step III performs optimization using the Adam optimizer with a learning rate of $10^{-2}$ for $T_3 = 2000$ iterations. For quantum datasets, quantities such as $S$ and gradients are estimated from $10^5$ simulated measurement shots by default.

## 4.3 EXPERIMENTAL RESULTS

We evaluate AQER on both classical and quantum datasets to verify its accuracy and efficiency, as well as its trainability and scalability on large systems, and to compare its performance against existing AQL methods. Additional numerical results are provided in Appendix F.

**AQER outperforms all reference AQL methods for both classical and quantum data.** We first compare the performance of various AQL methods by measuring their approximation errors (infidelity) for different two-qubit gate counts $G$ across multiple datasets. Table 1 lists these results for MNIST, CIFAR-10, SST-2, S-RQC, and GS-TFIM (with $N = 10$). Results for AQER are shown with $G \in \{20, 40, 80\}$. For the referenced AQL methods, $G$ is set to the same or slightly larger values, as determined by the feasibility constraints explained in Appendix E.2. It can be observed that AQER consistently surpasses existing AQL methods by achieving the lowest infidelity with the same or even smaller $G$. The most pronounced improvement is observed on S-RQC, where AQER reduces the infidelity by more than $60\%$ relative to the second-best method (AQCE) for $G \in \{40, 80\}$, and further achieves lower infidelity while using $50\%$ fewer two-qubit gates than other methods. These results validate the advantage of AQER, which achieves lower infidelity with equal or even fewer two-qubit gates than existing AQL methods.

**AQER consistently decreases the infidelity by reducing the entanglement measure $\mathcal{S}$.** We next examine the effectiveness of the entanglement entropy reduction of Step I in decreasing the approximation error of AQER. To do this, we record the value of the entanglement measure $S$ in Step I with different settings of the iteration times $T \in \{5, 10, 20, 40, 60, 80, 100\}$, and also record the corresponding approximation error after Step II. These results are shown in Fig. 3(a). For all datasets, increasing $T$ generally shifts points toward lower $S$ and infidelity, which stay within the theoretical upper and lower bounds given by Theorem 3.1, demonstrating that AQER progressively expands the circuit to reduce entanglement and improve AQL accuracy. For example, increasing $T$ from 5 to 100 reduces $S$ by roughly fourfold for the MNIST dataset, with the corresponding infidelity decreasing by a similar factor, which validates the effect of entanglement reduction on lowering infidelity.

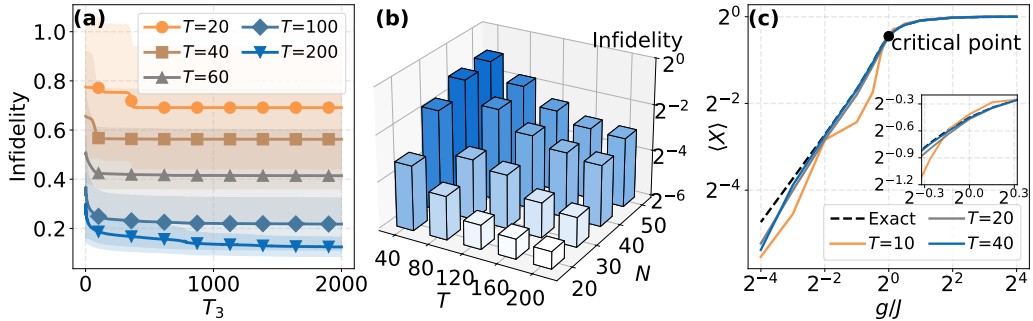

Figure 4: Performance of AQER on the GS-TFIM dataset. (a) Infidelity during Step III optimization across different $T$ values with $N = 50$ qubits. (b) Infidelity for different qubit numbers $N$ and Step I iteration times $T$. (c) Expectation values of the averaged magnetization $\langle X \rangle$ measured on AQER-loaded GS-TFIM states for different $g/J$ values with $N = 10$. Each curve corresponds to a different $T \in \{10, 20, 40\}$.

**Effect of two-qubit gate count and shot number on AQER performance.** We further investigate how the number of two-qubit gates and measurement shots influence AQER. In particular, we conduct experiments with varying $T \in \{5, 10, 20, 40, 60, 80, 100\}$ and shots in $\{10^2, 10^3, 10^4, 10^5\}$. The infidelity versus varying $T$ after Step III of AQER for different datasets is shown in Fig. 3(b), while the effect of different measurement shots for the GS-TFIM dataset is illustrated in Fig. 3(c). Larger $T$ values lead to a significant reduction in infidelity. For example, increasing $T$ from 5 to 100 decreases the infidelity for the GS-TFIM dataset from above $2^{-2}$ to below $2^{-8}$. Similarly, increasing the number of shots generally reduces infidelity by suppressing statistical noise in circuit generation and optimization. This effect is more pronounced for larger $T$, with the reduction being less than 4 for $T = 10$ and more than 16 for $T = 100$. These results quantitatively demonstrate that larger circuits combined with sufficient measurement shots effectively improve AQER performance.

**The trainability of AQER.** We then demonstrate the trainability of AQER on large systems via experiments on the GS-TFIM dataset with $N = 50$ qubits. The parameter optimization in Step III of AQER for varying $T \in \{20, 40, 60, 100, 200\}$ is shown in Fig. 4(a). The optimization curves do not exhibit barren plateaus, which would otherwise trap the process at high infidelity near 1. The initial infidelity is already far from 1, consistent with Theorem 3.1. For instance, with $T = 200$, the infidelity starts around 0.3 and decreases effectively to around 0.1. These results demonstrate that the entanglement-reduction mechanism in AQER successfully mitigates barren plateau effects in Step III, ensuring trainability and serving as a prerequisite for its scalability.

**Scalability of AQER.** To validate the scalability of AQER, we conduct experiments on the GS-TFIM dataset with varying qubit numbers $N \in \{20, 30, 40, 50\}$ and $T \in \{40, 80, 120, 160, 200\}$. As illustrated in Fig. 4(b), for all system sizes, infidelity consistently decreases as $T$ increases. In particular, AQER maintains roughly constant infidelity across different $N$ when $T$ scales linearly with $N$, specifically following $T = 4N - 40$, highlighting favorable scalability with respect to both qubit number and two-qubit gate count.

**Downstream performance of AQER.** Finally, we evaluate the performance of AQER in downstream tasks on both quantum and classical datasets. We begin with

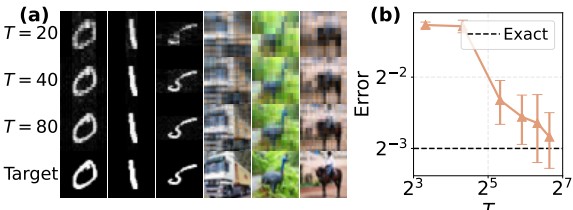

Figure 5: Downstream performance of AQER on classical data. (a) Reconstructed MNIST and CIFAR-10 images using AQER with $T \in \{20, 40, 80\}$. (b) Classification error on the SST-2 dataset using AQER-loaded states with $T \in \{10, 20, 40, 60, 80, 100\}$, compared to exact loading (black dashed line).

the detection of quantum phase transitions in the TFIM, which is measured by the averaged magnetization $\langle X \rangle := \frac{1}{N} \sum_{n=1}^{N} \langle X_n \rangle$ of the ground state as the order parameter. We record the values of $\langle X \rangle$ for the AQER-loaded states with varying $T$ values as a function of $g/J$ as shown in Fig. 4(c).

These results demonstrate that AQER-loaded states can capture the transition from the ferromagnetic phase with $g/J < 1$ to the paramagnetic phase with $g/J > 1$, with a rapid change near the critical point at $g/J = 1$. Increasing $T$ generally results in a more accurate approximation of exact values, indicating that larger circuits can encode more correlations relevant for the order parameter. Notably, even states with moderate $T = 10$ capture the overall trend, suggesting that AQER-loaded states can effectively capture quantum phase transitions with limited quantum resources. Next, for classical data, we reconstruct MNIST and CIFAR-10 images with AQER using varying $T$ values. As illustrated in Fig. 5(a), reconstructions approach the targets as $T$ increases. We also evaluate binary classification on SST-2 using a quantum kernel method with details in Appendix E.3. As shown in Fig. 5(b), the error decreases with larger $T$ and approaches near the exact-loading error of $2^{-3}$ at $T = 100$. These results demonstrate improved downstream performance with larger $T$.

## 5 CONCLUSION

In this work, we introduced a unified framework for AQLs and derived information-theoretic bounds showing that the infidelity is fundamentally controlled by the entanglement of quantum states. Based on this insight, we proposed AQER, a scalable and efficient method that systematically reduces entanglement to achieve low infidelity with efficient gate usage. Extensive benchmarks on classical and quantum datasets show that AQER consistently outperforms existing methods in both accuracy and circuit efficiency. These results provide both theoretical guarantees and a practical approach for efficient quantum data loading, enabling broader applications in data-dependent quantum algorithms.

## ACKNOWLEDGMENTS

This project is supported by the National Research Foundation, Singapore, under its NRF Professorship Award No. NRF-P2024-001.

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

# A PRELIMINARY

## A.1 NOTATIONS

Here, we unify the notations used throughout this manuscript. We denote by $[N]$ the set $\{1, \cdots, N\}$. The symbol $\boldsymbol{a}_j$ denotes the $j$-th component of a vector $\boldsymbol{a}$. The tensor product operation is denoted as "$\otimes$". The conjugate transpose of a matrix $A$ is denoted as $A^\dagger$. The trace of a matrix $A$ is denoted as $\mathrm{Tr}[A]$. The notation $\lfloor x \rfloor$ denotes the largest integer that is smaller than or equal to $x$. We employ $\mathcal{O}$ to describe complexity notions. The phase of a complex value $x$ is denoted by $\arg(x)$.

For vectors, the notation $\| \cdot \|_2$ represents the $\ell_2$ norm. We use Schatten norms for operators: the trace norm $\|X\|_1 := \mathrm{Tr}\sqrt{X^\dagger X}$ and the operator norm $\|X\|_\infty := \sup_{\|\psi\|_2=1} \|X\psi\|_2$, which equals the largest singular value of $X$. In particular, any density operator $\rho$ satisfies $\|\rho\|_1 = 1$ and $\|\rho\|_\infty \leq 1$.

For a linear map (quantum channel) $\Phi$ acting on operators, we denote its diamond norm by $\|\Phi\|_\diamond$, $\|\Phi\|_\diamond := \sup_{d \geq 1} \sup_{X \neq 0} \frac{\|(\Phi \otimes \mathrm{id}_d)(X)\|_1}{\|X\|_1}$, where $\mathrm{id}_d$ denotes the identity channel on a $d$-dimensional ancilla system. We use $I$ (or $I_d$) to denote the identity matrix and $\mathrm{id}$ to denote the identity channel on the underlying system. A quantum channel $\mathcal{E}$ is completely positive and trace-preserving (CPTP).

## A.2 RELATED WORK

The problem of quantum data loading has been extensively studied across multiple lines of research. For example, early investigations established that exact amplitude loading of an arbitrary $N$-qubit pure state requires exponentially many quantum resources. Specifically, it was shown that without assuming prior structure, $\mathcal{O}(2^N)$ single- and two-qubit gates are necessary for generic state preparation (Grover, 2000; Kaye and Mosca, 2001; Long and Sun, 2001; Bergholm et al., 2005; Plesch and Brukner, 2011). This scaling matches the intrinsic complexity of generic states: an $N$-qubit pure state corresponds to a normalized vector in $\mathbb{C}^{2^N}$ with $2^{N+1} - 1$ degrees of freedom. Even with auxiliary qubits introduced to reduce circuit depth (Sun et al., 2023), the total gate count remains exponential. For structured instances, more efficient constructions are possible: for instance, an $S$-sparse vector can be exactly loaded using $\mathcal{O}(SN)$ CNOT gates (Gleinig and Hoefler, 2021). However, most states derived from classical datasets, such as images (Lecun et al., 1998; Krizhevsky et al., 2009), are neither sparse nor unstructured, motivating approximate approaches.

**Tensor-Network (TN)-Based Approximate Loading.** Tensor-network methods leverage low-entanglement structure to efficiently represent quantum states and construct corresponding loading circuits. The most widely applied TN family for quantum data loading is the matrix product state (MPS), which provides a sequential representation of an $N$-qubit state using a bond dimension $k$. Given an MPS, one can construct a circuit of $\mathcal{O}(Nk^2)$ two-qubit gates that exactly prepares the state (Schön et al., 2005), while truncated singular value decomposition (SVD) allows controlled approximation for more general target states (Schollwöck, 2011; Ran, 2020). Recent works have applied MPS-based methods to both synthetic distributions and real-world classical datasets. For example, Iaconis and Johri (2023) presented a method to convert MPS representations of images into quantum circuits with logarithmic scaling in the number of pixels, experimentally demonstrating amplitude encoding of complex images on a trapped-ion device. Jobst et al. (2024) exploited the decaying Fourier spectrum of classical images to construct approximate MPS encodings, which reduce quantum circuit complexity and can be further refined using simple sequential circuits inspired by MPS structure. Other works have focused on efficiently preparing real-valued smooth probability distributions with linear-depth circuits derived from MPS (Holmes and Matsuura, 2020; Iaconis et al., 2024). These procedures combine classical preprocessing (e.g., Fourier feature compression) with MPS-based circuit synthesis to achieve high-fidelity state loading suitable for near-term quantum hardware. While TN-based methods offer principled and resource-efficient constructions for low-entanglement states, their applicability is limited when the target state exhibits high entanglement or strong long-range correlations, as the required bond dimension, which corresponds to the circuit depth and two-qubit gate count, can grow rapidly.

**Circuit-based Approximate Loading.** Circuit-based methods directly construct quantum circuits to approximate a target state. These methods can be broadly categorized into variational and non-variational approaches.

*Variational approaches.* Variational methods employ parameterized quantum circuits (PQCs), such as hardware-efficient or sequential ansatzes, and optimize their parameters to minimize infidelity with respect to the target state (Nakaji et al., 2022; Mitsuda et al., 2024; Rudolph et al., 2023b; Wang et al., 2024a). For real-valued data vectors, classical routines and measurements in the computational basis and, when necessary, in the Hadamard-transformed basis are used to capture both amplitude and sign information (Nakaji et al., 2022). For complex-valued data vectors, fidelity can be used directly as the cost function, with classical shadow techniques employed to efficiently estimate the fidelity and its gradient during optimization (Mitsuda et al., 2024). Moreover, hybrid strategies are proposed to utilize TN-based initialization: the target state is first approximated using a tensor-network representation (e.g., MPS), mapped to a parameterized circuit, and then variationally fine-tuned (Rudolph et al., 2023b). This approach retains the structural advantages of TNs while extending applicability beyond strictly low-entanglement states.

*Non-variational approaches.* Non-variational methods iteratively refine local gates to approximate the target state. Shirakawa et al. (Shirakawa et al., 2024) introduced the AQCE algorithm, which sequentially updates two-qubit unitaries to improve fidelity. Rudolph et al. (Rudolph et al., 2023a) enhanced this framework by initializing AQCE with an MPS approximation rather than a trivial identity, demonstrating faster convergence and higher accuracy. Hybrid workflows that start from an MPS-based circuit and perform local iterative refinement combine the controlled approximation of TNs with the adaptive flexibility of circuit-level updates, providing efficient and accurate loading for moderately entangled datasets.

Despite their flexibility and broad applicability, they can suffer from barren plateaus and optimization traps, and unfavorable scaling for large qubit numbers or highly entangled target states (You and Wu, 2021; Wang et al., 2023; You et al., 2022; Liu et al., 2023; You et al., 2023). Moreover, circuit-based methods remain heuristic. Rigorous theoretical guarantees on fidelity and resource costs are generally absent, limiting systematic design and predictability for practical quantum data loading.

## A.3    OVERVIEW OF TYPICAL AQL METHODS

In this section, we summarize several representative methods for approximate quantum loading (AQL) and show how they can be unified within the general AQL framework introduced in Section 3.1 in the main text. Recall that in this framework, an AQL procedure constructs a quantum circuit $U(\boldsymbol{\theta}; \mathcal{A})$ that loads the target state $|\boldsymbol{v}_{\text{target}}\rangle$ from an initial product state $|\psi_{\text{product}}\rangle$, and optimizes it by adjusting the parameters $\boldsymbol{\theta}$, or the architecture $\mathcal{A}$, or both.

### A.3.1    TENSOR-NETWORK (TN) BASED METHODS

Here we introduce the AQL method based on the matrix product state (MPS) (Ran, 2020), which forms a one-dimensional tensor network. The MPS method iteratively constructs an MPS approximation of the current state with bond dimension 2. From this approximation, a sequence of $N-1$ two-qubit unitaries is obtained, which are inverted and applied to the current state to generate an updated state. Specifically, in the $(i+1)$-th iteration, the MPS method considers the current state

$$|\psi_{\text{MPS}}^{(i)}\rangle = G_{(N-1)i}^{\dagger} \cdots G_1^{\dagger} |\boldsymbol{v}_{\text{target}}\rangle, \tag{4}$$

from which a new set of two-qubit unitaries

$$G_{(N-1)(i+1)}, \ldots, G_{(N-1)i+1} \tag{5}$$

is extracted and appended to the existing circuit. By repeating this procedure sufficiently many times, the state in Eq. (4) approaches the product state $|0\rangle^{\otimes N}$. Consequently, the collection $\{G_i\}$ defines the encoding circuit for AQL. We note that each two-qubit unitary $G_i$ is further decomposed into hardware-native gates via the KAK decomposition (Tucci, 2005), resulting in 2 CNOT (or CZ) gates for real matrices and 3 gates for general complex matrices.

Within our framework, MPS-based AQL corresponds to sequentially updating both the circuit architecture and parameters:

$$U(\boldsymbol{\theta}; \mathcal{A}) \rightarrow U(\boldsymbol{\theta} \cup \boldsymbol{\theta}_{\text{new}}; \mathcal{A} \cup \mathcal{A}_{\text{new}}), \tag{6}$$

while previously added gates remain fixed. Increasing the number of iterations systematically reduces the infidelity with respect to the target state.

### A.3.2 CIRCUIT-BASED METHODS

**Variational:** Here we provide an example of the variational methods, i.e., Hardware-Efficient (HE) circuits (Nakaji et al., 2022), which are commonly used as AQL circuits. The HE circuit $U_{\mathrm{HE}}(\boldsymbol{\theta})$ consists of repeated layers of parameterized single-qubit rotations and CNOT gates acting on adjacent qubits. Each layer $i$ applies a rotation layer $R_Y$ followed by $R_Z$, and then a CNOT layer. The CNOT pattern alternates between even-odd and odd-even qubit pairs: in the $i$-th layer, CNOT gates are applied on pairs $(2n, (2n+1)\%N)$ for $i \bmod 2 = 0$ and on pairs $(2n+1, (2n+2)\%N)$ for $i \bmod 2 = 1$, where $0 \le 2n \le N-1$ or $0 \le 2n \le N-2$ as appropriate. The parameters $\boldsymbol{\theta}$ are typically updated using gradient-based optimizers to minimize the loss function

$$\ell(\boldsymbol{\theta}) = 1 - |\langle \boldsymbol{v}_{\mathrm{target}}|U_{\mathrm{HE}}(\boldsymbol{\theta})|\psi_0\rangle|^2. \tag{7}$$

Within our general framework, HE circuits represent a variational approach in which only the parameters $\boldsymbol{\theta}$ are optimized, while the circuit architecture $\mathcal{A}$ is fixed.

**Non-variational:** The Automatic Quantum Circuit Encoding (AQCE) method (Shirakawa et al., 2024) is a non-variational approach that iteratively updates two-qubit unitaries in a prescribed circuit in a forward-backward fashion. Suppose the current encoding unitary list is $\{G_1, \ldots, G_M\}$. The locally optimal update for the $m$-th two-qubit gate is obtained as follows. For a given choice of the two-qubit subsystem $\mathcal{Q}_m$, define

$$F_m = \mathrm{Tr}_{[N]\backslash\mathcal{Q}_m} \left[ G_{m+1}^\dagger \cdots G_M^\dagger |\boldsymbol{v}_{\mathrm{target}}\rangle\langle\psi_0|G_1^\dagger \cdots G_{m-1}^\dagger \right], \tag{8}$$

where $|\psi_0\rangle$ is the initial product state. Performing a singular value decomposition (SVD)

$$F_m = XDY, \tag{9}$$

the updated two-qubit unitary is

$$G_m^{\mathrm{new}} = XY. \tag{10}$$

In practice, the algorithm considers all possible choices of the two-qubit subsystem $\mathcal{Q}_m$ and selects the one that maximizes $|\mathrm{Tr}[D]|$. This choice can be understood as follows: the fidelity between the encoding state corresponding to the updated unitary and the target state is

$$
\begin{aligned}
F &= \left| \langle\psi_0|G_1^\dagger \cdots G_{m-1}^\dagger G_m^{\mathrm{new}\dagger} G_{m+1}^\dagger \cdots G_M^\dagger |\boldsymbol{v}_{\mathrm{target}}\rangle \right|^2 \\
&= \left| \mathrm{Tr}\left[ G_m^{\mathrm{new}\dagger} G_{m+1}^\dagger \cdots G_M^\dagger |\boldsymbol{v}_{\mathrm{target}}\rangle\langle\psi_0|G_1^\dagger \cdots G_{m-1}^\dagger \right] \right|^2 \\
&= \left| \mathrm{Tr}_{\mathcal{Q}_m}\left[ G_m^{\mathrm{new}\dagger} \mathrm{Tr}_{[N]\backslash\mathcal{Q}_m}\left[ G_{m+1}^\dagger \cdots G_M^\dagger |\boldsymbol{v}_{\mathrm{target}}\rangle\langle\psi_0|G_1^\dagger \cdots G_{m-1}^\dagger \right] \right] \right|^2 \\
&= \left| \mathrm{Tr}_{\mathcal{Q}_m}\left[ G_m^{\mathrm{new}\dagger} F_m \right] \right|^2 \\
&= \left| \mathrm{Tr}\left[ Y^\dagger X^\dagger XDY \right] \right|^2 \\
&= \left| \mathrm{Tr}[D] \right|^2.
\end{aligned}
$$

Thus, maximizing $|\mathrm{Tr}[D]|$ corresponds to maximizing the fidelity $F$.

After several forward-backward sweeps, new gates are added to the circuit following the same procedure by treating $G_{M+1} = I$. The final sequence of two-qubit unitaries $\{G_i\}$ is then further decomposed into hardware-friendly gates via the KAK decomposition (Tucci, 2005), resulting in 2 CNOT (or CZ) gates for real matrices and 3 gates for general complex matrices.

Within our framework, AQCE fits naturally as a non-variational method that sequentially updates both the circuit parameters $\boldsymbol{\theta}$, obtained from the KAK decomposition of the updated two-qubit unitaries, and the circuit architecture $\mathcal{A}$, determined by the qubit pairs on which the updated unitaries act.

## B PROOF OF THEOREMS

### B.1 TECHNICAL LEMMAS

Before the proof of main theorems, we provide a technical lemma, which gives the maximum fidelity to approximate a mixed single-qubit state by any single-qubit pure state.

**Lemma B.1.** *Denote by $\rho$ a single-qubit mixed state. Then*

$$\max_{|\phi\rangle} \mathrm{Tr}[|\phi\rangle\langle\phi|\rho] = \frac{1 + \sqrt{2^{1-\mathcal{S}(\rho)} - 1}}{2}, \tag{11}$$

*where $\mathcal{S}(\rho)$ denotes the Renyi-2 entropy of the state $\rho$.*

*Proof.* For convenience, we denote by $\rho = \sum_{j=1}^2 \lambda_j |\psi_j\rangle\langle\psi_j|$ the spectral decomposition of $\rho$, where $\lambda_{1,2} \in \mathbb{R}$ and $\lambda_1 + \lambda_2 = \mathrm{Tr}[\rho] = 1$ due to the density matrix property. Without loss of generality, we assume $\lambda_1 \geq \lambda_2$. Thus, the left side of Eq. (11) achieves the largest value when $|\phi\rangle = |\psi_1\rangle$, i.e.

$$\max_{|\phi\rangle} \mathrm{Tr}[|\phi\rangle\langle\phi|\rho] = \mathrm{Tr}[|\psi_1\rangle\langle\psi_1|\rho] = \lambda_1. \tag{12}$$

Additionally, the spectral decomposition could also be employed in the Renyi entropy

$$
\begin{aligned}
\mathcal{S}(\rho) &= -\log_2 \mathrm{Tr}[\rho^2] \\
&= -\log_2 \mathrm{Tr}\left[ \sum_{j=1}^2 \sum_{k=1}^2 \lambda_j \lambda_k |\psi_j\rangle\langle\psi_j|\psi_k\rangle\langle\psi_k| \right] \\
&= -\log_2 \mathrm{Tr}\left[ \sum_{j=1}^2 \lambda_j^2 |\psi_j\rangle\langle\psi_j| \right] \\
&= -\log_2 \left( \lambda_1^2 + \lambda_2^2 \right).
\end{aligned}
\tag{13}
$$

By combining Eq. (13) with $\lambda_1 + \lambda_2 = 1$, we could obtain the formulation of $\lambda_1$ with respect to the Renyi entropy $\mathcal{S}(\rho)$ as follows,

$$\lambda_1 = \frac{1 + \sqrt{2^{1-\mathcal{S}(\rho)} - 1}}{2}. \tag{14}$$

Comparing Eqs.(12) and (14), we have

$$\max_{|\phi\rangle} \mathrm{Tr}[|\phi\rangle\langle\phi|\rho] = \frac{1 + \sqrt{2^{1-\mathcal{S}(\rho)} - 1}}{2}.$$

Thus, Lemma B.1 is proved.

$\square$

**Lemma B.2.** *Denote by $\rho_{ij}$ the $(i,j)$-th element in the density matrix form of $\rho$. Then, the maximum fidelity in Lemma B.1 could be achieved by $|\phi\rangle = R_Z(\beta) R_Y(\gamma)|0\rangle$, where $\beta = \arg(\rho_{10})$ and $\gamma = \frac{\pi}{2} - \arcsin \frac{\rho_{00} - \rho_{11}}{\sqrt{4|\rho_{10}|^2 + (\rho_{00} - \rho_{11})^2}}$.*

*Proof.* First, we expand the state $|\phi\rangle$ in terms of $\beta$ and $\gamma$:

$$|\phi\rangle = e^{-\beta Z/2} e^{-\gamma Y/2} |0\rangle = \begin{bmatrix} e^{-i\frac{\beta}{2}} & 0 \\ 0 & e^{i\frac{\beta}{2}} \end{bmatrix} \begin{bmatrix} \cos\frac{\gamma}{2} & -\sin\frac{\gamma}{2} \\ \sin\frac{\gamma}{2} & \cos\frac{\gamma}{2} \end{bmatrix} \begin{bmatrix} 1 \\ 0 \end{bmatrix} = \begin{bmatrix} e^{-i\frac{\beta}{2}} \cos\frac{\gamma}{2} \\ e^{i\frac{\beta}{2}} \sin\frac{\gamma}{2} \end{bmatrix}. \tag{15}$$

Then, the fidelity between $\rho$ and $|\phi\rangle$ is

$$
\begin{aligned}
\mathrm{Tr}[|\phi\rangle\langle\phi|\rho] &= \mathrm{Tr}\left[ \begin{bmatrix} \frac{1}{2} + \frac{1}{2}\cos\gamma & \frac{1}{2}e^{-i\beta}\sin\gamma \\ \frac{1}{2}e^{i\beta}\sin\gamma & \frac{1}{2} - \frac{1}{2}\cos\gamma \end{bmatrix} \begin{bmatrix} \rho_{00} & \rho_{01} \\ \rho_{10} & \rho_{11} \end{bmatrix} \right] \\
&= \left( \frac{1}{2} + \frac{1}{2}\cos\gamma \right) \rho_{00} + \frac{1}{2} e^{-i\beta} \sin\gamma \rho_{10} + \frac{1}{2} e^{i\beta} \sin\gamma \rho_{01} + \left( \frac{1}{2} - \frac{1}{2}\cos\gamma \right) \rho_{11} \\
&= \frac{1}{2} + \frac{\rho_{00} - \rho_{11}}{2} \cos\gamma + \mathrm{Re}\left[ e^{-i\beta} \rho_{10} \right] \sin\gamma,
\end{aligned}
\tag{16}
$$

where Eq. (16) follows from density matrix properties $\mathrm{Tr}[\rho] = 1$ and $\rho = \rho^\dagger$. Subsequently, by considering the formulation of $\beta$ and $\gamma$ with respect to $\rho$, we have

$$\mathrm{Tr}[|\phi\rangle\langle\phi|\rho] = \frac{1}{2} + \frac{\rho_{00} - \rho_{11}}{2}\cos\gamma + |\rho_{10}|\sin\gamma$$

$$= \frac{1}{2} + \sqrt{\left(\frac{\rho_{00} - \rho_{11}}{2}\right)^2 + |\rho_{10}|^2}. \tag{17}$$

Moreover, the Renyi entropy $\mathcal{S}(\rho)$ could be formulated in terms of $\rho$ as follows

$$\mathcal{S}(\rho) = -\log_2 \mathrm{Tr}[\rho^2]$$

$$= -\log_2\left(\rho_{00}^2 + \rho_{11}^2 + 2|\rho_{10}|^2\right).$$

Therefore, we have

$$2^{1-S(\rho)} - 1 = 2\left(\rho_{00}^2 + \rho_{11}^2 + 2|\rho_{10}|^2\right) - 1$$

$$= 2\rho_{00}^2 + 2\rho_{11}^2 - (\rho_{00} + \rho_{11})^2 + 4|\rho_{10}|^2$$

$$= (\rho_{00} - \rho_{11})^2 + 4|\rho_{10}|^2. \tag{18}$$

Thus, the Lemma B.2 is proved by combining Eqs. (17) and (18) with the maximum fidelity value in Lemma B.1.

$\square$

**Lemma B.3.** *Let $\mathcal{N}$ be an $N$-qubit CPTP map and let $\mathcal{U}_\ell(\rho) := U_\ell \rho U_\ell^\dagger$ denote the unitary channel associated with an $N$-qubit unitary $U_\ell$ for each $\ell \in [L]$. Given an initial state $\rho_0$, define the ideal and noisy output states by*

$$\rho := \left(\mathcal{U}_L \circ \cdots \circ \mathcal{U}_1\right)(\rho_0), \tag{19}$$

$$\hat{\rho} := \left(\mathcal{N} \circ \mathcal{U}_L \circ \mathcal{N} \circ \cdots \circ \mathcal{N} \circ \mathcal{U}_1 \circ \mathcal{N}\right)(\rho_0), \tag{20}$$

*where $\hat{\rho}$ is obtained from $\rho_0$ by inserting $L+1$ layers of noise $\mathcal{N}$ alternating with the $L$ unitaries. Then,*

$$\|\hat{\rho} - \rho\|_1 \leq (L+1)\|\mathcal{N} - \mathrm{id}\|_\diamond, \tag{21}$$

*where* id *denotes the identity channel and $\|\cdot\|_\diamond$ is the diamond norm.*

*Proof.* For convenience, we denote the noiseless channel by

$$\mathcal{E}_0^{(L)} := \mathcal{U}_L \circ \cdots \circ \mathcal{U}_1 \tag{22}$$

and the fully noisy channel by

$$\mathcal{E}_\mathcal{N}^{(L)} := \mathcal{N} \circ \mathcal{U}_L \circ \mathcal{N} \circ \cdots \circ \mathcal{N} \circ \mathcal{U}_1 \circ \mathcal{N}. \tag{23}$$

By definition,

$$\rho = \mathcal{E}_0^{(L)}(\rho_0), \qquad \hat{\rho} = \mathcal{E}_\mathcal{N}^{(L)}(\rho_0). \tag{24}$$

Thus,

$$\|\hat{\rho} - \rho\|_1 = \left\|\mathcal{E}_\mathcal{N}^{(L)}(\rho_0) - \mathcal{E}_0^{(L)}(\rho_0)\right\|_1 \leq \left\|\mathcal{E}_\mathcal{N}^{(L)} - \mathcal{E}_0^{(L)}\right\|_\diamond, \tag{25}$$

where the inequality follows from the definition of the diamond norm, which upper bounds the trace-norm difference of the outputs on any input state.

In the following, we derive an upper bound on $\left\|\mathcal{E}_\mathcal{N}^{(L)} - \mathcal{E}_0^{(L)}\right\|_\diamond$. There are $L+1$ noise positions in the noisy channel $\mathcal{E}_\mathcal{N}^{(L)}$: one before $U_1$, one between each consecutive pair $U_j$ and $U_{j+1}$, and one after $U_L$. We construct a sequence of intermediate channels $\{\Phi^{(k)}\}_{k=0}^{L+1}$ such that

$$\Phi^{(0)} = \mathcal{E}_0^{(L)}, \qquad \Phi^{(L+1)} = \mathcal{E}_\mathcal{N}^{(L)}, \tag{26}$$

and for each $k \in \{1, \ldots, L\}$, the channel $\Phi^{(k)}$ is obtained by applying the first $k$ noise layers. Accordingly, we have the telescoping decomposition

$$\mathcal{E}_\mathcal{N}^{(L)} - \mathcal{E}_0^{(L)} = \Phi^{(L+1)} - \Phi^{(0)} = \sum_{k=1}^{L+1}\left(\Phi^{(k)} - \Phi^{(k-1)}\right). \tag{27}$$

Each $\Phi^{(k)} - \Phi^{(k-1)}$ corresponds to replacing a single identity channel by $\mathcal{N}$, while all other positions remain unchanged. Therefore, there exist CPTP maps $\Lambda_L^{(k)}$ and $\Lambda_R^{(k)}$ (given by compositions of unitary channels $\mathcal{U}_\ell$ and noise/identity channels) such that

$$\Phi^{(k)} - \Phi^{(k-1)} = \Lambda_R^{(k)} \circ (\mathcal{N} - \text{id}) \circ \Lambda_L^{(k)}. \tag{28}$$

Since both $\Lambda_L^{(k)}$ and $\Lambda_R^{(k)}$ are CPTP maps, their diamond norms satisfy

$$\|\Lambda_L^{(k)}\|_\diamond = \|\Lambda_R^{(k)}\|_\diamond = 1. \tag{29}$$

Thus, we have

$$\left\|\Phi^{(k)} - \Phi^{(k-1)}\right\|_\diamond \le \|\Lambda_R^{(k)}\|_\diamond \|\mathcal{N} - \text{id}\|_\diamond \|\Lambda_L^{(k)}\|_\diamond = \|\mathcal{N} - \text{id}\|_\diamond. \tag{30}$$

Using the triangle inequality and summing over all $(L+1)$ noise positions, we obtain

$$\left\|\mathcal{E}_\mathcal{N}^{(L)} - \mathcal{E}_0^{(L)}\right\|_\diamond \le \sum_{k=1}^{L+1} \left\|\Phi^{(k)} - \Phi^{(k-1)}\right\|_\diamond$$
$$\le (L+1)\,\|\mathcal{N} - \text{id}\|_\diamond. \tag{31}$$

Combining Eqs. (25) and (31), we obtain

$$\|\hat{\rho} - \rho\|_1 \le \left\|\mathcal{E}_\mathcal{N}^{(L)} - \mathcal{E}_0^{(L)}\right\|_\diamond \le (L+1)\,\|\mathcal{N} - \text{id}\|_\diamond. \tag{32}$$

Thus, we have proved Lemma B.3. $\square$

## B.2 PROOF OF MAIN THEOREM

Here we present the full versions of the theorem in the main text along with their proofs.

**Theorem B.4.** *Denote the entanglement measure for a $N$-qubit state $|\psi\rangle$ as $\mathcal{S}(|\psi\rangle) = \sum_{i=1}^N \mathcal{S}_{\{i\}}(|\psi\rangle)$. Then for the state $|\boldsymbol{v}_{\text{target}}\rangle$ and the circuit $U$ with $\mathcal{S}(U^\dagger|\boldsymbol{v}_{\text{target}}\rangle) = S$, the infidelity between $|\boldsymbol{v}_{\text{target}}\rangle$ and the state generated from $U$ on any product state $|\psi_{\text{product}}\rangle$ is lower bounded as*

$$1 - |\langle \boldsymbol{v}_{\text{target}}|U|\psi_{\text{product}}\rangle|^2 \ge f_1(S) := \frac{1 - \sqrt{2^{1 - \frac{1}{N}S} - 1}}{2}. \tag{33}$$

*Moreover, given access to $U^\dagger|\boldsymbol{v}_{\text{target}}\rangle$, we can construct a product state $|\psi'_{\text{product}}\rangle$, such that the infidelity is upper bounded as*

$$1 - \left|\langle \boldsymbol{v}_{\text{target}}|U|\psi'_{\text{product}}\rangle\right|^2 \le f_2(S) := \frac{1}{2}\left(1 - \sqrt{2^{1 - S + \lfloor S \rfloor} - 1} + \lfloor S \rfloor\right), \tag{34}$$

*where $\lfloor \cdot \rfloor$ denotes the floor function. We remark that $f_1(S) = \frac{\ln 2}{2N}S + \mathcal{O}(S^3)$ and $f_2(S) = \frac{\ln 2}{2}S + \mathcal{O}(S^3)$ by calculating the Taylor expansion.*

*Proof.* For convenience, we denote by $\rho$ the density matrix formulation of the state $U^\dagger|\boldsymbol{v}_{\text{target}}\rangle$. We denote by $\rho_n := \text{Tr}_{[N]/\{n\}}[\rho]$ the density matrix of the $n$-th single-qubit subsystem of $\rho$ and denote the Renyi entropy $S_n := \mathcal{S}(\rho_n)$ for simplicity. Since $\rho_n$ is a single-qubit density matrix, $S_n \in [0, 1]$. We construct a general set of single-qubit projectors $\{|\psi_n\rangle\}_{n=1}^N$ with $|\psi\rangle = \otimes_{n=1}^N|\psi_n\rangle$. Accordingly, we define a set of random variables $\{y_n\}_{n=1}^N$, where $y_n = 1$ if the measurement on the $n$-th qubit yields $|\psi_n\rangle$ and $y_n = 0$ otherwise. Thus, we can reformulate the infidelity between $|\boldsymbol{v}_{\text{target}}\rangle$ and $U|\psi\rangle$ as

$$1 - |\langle \boldsymbol{v}_{\text{target}}|U|\psi\rangle|^2 = 1 - \text{Tr}\left[\rho\left(\otimes_{n=1}^N|\psi_n\rangle\langle\psi_n|\right)\right] = 1 - \Pr\left(y_n = 1,\ \forall n \in [N]\right). \tag{35}$$

First, we derive the lower bound on the infidelity in Eq. (35) as follows. We have

$$1 - \Pr\left(y_n = 1,\ \forall n \in [N]\right) \ge 1 - \min_{n \in [N]} \Pr\left(y_n = 1\right)$$
$$= 1 - \min_{n \in [N]} \text{Tr}\left[\rho\left(I^{\otimes(n-1)} \otimes |\psi_n\rangle\langle\psi_n| \otimes I^{\otimes(N-n)}\right)\right]$$

$$= 1 - \min_{n \in [N]} \mathrm{Tr}\left[\rho_n |\psi_n\rangle\langle\psi_n|\right]$$

$$\geq 1 - \min_{n \in [N]} \frac{1 + \sqrt{2^{1-S_n} - 1}}{2}, \tag{36}$$

where the last inequality follows from Lemma B.1. We further proceed to deal with the terms in Eq. 36. In particular, the function $f(x) = \sqrt{2^{1-x} - 1}$ decreases for $x \in [0, 1]$. Thus,

$$\min_{n \in [N]} \sqrt{2^{1-S_n} - 1} = \min_{n \in [N]} f(S_n) \leq f\left(\frac{1}{N}\sum_{n=1}^{N} S_n\right) = f\left(\frac{S}{N}\right) = \sqrt{2^{1-\frac{S}{N}} - 1}. \tag{37}$$

Combining Eqs. (36) and (37), we obtain

$$1 - |\langle \boldsymbol{v}_{\mathrm{target}}|U|\psi\rangle|^2 \geq \frac{1 - \sqrt{2^{1-\frac{S}{N}} - 1}}{2}. \tag{38}$$

For the case of $S \to 0$, Eq. (38) tends to $\frac{\ln 2}{2N}S + \mathcal{O}(S^3)$ by calculating the Taylor expansion.

Next, we derive the upper bound of the infidelity. We consider a specific choice of projectors $\{|\phi_n\rangle\}_{n=1}^{N}$, where each $|\phi_n\rangle$ is the pure state approximation of $\rho_{\{n\}}$ with the largest fidelity given by Lemmas B.1 and B.2. Similarly, we define a set of random variables $\{x_n\}_{n=1}^{N}$, where $x_n = 1$ if measuring the $n$-th qubit of $\rho$ yields the state $|\phi_n\rangle$ and $x_n = 0$ otherwise. Based on Lemma B.1, we have

$$\Pr(x_n = 1) = \frac{1 + \sqrt{2^{1-S_n} - 1}}{2}, \ \Pr(x_n = 0) = \frac{1 - \sqrt{2^{1-S_n} - 1}}{2}. \tag{39}$$

Thus, the infidelity between states $|\boldsymbol{v}_{\mathrm{target}}\rangle$ and $U|\phi\rangle = U(\otimes_{n=1}^{N}|\phi_n\rangle)$ can be reformulated as

$$1 - |\langle \boldsymbol{v}_{\mathrm{target}}|U|\phi\rangle|^2 = 1 - \mathrm{Tr}\left[\rho\left(\otimes_{n=1}^{N}|\phi_n\rangle\langle\phi_n|\right)\right] = 1 - \Pr(x_n = 1, \ \forall n \in [N]). \tag{40}$$

In the following, we proceed with the derivation from the probability in Eq. (40) to obtain the upper bound of the infidelity. We have

$$\Pr(x_n = 1, \ \forall n \in [N]) = 1 - \Pr(\exists\, n \in [N], \ \mathrm{s.t.}\ x_n = 0)$$

$$\geq 1 - \sum_{n=1}^{N} \Pr(x_n = 0)$$

$$= 1 - \sum_{n=1}^{N} \frac{1 - \sqrt{2^{1-S_n} - 1}}{2}, \tag{41}$$

where Eq. (41) yields from Eq. (39). The function $f(x) = \sqrt{2^{1-x} - 1}$ is concave for $x \in [0, 1]$ since it has the second-order derivative $f''(x) = \frac{(\ln 2)^2\, 2^{1-x}(2^{1-x} - 2)}{4(2^{1-x} - 1)^{3/2}} \leq 0$ for $x \in [0, 1]$. Therefore, we have

$$f(a) + f(b) \geq \begin{cases} f(0) + f(a + b) & \text{when } a, b, a + b \in [0, 1], \\ f(a + b - 1) + f(1) & \text{when } a, b \in [0, 1] \text{ and } a + b \in [1, 2]. \end{cases} \tag{42}$$

due to the property of concave functions. Eq. (42) can be employed to Eq. (41) by selecting qubit pair $(i, j)$ such that $S_i, S_j \notin \{0, 1\}$ and pushing the new value $S_i', S_j'$ towards the boundary of $[0, 1]$. By conducting the above procedure for at most $N - 1$ times, we could bound the probability as follows

$$\Pr(x_n = 1, \ \forall n \in [N]) \geq 1 - \frac{N}{2} + \frac{1}{2}\left(\lfloor S_{\boldsymbol{\theta}}\rfloor f(1) + f(S - \lfloor S\rfloor) + (N - \lfloor S\rfloor - 1)f(0)\right)$$

$$= \frac{1}{2}\left(\sqrt{2^{1-S+\lfloor S\rfloor} - 1} - \lfloor S\rfloor + 1\right), \tag{43}$$

where Eq. (43) yields from calculating the value of function $f$. Thus, we obtain the upper bound for the infidelity:

$$1 - |\langle \boldsymbol{v}_{\mathrm{target}}|U|\phi\rangle|^2 = 1 - \Pr(x_n = 1, \ \forall n \in [N]) \leq \frac{1}{2}\left(1 - \sqrt{2^{1-S+\lfloor S\rfloor} - 1} + \lfloor S\rfloor\right). \tag{44}$$

For the case of $S \to 0$, Eq. (44) tends to $\frac{\ln 2}{2}S + \mathcal{O}(S^3)$ by calculating the Taylor expansion. Thus, we have proved Theorem B.4. $\qquad\square$

## C  THEORETICAL RESULTS OF NOISY CASES

While AQL is designed for both noiseless and noisy regimes, its behavior in the noisy case is particularly critical for near-term applications. In practical NISQ settings, hardware noise perturbs the circuit evolution, and its impact on the error beyond the noiseless bounds needs to be analyzed. Here, we extend the results in Theorem B.4 into the general CPTP channel case.

**Theorem C.1.** *Denote the entanglement measure for an $N$-qubit (pure or mixed) state $\rho$ as $\mathcal{S}(\rho) = \sum_{n=1}^{N} \mathcal{S}_{\{n\}}(\rho)$. Let $\mathcal{S}(\mathcal{E}(\rho_{\mathrm{target}})) = S$, where $\mathcal{E}$ is a CPTP map. Then, for any product state $|\psi_{\mathrm{product}}\rangle$, the infidelity between $\mathcal{E}(\rho_{\mathrm{target}})$ and $|\psi_{\mathrm{product}}\rangle$ is lower bounded as*

$$1 - \langle\psi_{\mathrm{product}}|\mathcal{E}(\rho_{\mathrm{target}})|\psi_{\mathrm{product}}\rangle \geq f_1(S) := \frac{1 - \sqrt{2^{1-\frac{1}{N}S} - 1}}{2}. \tag{45}$$

*Moreover, given access to $\mathcal{E}(\rho_{\mathrm{target}})$, we can construct a product state $|\psi'_{\mathrm{product}}\rangle$, such that the infidelity is upper bounded as*

$$1 - \langle\psi'_{\mathrm{product}}|\mathcal{E}(\rho_{\mathrm{target}})|\psi'_{\mathrm{product}}\rangle \leq f_2(S) := \frac{1}{2}\left(1 - \sqrt{2^{1-S+\lfloor S\rfloor} - 1} + \lfloor S\rfloor\right), \tag{46}$$

*where $\lfloor\cdot\rfloor$ denotes the floor function. We remark that $f_1(S) = \frac{\ln 2}{2N}S + \mathcal{O}(S^3)$ and $f_2(S) = \frac{\ln 2}{2}S + \mathcal{O}(S^3)$ by calculating the Taylor expansion.*

Next, we consider a layered noisy circuit model and derive noise-dependent bounds, as stated in Theorem C.2.

**Theorem C.2.** *Denote the entanglement measure for an $N$-qubit state $\rho$ as $\mathcal{S}(\rho) = \sum_{n=1}^{N} \mathcal{S}_{\{n\}}(\rho)$. Let $|v_{\mathrm{target}}\rangle$ be an $N$-qubit pure target state and set $\rho_{\mathrm{target}} = |v_{\mathrm{target}}\rangle\langle v_{\mathrm{target}}|$. Let $\mathcal{M}, \mathcal{N}$ be two CPTP noise channels and $\mathcal{U}_\ell(\rho) := U_\ell\rho U_\ell^\dagger$ for unitary $U_\ell$. Consider two $L$-layer noisy circuits $\mathcal{E}^{(L)} := \mathcal{M}\circ\mathcal{U}_L\circ\mathcal{M}\circ\cdots\circ\mathcal{M}\circ\mathcal{U}_1\circ\mathcal{M}$ and $\mathcal{F}^{(L)} := \mathcal{N}\circ\mathcal{U}_1^\dagger\circ\mathcal{N}\circ\cdots\circ\mathcal{N}\circ\mathcal{U}_L^\dagger\circ\mathcal{N}$. Suppose $\mathcal{S}(\mathcal{F}^{(L)}(\rho_{\mathrm{target}})) = S$. Then for any product state $|\psi_{\mathrm{product}}\rangle$, the infidelity between the target $\rho_{\mathrm{target}}$ and the state $\mathcal{E}^{(L)}(|\psi_{\mathrm{product}}\rangle\langle\psi_{\mathrm{product}}|)$ is lower bounded as*

$$1 - \mathrm{Tr}[\rho_{\mathrm{target}}\mathcal{E}^{(L)}(|\psi_{\mathrm{product}}\rangle\langle\psi_{\mathrm{product}}|)] \geq f_1(S) - (L+1)\left(\|\mathcal{M} - \mathrm{id}\|_\diamond + \|\mathcal{N} - \mathrm{id}\|_\diamond\right). \tag{47}$$

*Moreover, given access to $\mathcal{F}^{(L)}(\rho_{\mathrm{target}})$, we can construct a product state $|\psi'_{\mathrm{product}}\rangle$, such that the infidelity is upper bounded as*

$$1 - \mathrm{Tr}[\rho_{\mathrm{target}}\mathcal{E}^{(L)}(|\psi'_{\mathrm{product}}\rangle\langle\psi'_{\mathrm{product}}|)] \leq f_2(S) + (L+1)\left(\|\mathcal{M} - \mathrm{id}\|_\diamond + \|\mathcal{N} - \mathrm{id}\|_\diamond\right). \tag{48}$$

*Functions $f_1$ and $f_2$ follow the definitions in Theorem C.1.*

The above results show that the entanglement-governed terms in Theorem B.4 persist in the noisy setting, with an additional noise term that scales linearly with the depth $L$ and the noise strength. More precisely, the noise strength is captured by the accumulated quantity $(L+1)\left(\|\mathcal{M}-\mathrm{id}\|_\diamond+\|\mathcal{N}-\mathrm{id}\|_\diamond\right)$. For concrete CPTP noise models, this correction is easy to interpret. For example, a depolarizing channel $\mathcal{D}_p(\rho) = (1-p)\rho + pI/d$ with error rate $p$ satisfies $\|\mathcal{D}_p - \mathrm{id}\|_\diamond = \mathcal{O}(p)$, leading to a correction of order $(L+1)p$. Similar linear scalings hold for other common noise models such as dephasing and amplitude-damping channels. When both the entanglement measure and the accumulated noise remain moderate, AQL can achieve a small approximation error on NISQ hardware.

We remark that for noisy quantum channels, the entanglement measure $\mathcal{S}$ is generally higher than that in the corresponding noiseless case. Here we provide an example of the depolarizing channel in Theorem C.3. Therefore, due to the exactly same bounds formulations in Theorems B.4 and C.1, the infidelity achieved by noisy circuits would be worse than that of noiseless circuits.

**Theorem C.3.** *Denote the entanglement measure for an $N$-qubit (pure or mixed) state $\rho$ as $\mathcal{S}(\rho) = \sum_{n=1}^{N} \mathcal{S}_{\{n\}}(\rho)$. Let $\mathcal{D}_p\rho = (1-p)\rho + pI/d$ be the depolarizing channel with error rate $p$. Then*

$$\left(1 - \frac{p}{\ln 4}\right)\mathcal{S}(\rho) + \frac{Np}{\ln 4} \leq \mathcal{S}(\mathcal{D}_p(\rho)) \leq \mathcal{S}(\rho) + N\log_2\frac{2}{1+(1-p)^2}. \tag{49}$$

### C.1 PROOF OF THEOREM C.1.

*Proof.* For convenience, we denote by $\rho$ the $N$-qubit state $\mathcal{E}(\rho_{\text{target}})$ and by $\rho_n := \text{Tr}_{[N]/\{n\}}[\rho]$ the density matrix of the $n$-th single-qubit subsystem of $\rho$. We denote the Renyi entropy $S_n := \mathcal{S}(\rho_n)$ for simplicity. Since $\rho_n$ is a single-qubit density matrix, $S_n \in [0,1]$. We construct a general set of single-qubit projectors $\{|\psi_n\rangle\}_{n=1}^N$ with $|\psi_{\text{product}}\rangle = \otimes_{n=1}^N |\psi_n\rangle$. Accordingly, we define a set of random variables $\{y_n\}_{n=1}^N$, where $y_n = 1$ if the measurement on the $n$-th qubit yields $|\psi_n\rangle$ and $y_n = 0$ otherwise. Thus, we can reformulate the infidelity between $\rho$ and $|\psi_{\text{product}}\rangle$ as

$$1 - \langle\psi_{\text{product}}|\rho|\psi_{\text{product}}\rangle = 1 - \text{Tr}\left[\rho\left(\otimes_{n=1}^N |\psi_n\rangle\langle\psi_n|\right)\right]$$
$$= 1 - \text{Pr}(y_n = 1, \forall n \in [N]). \tag{50}$$

First, we derive the lower bound on the infidelity in Eq. (50) as follows. We have

$$1 - \text{Pr}(y_n = 1, \forall n \in [N]) \geq 1 - \min_{n \in [N]} \text{Pr}(y_n = 1)$$
$$= 1 - \min_{n \in [N]} \text{Tr}\left[\rho\left(I^{\otimes(n-1)} \otimes |\psi_n\rangle\langle\psi_n| \otimes I^{\otimes(N-n)}\right)\right]$$
$$= 1 - \min_{n \in [N]} \text{Tr}\left[\rho_n|\psi_n\rangle\langle\psi_n|\right]$$
$$\geq 1 - \min_{n \in [N]} \frac{1 + \sqrt{2^{1-S_n} - 1}}{2}, \tag{51}$$

where the last inequality follows from Lemma B.1, which gives the maximal fidelity between a single-qubit density matrix and a pure state in terms of its Rényi entropy.

We further proceed to deal with the terms in Eq. (51). In particular, the function $f(x) = \sqrt{2^{1-x} - 1}$ decreases for $x \in [0,1]$. Thus,

$$\min_{n \in [N]} \sqrt{2^{1-S_n} - 1} = \min_{n \in [N]} f(S_n) \leq f\left(\frac{1}{N}\sum_{n=1}^N S_n\right) = f\left(\frac{S}{N}\right) = \sqrt{2^{1-\frac{S}{N}} - 1}. \tag{52}$$

Combining Eqs. (51) and (52), we obtain

$$1 - \langle\psi_{\text{product}}|\rho|\psi_{\text{product}}\rangle \geq \frac{1 - \sqrt{2^{1-\frac{S}{N}} - 1}}{2} =: f_1(S). \tag{53}$$

For the case of $S \to 0$, Eq. (53) tends to $\frac{\ln 2}{2N}S + \mathcal{O}(S^3)$ by calculating the Taylor expansion.

Next, we derive the upper bound of the infidelity. We consider a specific choice of projectors $\{|\phi_n\rangle\}_{n=1}^N$, where each $|\phi_n\rangle$ is the pure-state approximation of $\rho_{\{n\}}$ with the largest fidelity given by Lemmas B.1 and B.2. That is, $|\phi_n\rangle$ is chosen to maximize $\text{Tr}[\rho_n|\phi_n\rangle\langle\phi_n|]$. Similarly, we define a set of random variables $\{x_n\}_{n=1}^N$, where $x_n = 1$ if measuring the $n$-th qubit of $\rho$ yields the state $|\phi_n\rangle$ and $x_n = 0$ otherwise. Based on Lemma B.1, we have

$$\text{Pr}(x_n = 1) = \frac{1 + \sqrt{2^{1-S_n} - 1}}{2}, \quad \text{Pr}(x_n = 0) = \frac{1 - \sqrt{2^{1-S_n} - 1}}{2}. \tag{54}$$

Thus, the infidelity between $\rho$ and the product state $|\phi\rangle = \otimes_{n=1}^N |\phi_n\rangle$ can be reformulated as

$$1 - \langle\phi|\rho|\phi\rangle = 1 - \text{Tr}\left[\rho\left(\otimes_{n=1}^N |\phi_n\rangle\langle\phi_n|\right)\right]$$
$$= 1 - \text{Pr}(x_n = 1, \forall n \in [N]). \tag{55}$$

In the following, we proceed with the derivation from the probability in Eq. (55) to obtain the upper bound of the infidelity. We have

$$\text{Pr}(x_n = 1, \forall n \in [N]) = 1 - \text{Pr}(\exists n \in [N], \text{ s.t. } x_n = 0)$$
$$\geq 1 - \sum_{n=1}^N \text{Pr}(x_n = 0)$$
$$= 1 - \sum_{n=1}^N \frac{1 - \sqrt{2^{1-S_n} - 1}}{2}, \tag{56}$$

where Eq. (56) follows from Eq. (54) and the union bound.

The function $f(x) = \sqrt{2^{1-x} - 1}$ is concave for $x \in [0, 1]$ since it has the second-order derivative

$$f''(x) = \frac{(\ln 2)^2 \, 2^{1-x} \, (2^{1-x} - 2)}{4 \, (2^{1-x} - 1)^{3/2}} \leq 0 \tag{57}$$

for $x \in [0, 1]$. Therefore, we have

$$f(a) + f(b) \geq \begin{cases} f(0) + f(a+b) & \text{when } a, b, a+b \in [0, 1], \\ f(a+b-1) + f(1) & \text{when } a, b \in [0, 1] \text{ and } a + b \in [1, 2], \end{cases} \tag{58}$$

due to the property of concave functions. Eq. (58) can be employed to Eq. (56) by selecting a qubit pair $(i, j)$ such that $S_i, S_j \notin \{0, 1\}$ and pushing the new values $S_i', S_j'$ towards the boundary of $[0, 1]$ while keeping $S_i' + S_j' = S_i + S_j$. By conducting the above procedure for at most $N - 1$ times, we can bound the probability as follows:

$$\Pr(x_n = 1, \ \forall n \in [N]) \geq 1 - \frac{N}{2} + \frac{1}{2}\left( \lfloor S \rfloor f(1) + f(S - \lfloor S \rfloor) + (N - \lfloor S \rfloor - 1) \, f(0) \right)$$

$$= \frac{1}{2}\left( \sqrt{2^{1-S+\lfloor S \rfloor} - 1} - \lfloor S \rfloor + 1 \right), \tag{59}$$

where Eq. (59) yields from calculating the value of the function $f$ at 0 and 1. Thus, we obtain the upper bound for the infidelity:

$$1 - \langle \phi | \rho | \phi \rangle = 1 - \Pr(x_n = 1, \ \forall n \in [N])$$

$$\leq \frac{1}{2}\left( 1 - \sqrt{2^{1-S+\lfloor S \rfloor} - 1} + \lfloor S \rfloor \right) =: f_2(S). \tag{60}$$

For the case of $S \to 0$, Eq. (60) tends to $\frac{\ln 2}{2} S + \mathcal{O}(S^3)$ by calculating the Taylor expansion. Thus, we have proved Theorem C.1. $\qquad\square$

## C.2 PROOF OF THEOREM C.2.

*Proof.* For convenience, we denote by $\hat{\rho} := \mathcal{F}^{(L)}(\rho_{\text{target}})$ and $\rho := \mathcal{U}_1^\dagger \circ \cdots \circ \mathcal{U}_L^\dagger(\rho_{\text{target}})$ the state obtained from $\rho_{\text{target}}$ with and without noise channels, respectively. Next, we focus on the lower bound in Eq. (47), while the upper bound in Eq. (48) can be derived similarly. By employing Theorem C.1, we have

$$1 - \langle \psi_{\text{product}} | \hat{\rho} | \psi_{\text{product}} \rangle = 1 - \text{Tr}[\hat{\rho} \sigma_{\text{product}}] \geq f_1(S) \tag{61}$$

for all product state $|\psi_{\text{product}}\rangle$, where $\sigma_{\text{product}} := |\psi_{\text{product}}\rangle\langle\psi_{\text{product}}|$. Thus, the infidelity between the target state $\rho_{\text{target}}$ and the state recovered by $\mathcal{E}^{(L)}$ is

$$1 - \text{Tr}[\rho_{\text{target}} \mathcal{E}^{(L)}(\sigma_{\text{product}})]$$

$$= 1 - \text{Tr}[\hat{\rho} \sigma_{\text{product}}] + \text{Tr}[\hat{\rho} \sigma_{\text{product}}] - \text{Tr}[\rho \sigma_{\text{product}}] + \text{Tr}[\rho \sigma_{\text{product}}] - \text{Tr}[\rho_{\text{target}} \mathcal{E}^{(L)}(\sigma_{\text{product}})]$$

$$\geq f_1(S) - \left| \text{Tr}[\hat{\rho} \sigma_{\text{product}}] - \text{Tr}[\rho \sigma_{\text{product}}] \right| - \left| \text{Tr}[\rho \sigma_{\text{product}}] - \text{Tr}[\rho_{\text{target}} \mathcal{E}^{(L)}(\sigma_{\text{product}})] \right|. \tag{62}$$

The second term in Eq. (62) can be bounded as

$$\left| \text{Tr}[\hat{\rho} \sigma_{\text{product}}] - \text{Tr}[\rho \sigma_{\text{product}}] \right| \leq \|\hat{\rho} - \rho\|_1 \|\sigma_{\text{product}}\|_\infty \leq (L+1)\|\mathcal{N} - \text{id}\|_\diamond, \tag{63}$$

where the last inequality yields from Lemma B.3 and the operator norm $\|\sigma\|_\infty \leq 1$.

The third term in Eq. (62) is bounded in the similar way

$$\left| \text{Tr}[\rho \sigma_{\text{product}}] - \text{Tr}[\rho_{\text{target}} \mathcal{E}^{(L)}(\sigma_{\text{product}})] \right|$$

$$= \left| \text{Tr}[\mathcal{U}_1^\dagger \circ \cdots \circ \mathcal{U}_L^\dagger(\rho_{\text{target}}) \sigma_{\text{product}}] - \text{Tr}[\rho_{\text{target}} \mathcal{E}^{(L)}(\sigma_{\text{product}})] \right|$$

$$= \left| \text{Tr}[\rho_{\text{target}} \mathcal{U}_L \circ \cdots \circ \mathcal{U}_1(\sigma_{\text{product}})] - \text{Tr}[\rho_{\text{target}} \mathcal{E}^{(L)}(\sigma_{\text{product}})] \right|$$

$$\leq \left\| \mathcal{U}_L \circ \cdots \circ \mathcal{U}_1(\sigma_{\text{product}}) - \mathcal{E}^{(L)}(\sigma_{\text{product}}) \right\|_1 \|\rho_{\text{target}}\|_\infty$$

$$\leq (L+1)\|\mathcal{M} - \text{id}\|_\diamond. \tag{64}$$

Combining Eqs. (63) and (64) with Eq. (62), we obtain

$$1 - \text{Tr}[\rho_{\text{target}} \mathcal{E}^{(L)}(\sigma_{\text{product}})] \geq f_1(S) - (L+1)\left( \|\mathcal{M} - \text{id}\|_\diamond + \|\mathcal{N} - \text{id}\|_\diamond \right),$$

which is Eq. (47). Eq. (48) can be derived similarly. Thus, we have proved Theorem C.2.

$\qquad\qquad\qquad\qquad\qquad\qquad\qquad\qquad\qquad\qquad\qquad\qquad\qquad\qquad\qquad\square$

## C.3 PROOF OF THEOREM C.3

*Proof.* For convenience, we denote by $\rho_n$ and $\sigma_n$ the reduced density matrix of $\rho$ and $\sigma = \mathcal{D}_p(\rho)$ at the $n$-th qubit, respectively. Then, we have

$$\sigma_n = \text{Tr}_{[N]/\{n\}}\left[\mathcal{D}_p(\rho)\right] = \text{Tr}_{[N]/\{n\}}\left[(1-p)\rho + pI/2^N\right] = (1-p)\rho_n + \frac{p}{2}I. \tag{65}$$

Thus, the Renyi entropy of the state $\sigma_n$ can be obtained as

$$\begin{aligned}
\mathcal{S}(\sigma_n) &= -\log_2 \text{Tr}[\sigma_n^2] = -\log_2 \text{Tr}\left[((1-p)\rho_n + \frac{p}{2}I)^2\right] \\
&= -\log_2\left\{(1-p)^2\text{Tr}[\rho_n^2] + p(1-p)\text{Tr}[\rho_n] + \frac{p^2}{4}\text{Tr}[I]\right\} \\
&= -\log_2\left\{(1-p)^2\text{Tr}[\rho_n^2] + p\left(1 - \frac{p}{2}\right)\right\}.
\end{aligned} \tag{66}$$

Next, we derive inequalities for $\mathcal{S}(\sigma_n)$ based on Eq. (66). We have

$$\begin{aligned}
\mathcal{S}(\sigma_n) &= -\log_2\left\{(1-p)^2\text{Tr}[\rho_n^2] + p\left(1 - \frac{p}{2}\right)\right\} \\
&\leq -\log_2\left\{(1-p)^2\text{Tr}[\rho_n^2] + p\left(1 - \frac{p}{2}\right)\text{Tr}[\rho_n^2]\right\} \\
&= \mathcal{S}(\rho_n) - \log_2\left(1 - p + \frac{1}{2}p^2\right) \\
&= \mathcal{S}(\rho_n) + \log_2\frac{2}{1 + (1-p)^2},
\end{aligned} \tag{67}$$

where the inequality follows from $\text{Tr}[\rho_n^2] \leq 1$.

On the other side,

$$\begin{aligned}
\mathcal{S}(\sigma_n) &= -\log_2\left\{(1-p)^2\text{Tr}[\rho_n^2] + p\left(1 - \frac{p}{2}\right)\right\} \\
&= -\log_2\text{Tr}[\rho_n^2] - \log_2\left\{(1-p)^2 + \frac{p\left(1 - \frac{p}{2}\right)}{\text{Tr}[\rho_n^2]}\right\} \\
&= \mathcal{S}(\rho_n) - \log_2\left\{1 - p\left(1 - \frac{p}{2}\right)\left(2 - \frac{1}{\text{Tr}[\rho_n^2]}\right)\right\} \\
&\geq \mathcal{S}(\rho_n) + \frac{1}{\ln 2}p\left(1 - \frac{p}{2}\right)\left(2 - \frac{1}{\text{Tr}[\rho_n^2]}\right) \\
&\geq \mathcal{S}(\rho_n) + \frac{p}{\ln 4}\left(2 - \frac{1}{\text{Tr}[\rho_n^2]}\right) \\
&= \mathcal{S}(\rho_n) + \frac{p}{\ln 4}\left(2 - 2^{\mathcal{S}(\rho_n)}\right) \\
&\geq \mathcal{S}(\rho_n) + \frac{p}{\ln 4}\left(2 - (1 + \mathcal{S}(\rho_n))\right) \\
&= \left(1 - \frac{p}{\ln 4}\right)\mathcal{S}(\rho_n) + \frac{p}{\ln 4},
\end{aligned} \tag{68}$$

where the first inequality follows from $\log_2(1 - x) \leq -\frac{x}{\ln 2}$, the second inequality follows from $p \in [0, 1]$, and the third inequality follows from $2^x \leq 1 + x$ for $x \in [0, 1]$. By summing Eqs. (66) and (68) over $n \in [N]$, respectively, we obtain Eq. (49).

$\square$

## D MORE IMPLEMENTATION DETAILS OF THE AQER ALGORITHM

In this section, we provide a detailed discussion of the technical aspects of AQER, including the computation and optimization of the entanglement measure $\mathcal{S}$, the explicit construction of product-state approximations, and the trainability of variational circuit optimization in AQER. We also discuss why $\mathcal{S}$ serves as an efficient proxy for the global approximation error and how it facilitates scalable training on large quantum systems. Finally, we explain how to employ AQER with classical data.

### D.1 STEP I: COMPUTATION AND OPTIMIZATION OF THE ENTANGLEMENT MEASURE

The computation and optimization of the entanglement measure $\mathcal{S}$ in Step I of AQER requires only a limited amount of quantum resources. Since $\mathcal{S}$, which is defined as the sum of single-qubit Renyi entropies, consists solely of local terms, it can be efficiently evaluated using local measurements. Specifically, the Renyi entanglement entropy of any subsystem of constant size can be estimated via quantum state tomography on its reduced density matrix, which requires only $\mathcal{O}(1/\epsilon^2)$ measurements to achieve an $\epsilon$-precision estimate. Besides, for a candidate 2-qubit gate block in each iteration of Step I, only the single-qubit Renyi entropies of the two involved qubits are affected. Therefore, the change in $\mathcal{S}$ due to adding a new gate block equals the change in the sum of these two single-qubit entropies. Thus, in each iteration of Step I, the gate block that induces the largest decrease in the relevant single-qubit entropies is selected as the new structure. The parameters within these blocks is optimized using classical methods such as the Nelder-Mead algorithm. Through this procedure, entanglement is systematically reduced using only local information. This reduction allows efficient scaling to large quantum systems.

### D.2 STEP II: PARAMETER COMPUTATION

The parameters for single-qubit rotations in Step II of AQER can be computed explicitly and efficiently via single-qubit state tomography. Based on Theorem 3.1, the state $|v_T\rangle$ after Step I has reduced entanglement and can be well approximated by a product state, which is prepared from $|0\rangle^{\otimes N}$ by using single-qubit rotations. In Step II, the parameters $(\boldsymbol{\beta}_i, \boldsymbol{\gamma}_i)$ for the single-qubit rotations $R_Z(\boldsymbol{\beta}_i) R_Y(\boldsymbol{\gamma}_i)$ can be explicitly computed from the elements of the single-qubit reduced density matrices obtained via tomography. Specifically, according to Lemma B.2, each $\beta_i$ and $\gamma_i$ is directly computed from the matrix elements $\rho_{00}$, $\rho_{11}$, and $\rho_{10}$ of the corresponding reduced density matrix. As mentioned in the discussion regarding Step I, the tomography for each qubit requires $\mathcal{O}(1/\epsilon^2)$ measurements to achieve an $\epsilon$-precision estimate, which is independent of the total system size $N$. Once obtained, these parameters provide an explicit product-state approximation of $|v_T\rangle$ without any iterative optimization. This yields a high-fidelity initialization for Step III.

### D.3 STEP III: THE TRAINABILITY OF PARAMETER OPTIMIZATION

By reducing entanglement in Step I and explicitly constructing single-qubit gates in Step II, AQER mitigates the effects of barren plateaus on subsequent parameter optimization in Step III. Generally, in the training of VQAs, barren plateau phenomena occur when the loss function is initialized near its average value with exponentially small gradients. This makes parameter optimization extremely challenging. For loss functions defined as the infidelity with respect to an $N$-qubit target state, the average loss for randomly initialized parameterized circuits scales as $1 - \mathcal{O}(1/2^N)$. In contrast, in Step III of AQER, the entanglement measure $\mathcal{S}$ is suppressed before full optimization. By Theorem 3.1, smaller $\mathcal{S}$ values correspond to states with smaller infidelity loss. Therefore, the initial point of the variational circuit is effectively positioned away from the barren plateau region. This ensures that subsequent optimization starts from a well-conditioned region where gradients are meaningful, mitigating the impact of barren plateaus and improving trainability. This mechanism differs from previous strategies for avoiding or mitigating barren plateaus that primarily reshape the optimization landscape or amplify gradient signals (e.g., via architectural constraints (Pesah et al., 2021; Zhang et al., 2022a), smart parameter initialization strategies (Grant et al., 2019; Zhang et al., 2022b; Wang et al., 2024c; Peng et al., 2025), or local cost functions (Cerezo et al., 2021b)) while not necessarily guaranteeing a low-loss starting state. Moreover, AQER mainly mitigates parameter-space barren plateaus, which does not resolve data-induced barren plateaus that can still cause severe trainability and generalization issues (Wang et al., 2024b; Zhang et al., 2024).

### D.4 AQER: ENTANGLEMENT MEASURE AS A PROXY FOR APPROXIMATION ERROR

Using the entanglement measure $\mathcal{S}$ as a proxy for the global approximation error provides multiple advantages. Directly optimizing the infidelity loss requires global measurements, which are computationally expensive and sensitive to statistical variance, especially in large systems. In contrast, $\mathcal{S}$ can be efficiently evaluated and reduced using only local measurements. By first reducing $\mathcal{S}$, AQER not only lowers the approximation error but also prepares the circuit in a favorable regime for

further global optimization. This strategy enhances both the efficiency of evaluation and the overall trainability and scalability of AQER on large-qubit quantum systems.

### D.5 AQER with Classical Data

For classical data, AQER is implemented through classical simulation while maintaining the same three-step structure. In particular, the quantum state corresponding to classical data is represented in the form of classical vectors or tensor networks.

**Classical Simulation:** In Step I, the entanglement measure $\mathcal{S}$ is computed by classically evaluating reduced density matrices and their Renyi entropies. Gate optimization proceeds through classical simulation of candidate blocks, with parameters optimized using gradient-based methods with exact gradient information. In Step II, single-qubit rotation parameters $(\beta_i, \gamma_i)$ are extracted directly from computed reduced density matrices according to Lemma B.2. Step III utilizes classical simulation for exact loss evaluation and gradient computation without statistical noise.

## E    Experimental Setup

In this section, we first describe the construction and preprocessing of both classical and quantum datasets. We then present the hyperparameter configurations of the reference quantum data loaders employed in our experiments.

**Numerical simulation settings.** All experiments presented in this work were performed using classical simulators. For computational tasks involving fewer than 20 qubits, we employed Penny-Lane (Bergholm et al., 2018), a Python-based package. For larger quantum systems with 20 or more qubits, we utilized PastaQ (Torlai and Fishman, 2020), a Julia-based package that employs tensor network techniques to support efficient simulation of large-scale quantum systems.

### E.1    Construction of classical and quantum datasets

**Classical data.** For MNIST, each image is zero-padded to $32 \times 32$, flattened into a 1024-dimensional vector $\boldsymbol{v}$, and normalized to have unit $\ell_2$ norm. The corresponding target state is defined by the amplitude encoding $|\boldsymbol{v}\rangle = \sum_{j=1}^{2^N} \boldsymbol{v}_j |j\rangle$ with $N = 10$ qubits. For CIFAR-10, each $32 \times 32$ RGB image is flattened by concatenating the three color channels into a 3072-dimensional vector, zero-padded to 4096 dimensions, and then normalized. The second half of the vector is further treated as the imaginary part and combined with the first half to construct the compact encoding (Blank et al., 2022): $|\boldsymbol{v}\rangle = \sum_{j=1}^{2^N} (\boldsymbol{v}_j + i\,\boldsymbol{v}_{j+2048})|j\rangle$ with $N = 11$ qubits, which serves as the target to evaluate AQER under different encoding schemes. For SST-2, each sentence is initially represented by a 1024-dimensional embedding vector obtained from a pretrained Sentence-BERT model. The embeddings are then normalized, and the corresponding target states are defined via amplitude encoding using $N = 10$ qubits.

**Quantum data.** We consider two types of quantum datasets: synthetic states generated from random quantum circuits (RQCs) and ground states of the one-dimensional transverse-field Ising model (1D TFIM). These datasets serve as typical examples of physically relevant states arising from quantum circuit evolutions or quantum many-body systems. Specifically, the first dataset is consists of $M = 50$ states generated from different RQCs on the state $|0\rangle$. Each RQC is sampled from the set $RandomShuffle\left(\{CZ_{p_k,q_k}\}_{k=1}^{W} \cup \{R_{\sigma_k,p_k}(\theta_k)\}_{k=W+1}^{4W}\right)$, where $1 \leq p_k \neq q_k \leq N$ are randomly sampled from $[N]$. Each $R_{\sigma,p_k}$ is a single-qubit rotation on the qubit $p_k$ with $\sigma_k \sim \mathrm{Uniform}\{X, Y, Z\}$ and $\theta_k \sim \mathrm{Uniform}[0, 2\pi]$. Such circuits produce highly entangled states when the number of two-qubit gates $W \geq \mathcal{O}(N)$ that serve as representative synthetic quantum data. In practice, we use $N = 10$ and $W = 40$. The second dataset consists of ground states of the 1D TFIM, which is defined by the Hamiltonian $H_{\mathrm{TFIM}} = -\sum_{i=1}^{N-1} J Z_i Z_{i+1} - \sum_{i=1}^{N} g X_i$. We consider coefficients near the phase transition point, i.e. $g = 1$ and $J \in \{0.8, 0.9, 1, 1.1, 1.2\}$ to construct datasets with the size $M = 5$ for $N \in \{10, 20, 30, 40, 50\}$.

Table 2: Feasible numbers of CNOT/CZ gates used in different encoding methods. The term $N$ is the number of qubits, and the term $k \in \mathbb{N}$ can be any positive integer.

| Method | two-qubit gate count $G$ |
|---|---|
| AQCE ($\mathbb{C}$) | $G = 15k$ |
| AQCE ($\mathbb{R}$) | $G = 10k$ |
| MPS ($\mathbb{C}$) | $G = 3(N-1)k$ |
| MPS ($\mathbb{R}$) | $G = 2(N-1)k$ |
| HEC | $G = \lceil \frac{1}{2}Nk \rceil$ |
| AQER (Ours) | $G = k$ |

### E.2 HYPERPARAMETER SETTINGS OF REFERENCE QUANTUM DATA LOADERS

Here, we provide the hyperparameter settings of the reference quantum data loaders used in our experiments. The descriptions of these methods can be found in Section A.3.

**Automatic quantum circuit encoding (AQCE).** The AQCE method performs the two-qubit unitary updation sequentially in a forward–backward manner several times before adding new gates into the circuit. In the experiment, we add 5 new unitaries for each adding-gate step, followed by 200 rounds of forward–backward gate updation.

**Hardware-efficient circuits (HEC).** We adopt a layered hardware-efficient (HE) architecture, where each layer consists of an $R_Y$ rotation layer, an $R_Z$ rotation layer, and a CNOT layer acting on adjacent qubit pairs. Specifically, for the $i$-th layer, CNOT gates are applied to qubit pairs $(2n, (2n+1)\%N)$, $0 \le 2n \le N-1$ when $i\%2 = 0$ and $(2n+1, (2n+2)\%N)$, $0 \le 2n \le N-2$ when $i\%2 = 1$. Parameters in HEC are initialized randomly from the uniform distribution over $[0, 2\pi]$. Training is performed using the Adam optimizer for 2000 iterations, which is consistent with the setting in AQER.

We remark that reference AQL methods introduce different constraints on the number of hardware-available two-qubit gates, such as CNOT and CZ. We summarize the feasible two-qubit gate count under these methods in Tab. 2. Specifically, the decomposition of an arbitrary two-qubit unitary requires two and three CNOT/CZ gates in the real and general complex cases, respectively (Vatan and Williams, 2004).

### E.3 QUANTUM KERNEL METHOD WITH CROSS-VALIDATION

We employ the quantum kernel method (QKM) (Rebentrost et al., 2014; Wang et al., 2021) for binary classification. QKM employs a quantum feature map to embed classical input vectors $\boldsymbol{x}^{(i)}$ into quantum states $|\psi(\boldsymbol{x}^{(i)})\rangle$, which are used to construct a kernel matrix $K_{ij} = |\langle\psi(\boldsymbol{x}^{(i)})|\psi(\boldsymbol{x}^{(j)})\rangle|^2$ that quantifies pairwise similarities between data points. Denote by $y_i \in \{-1, 1\}$ the label of data $\boldsymbol{x}^{(i)}$. Then, the training of support vector machines (SVM) corresponds to solving the dual optimization problem

$$\max_{\boldsymbol{\alpha}} \left[ \sum_i \boldsymbol{\alpha}_i - \frac{1}{2} \sum_{ij} \boldsymbol{\alpha}_i \boldsymbol{\alpha}_j y_i y_j K_{ij} \right], \text{ s.t. } 0 \le \boldsymbol{\alpha}_i \le C, \sum_i \boldsymbol{\alpha}_i y_i = 0,$$

where $C$ is the penalty parameter that is set to 1 by default. Once the parameter $\boldsymbol{\alpha}$ of the QKM is trained, the predicted label for a new sample $\boldsymbol{x}$ is computed using the decision function

$$\hat{y} = \text{sign} \left[ \sum_{i=1}^N \boldsymbol{\alpha}_i y_i K(\boldsymbol{x}^{(i)}, \boldsymbol{x}) \right],$$

where $K(\boldsymbol{x}^{(i)}, \boldsymbol{x}) = |\langle\psi(\boldsymbol{x}^{(i)})|\psi(\boldsymbol{x})\rangle|^2$ is the kernel between the training sample $\boldsymbol{x}^{(i)}$ and the new sample $\boldsymbol{x}$.

In the experiment, we compare the approximate encoding from AQER with the exact amplitude encoding. We employ stratified $k$-fold cross-validation (with $k = 5$) on the dataset with size 200 to estimate classification performance. For each fold, the SVM is trained on the training subset using

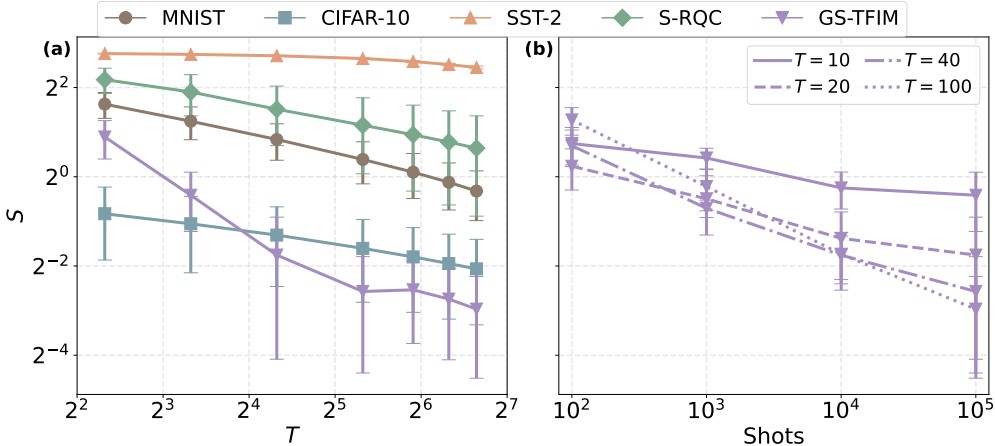

Figure 6: Performance of AQER on entanglement suppression across MNIST, CIFAR-10, SST-2, S-RQC, and GS-TFIM datasets, distinguished by different colors and markers. (a) Entanglement measure value $S$ versus different $T \in \{5, 10, 20, 40, 60, 80, 100\}$ after Step III of AQER across all datasets. (b) Entanglement measure value $S$ versus different measurement shots for the GS-TFIM dataset, with different $T \in \{10, 20, 40, 100\}$.

the precomputed kernel matrix, and predictions are made on the held-out validation subset using the corresponding submatrix of $K$. The reported error is the mean over all folds.

## F  ADDITIONAL EXPERIMENTAL RESULTS

In this section, we present additional numerical results about the performance of the AQER on both classical and quantum datasets.

**Step I of AQER effectively reduces entanglement with increasing iteration times.** We first examine how AQER progressively lowers the entanglement measure value $S$ for MNIST, CIFAR-10, SST-2, S-RQC, and GS-TFIM datasets. Fig. 6(a) shows $S$ after Step III of AQER for all datasets with varying $T \in \{5, 10, 20, 40, 60, 80, 100\}$. For most datasets, increasing $T$ consistently decreases $S$, indicating that AQER effectively reduces the entanglement of the target state by gradually expanding the circuit. The GS-TFIM dataset exhibits minor fluctuations due to statistical noise caused by a limited number of measurements, but the overall decreasing trend remains clear. These results corroborate the effectiveness of AQER in progressively mitigating entanglement as $T$ grows.

**Effect of shot number on entanglement reduction.** We next study the effect of different measurement shots on entanglement reduction. Fig. 6(b) illustrates the entanglement measure $S$ for the GS-TFIM dataset, with $T \in \{10, 20, 40, 100\}$. Increasing the number of shots generally reduces $S$, mirroring the trend observed in the main text for infidelity. This reduction is more pronounced for larger $T$. For instance, increasing the measurement shots from $10^2$ to $10^5$ reduces $S$ by roughly a factor of 2 for $T = 10$, whereas for $T = 100$ the reduction reaches approximately 16-fold. These results indicate that sufficient measurement shots significantly enhance the capability of AQER to suppress entanglement.

**Step III of AQER effectively reduces infidelity with increasing iteration steps.** We then investigate the effect of different numbers of Step III iterations $T_3 \in \{40, 80, 160, 320, 640, 1280, 2000\}$ on AQER performance. Fig. 7 shows the infidelity for MNIST, CIFAR-10, SST-2, S-RQC, and GS-TFIM datasets, with each subfigure corresponding to $T = 10, 20, 40$, and $100$. For all cases, increasing $T_3$ steadily decreases the infidelity, indicating that longer Step III optimization effectively improves the approximation quality. This reduction is more pronounced for larger $T$, demonstrating that AQER benefits from combining a larger initial circuit with more extensive optimization in Step III.

**Binary classification with AQER.** We further examine the classification performance of QKM using AQER-loaded states. Specifically, we consider a binary task on MNIST digits 0 vs 1 using

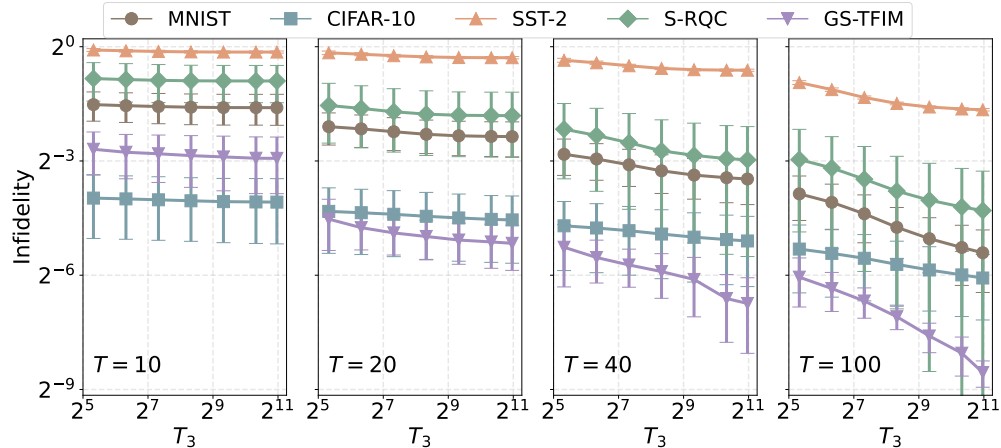

Figure 7: AQER infidelity with different optimization iterations in Step III, i.e. $T_3 \in \{40, 80, 160, 320, 640, 1280, 2000\}$. Each subfigure corresponds to $T = 10, 20, 40$, and $100$, respectively.

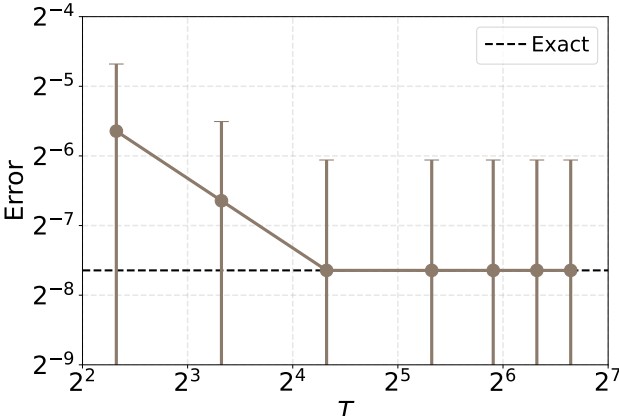

Figure 8: Binary classification error on MNIST digits 0 vs 1 using AQER-loaded states with $T \in \{5, 10, 20, 40, 60, 80, 100\}$, compared to exact loading (black dashed line).

AQER-loaded states with different $T$ values. The results are shown in Fig. 8, where the error rates for $T \in \{5, 10, 20, 40, 60, 80, 100\}$ are compared against exact loading. We observe that the classification error decreases steadily as $T$ increases, and from $T \geq 20$ onward the performance already matches the exact-loading error. This indicates that AQER can achieve near-exact downstream performance on classification tasks with relatively small circuit sizes.

**Resource–accuracy trade-offs of AQER versus AQL baselines.** We further compare the accuracy–resource trade-offs of different AQL methods on the S-RQC dataset. Fig. 9 plots the infidelity achieved by AQER, HEC, AQCE, and MPS as a function of four resource metrics: circuit depth, number of trainable parameters, total number of measurement shots, and classical training time in seconds. Specifically, the circuit depth is defined as the number of sequential gate layers in the compiled quantum circuit under full parallelization of commuting gates on different qubits. We count only layers that contain at least one two-qubit gate, since in practice two-qubit gates are typically an order of magnitude slower than single-qubit gates. The parameter count is the number of all independent continuous parameters in the quantum circuit (e.g., rotation angles). The measurement shots are the number of quantum measurements used in the entire AQL procedure. The training time is measured on a laptop equipped with an Apple M2 chip and 8 GB of RAM. Overall, AQER achieves lower infidelity than the other methods under comparable or lower circuit depth and parameter counts, indicating a more efficient use of circuit expressivity. For measurement shots and training time, the

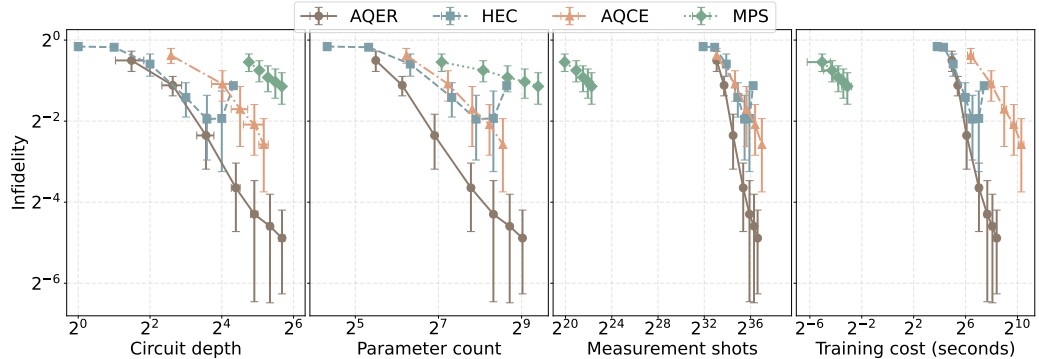

Figure 9: Infidelity as the function of metrics for different AQL methods on the S-RQC dataset. We consider four metrics: the circuit depth, the parameter count, the number of measurement shots, and the training cost in seconds. Each data point is the average over 5 samples.

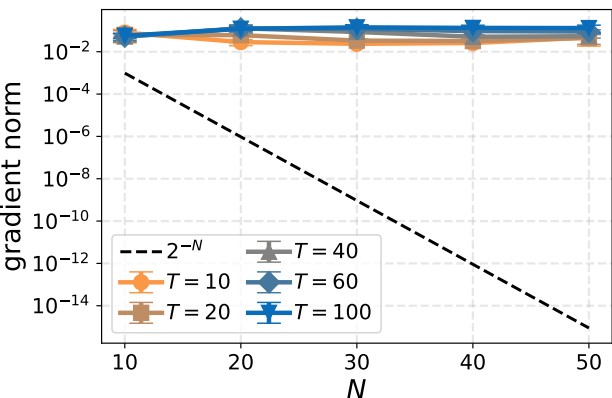

Figure 10: Scaling of AQER gradient norms with the number of qubits for the TFIM dataset. The plot shows the gradient norm at the initial stage of Step III training, as a function of system size $N \in \{10, 20, 30, 40, 50\}$, for different circuit sizes $T \in \{10, 20, 40, 60, 100\}$. The black dashed line depicts an exponentially decaying reference curve proportional to $2^{-N}$.

behavior is more nuanced. The MPS baseline, owing to the simplicity of its algorithm, requires fewer shots and shorter training time, but this comes at the cost of substantially worse performance in terms of circuit depth, two-qubit gate count, and parameter count when realized as a circuit, as well as significantly higher infidelity. When compared against circuit-based baselines (AQCE and HEC), AQER consistently achieves lower infidelity with comparable or lower measurement overhead and training cost. Therefore, AQER achieves a favorable resource-accuracy trade-off among circuit-based AQL methods.

**Gradient scaling of AQER with large qubit number.** We analyze the scaling of AQER gradient norms on the TFIM dataset with large qubit numbers to demonstrate that the Step III optimization in AQER is free from the barren plateau issue. For system sizes $N \in \{10, 20, 30, 40, 50\}$ and circuit sizes $T \in \{10, 20, 40, 60, 100\}$, we compute the gradient norm with respect to all trainable parameters at the beginning of the optimization. The result is shown in Fig. 10. Across all system sizes and circuit sizes, the initial gradient norms remain on the order of $10^{-2}$ and do not exhibit any exponential decay with $N$, in clear contrast to the exponentially vanishing reference curve $2^{-N}$ shown in the figure. These observations provide numerical evidence that the trainability of AQER remains stable as the system size increases from $N = 10$ to $N = 50$.

**Robustness of AQER under noisy channels.** We perform noisy simulations on the $N = 10$ TFIM task to assess the performance of AQER in realistic NISQ regimes. In particular, after each single-qubit and two-qubit gate layer, we apply a depolarizing channel with error rates $p_1 \in$

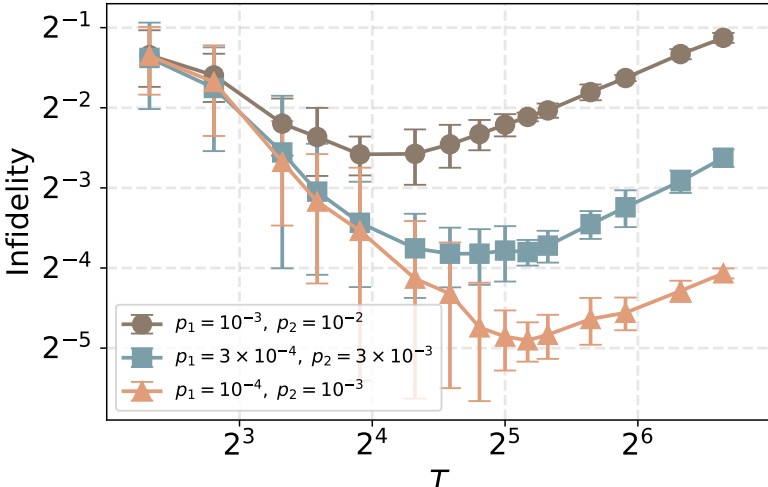

Figure 11: Infidelity of AQER under global depolarizing noise for the $N = 10$ TFIM task. The plot shows the final infidelity as a function of the two-qubit gate count $T$ for three noise levels, where single-qubit and two-qubit gate error rates are $(p_1, p_2) \in \{(10^{-3}, 10^{-2}), (3 \times 10^{-4}, 3 \times 10^{-3}), (10^{-4}, 10^{-3})\}$.

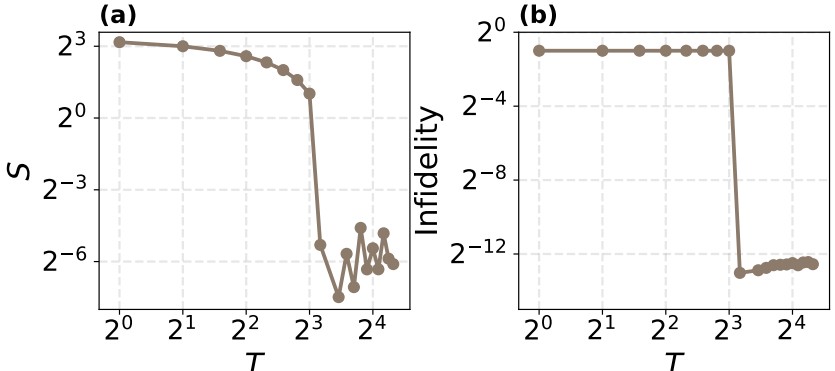

Figure 12: Performance of AQER on the $N = 10$ GHZ state. (a) Entanglement measure $\mathcal{S}$ as a function of the two-qubit gate count $T$. (b) Infidelity as a function of $T$.

$\{10^{-3}, 3 \times 10^{-4}, 10^{-4}\}$ and $p_2 = 10p_1$, respectively, which corresponds to representative error ranges reported for current quantum devices. For each noise setting and two-qubit gate count $T \in \{5, 7, 10, 12, 15, 20, 24, 28, 32, 36, 40, 50, 60, 80, 100\}$, we run AQER and record the final infidelity. The result is shown in Fig. 11. Across all three noise levels, we observe a noise-dependent optimal circuit size $T^*$: as $T$ increases from very small values, the infidelity decreases, which indicates that entanglement-guided circuit growth improves approximation quality before noise accumulation becomes dominant. Beyond $T^*$, further increasing $T$ leads to a gradual increase in infidelity as the effect of noise outweighs the expressivity gains. Moreover, as the physical error rates decrease from $(p_1, p_2) = (10^{-3}, 10^{-2})$ to $(10^{-4}, 10^{-3})$, the best achievable infidelity improves from above $2^{-3}$ to around $2^{-5}$, and the optimal $T^*$ increases. Altogether, these results demonstrate that AQER remains effective and robust in NISQ-level noise regimes.

**AQER for preparing the GHZ state.** We evaluate AQER on the $N = 10$ GHZ state as a case study of its performance on highly entangled yet structurally simple states. The GHZ state is highly entangled in the sense that each single-qubit reduced density matrix is maximally mixed and the initial entanglement measure satisfies $\mathcal{S} = N$. Nonetheless, AQER is able to reduce this entanglement very efficiently. As shown in Fig. 12, the entanglement measure $\mathcal{S}$ decays rapidly with the two-qubit

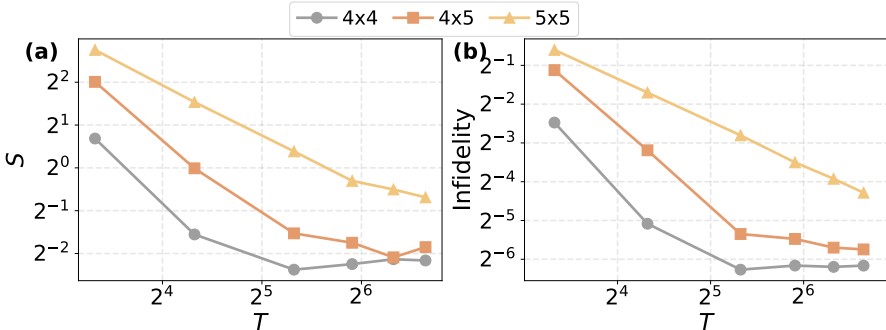

Figure 13: Performance of AQER on 2D random circuit states. Target states are generated by a depth-4 nearest-neighbor circuit on $4 \times 4$, $4 \times 5$, and $5 \times 5$ lattices. (a) Entanglement measure $\mathcal{S}$ versus the two-qubit gate count $T \in \{10, 20, 40, 60, 80, 100\}$. (b) Infidelity versus $T$ for the same set of 2D random circuit states.

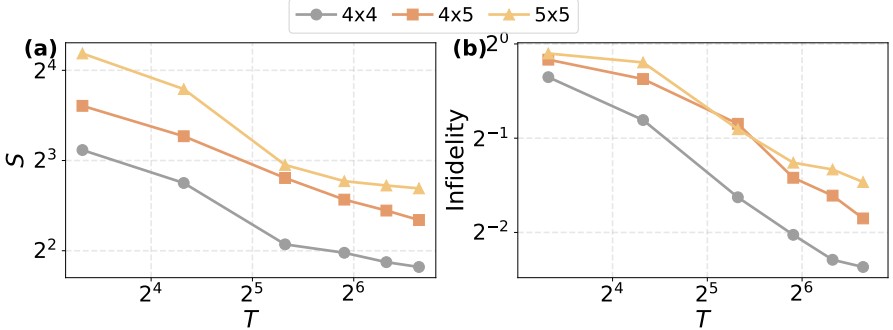

Figure 14: Performance of AQER on 2D XXZ ground states at the critical point. (a) Entanglement measure $\mathcal{S}$ as the function of the two-qubit gate count $T \in \{10, 20, 40, 60, 80, 100\}$ for $4 \times 4$, $4 \times 5$, and $5 \times 5$ lattices. (b) The infidelity as the function of $T$.

gate count $T$ and drops below $2^{-3}$ already at $T = 9$. The corresponding infidelity is reduced to below $2^{-12}$ using only 9 two-qubit gates. This example highlights that a quantum state can be highly entangled while still being structurally simple, and that in such cases AQER can efficiently find low-depth circuits that achieve very low infidelity despite the large initial value of $\mathcal{S}$.

**AQER for preparing 2D random circuit states.** We consider preparing 2D random circuit states on rectangular lattices to evaluate AQER on physically relevant targets. The target state is generated by a depth-4 circuit composed of alternating single-qubit and CZ layers. In each single-qubit layer, every qubit is acted on by a randomly chosen rotation from $\{R_X, R_Y, R_Z\}$ with a uniformly sampled angle. In each CZ layer, CZ gates are applied to nearest neighbors along horizontal or vertical directions according to one of four distinct tilings of the 2D grid (two horizontal and two vertical patterns). These four CZ patterns are cycled over the four layers so that, across the entire circuit, all nearest-neighbor couplings are activated. We consider $4 \times 4$, $4 \times 5$, and $5 \times 5$ lattices and apply AQER with two-qubit gate counts $T \in \{10, 20, 40, 60, 80, 100\}$. The resulting entanglement measures and infidelities are shown in Fig. 13. For all lattice sizes, the entanglement measure $\mathcal{S}$ decreases by at least a factor of 8 as $T$ increases from 10 to 100, indicating that the entanglement-reduction strategy remains effective for the 2D random circuit state. The corresponding infidelities also decrease as $T$ increases. For example, on the $4 \times 4$ lattice the infidelity drops from above $2^{-3}$ at $T = 10$ to around $2^{-6}$ at $T = 100$, and on the $5 \times 5$ lattice it decreases from above $2^{-1}$ to below $2^{-4}$. These results demonstrate that AQER can substantially and consistently decrease both the entanglement measure and the approximation error for shallow 2D random circuit states as the circuit size $T$ increases.

**AQER for preparing 2D XXZ ground states at the critical point.** We consider preparing the ground state of the spin-$\frac{1}{2}$ XXZ model on lattices to evaluate AQER on 2D many-body quantum

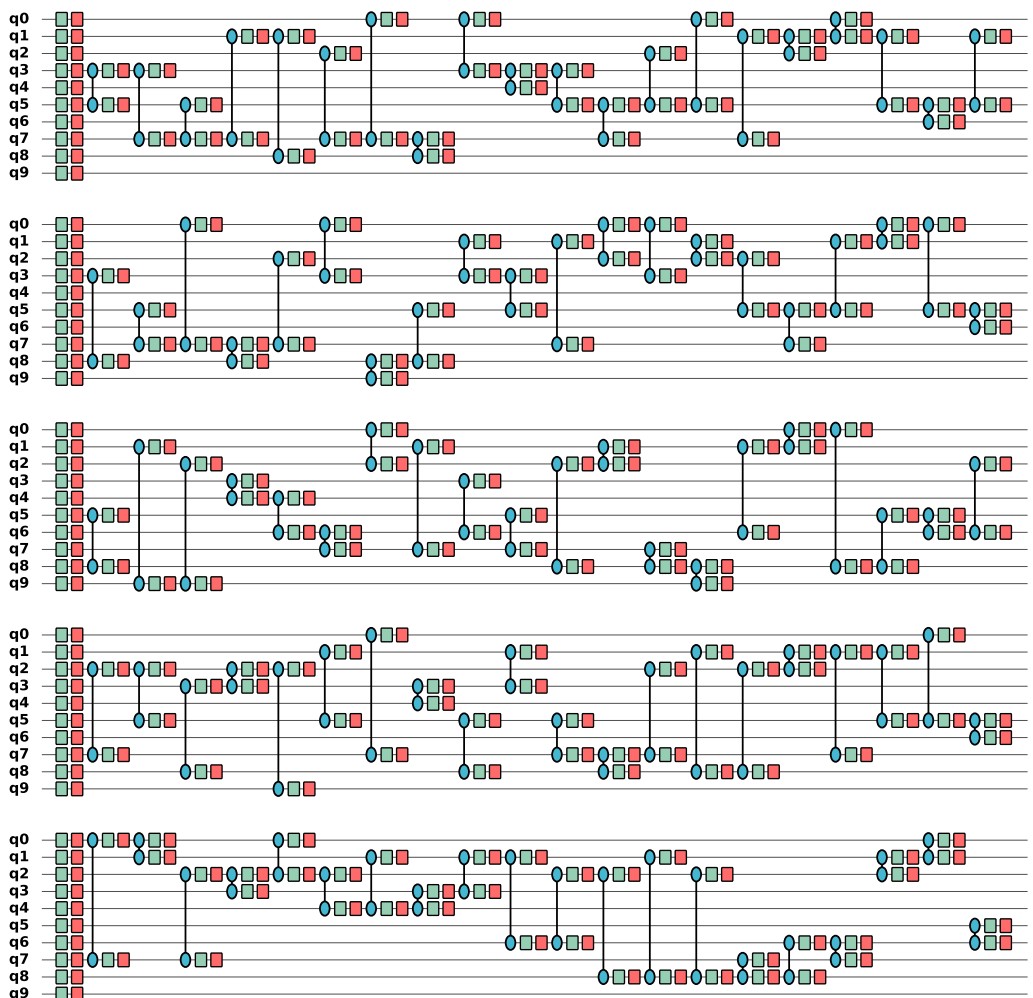

Figure 15: Loading circuits generated from AQER with $T = 20$ for the MNIST dataset. Green and red blocks denote $R_Y$ and $R_Z$ gates, respectively. Blue circles connected by lines denote $R_{ZZ}$ gates.

states. The Hamiltonian is given by

$$H_{\text{XXZ}} = \sum_{\langle i,j \rangle} \Big[ J_{xy}\big(\sigma_i^x \sigma_j^x + \sigma_i^y \sigma_j^y\big) + J_z\, \sigma_i^z \sigma_j^z \Big], \tag{69}$$

where $\langle i, j \rangle$ runs over nearest-neighbor pairs on the 2D grid and $\sigma^{x,y,z}$ denote Pauli matrices. We focus on the critical point $J_{xy} = J_z$ and consider $4 \times 4$, $4 \times 5$, and $5 \times 5$ lattices. For each lattice, we apply AQER with two-qubit gate counts $T \in \{10, 20, 40, 60, 80, 100\}$. The entanglement measures and infidelities are shown in Fig. 14. For all lattice sizes, the entanglement measure $\mathcal{S}$ decreases steadily as $T$ increases, typically by a factor of about 2–3 when $T$ grows from 10 to 100 (e.g., from above $2^3$ to below $2^2$ on the $4 \times 4$ lattice and from above $2^4$ to below $2^3$ on the $5 \times 5$ lattice). The corresponding infidelities also decrease monotonically with $T$ by more that a factor of 2. These results show that, AQER can reliably compress entanglement and reduce the approximation error for 2D quantum many-body systems, with both $\mathcal{S}$ and the infidelity decrease systematically as the circuit size $T$ increases.

**Quantum circuit visualization.** We visualize the quantum circuits generated by the AQER algorithm with $T = 10$ for the MNIST, CIFAR-10, SST-2, S-RQC, and GS-TFIM datasets in Figures 15-19, respectively. Each figure shows the circuit layouts of five examples, including single-qubit rotation gates $R_Y, R_Z$ and two-qubit $R_{ZZ}$ gates.

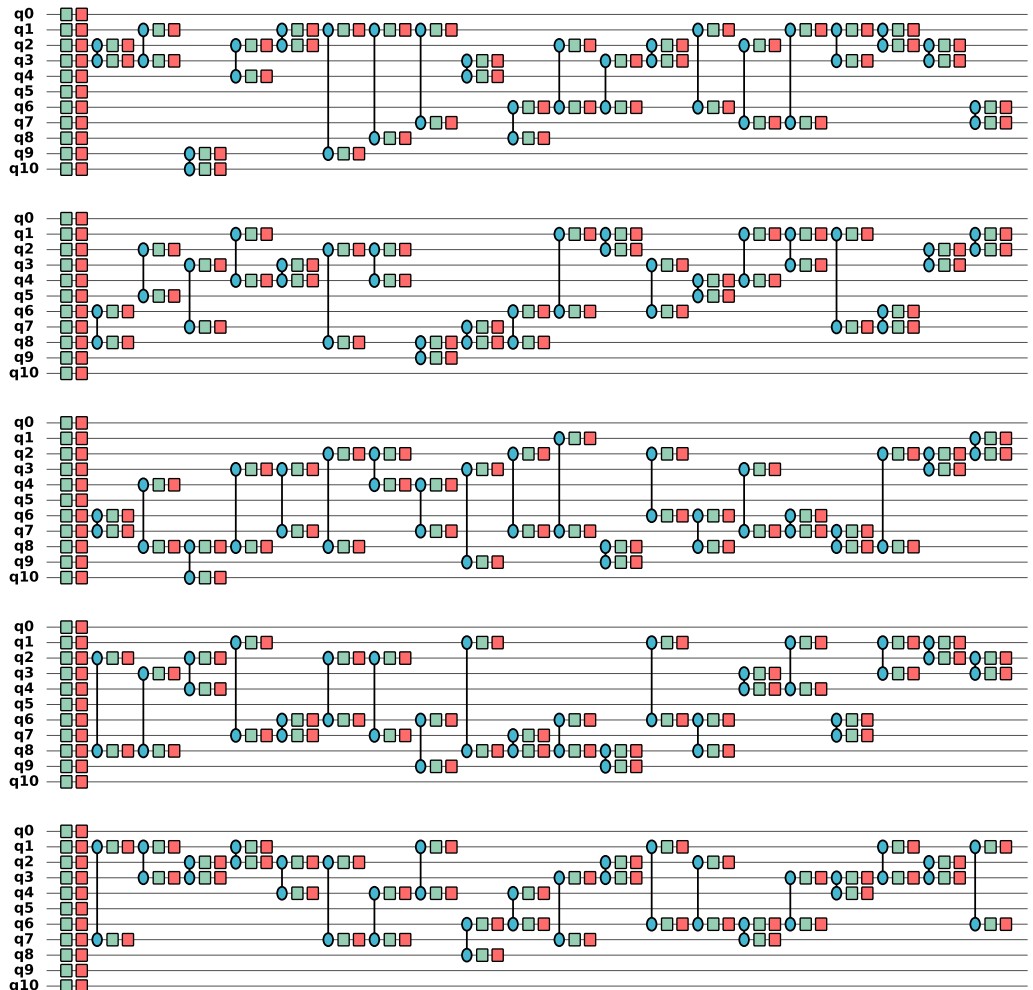

Figure 16: Loading circuits generated from AQER with $T = 20$ for the CIFAR-10 dataset. Green and red blocks denote $R_Y$ and $R_Z$ gates, respectively. Blue circles connected by lines denote $R_{ZZ}$ gates.

# G COMPUTATIONAL COMPLEXITY OF AQER

Here we provide the computational complexity of AQER. We separate the analysis into the classical data setting, where the target state is represented explicitly as a state vector on a classical computer, and the quantum data setting, where one has access only to copies of a target quantum state prepared on a quantum device. Throughout, we denote by $N$ the number of qubits and by $d = 2^N$ the dimension of classical data vector. The number of iterations in the Step I of AQER is denoted by $T$, and the number of iterations in the Step III of AQER is denoted by $T_3$.

## G.1 CLASSICAL DATA

We first analyze the cost of AQER when the target state is given as an explicit state vector $v \in \mathbb{C}^d$ on a classical computer. In this regime, all quantities required by AQER (reduced density matrices, entropies, gradients) can be computed exactly from $v$.

**Cost of applying gates.** Each gate used in AQER is either diagonal ($R_Z$ and $R_{ZZ}$) or 2-sparse (for $R_Y$). Therefore, applying a single- or two-qubit gate $U$ to the state vector $v \in \mathbb{C}^d$ can be implemented as a matrix-vector multiplication with a diagonal or 2-sparse matrix, and thus costs $\mathcal{O}(d)$ arithmetic operations. A forward pass of the circuit with $\mathcal{O}(T)$ then costs $\mathcal{O}(dT)$.

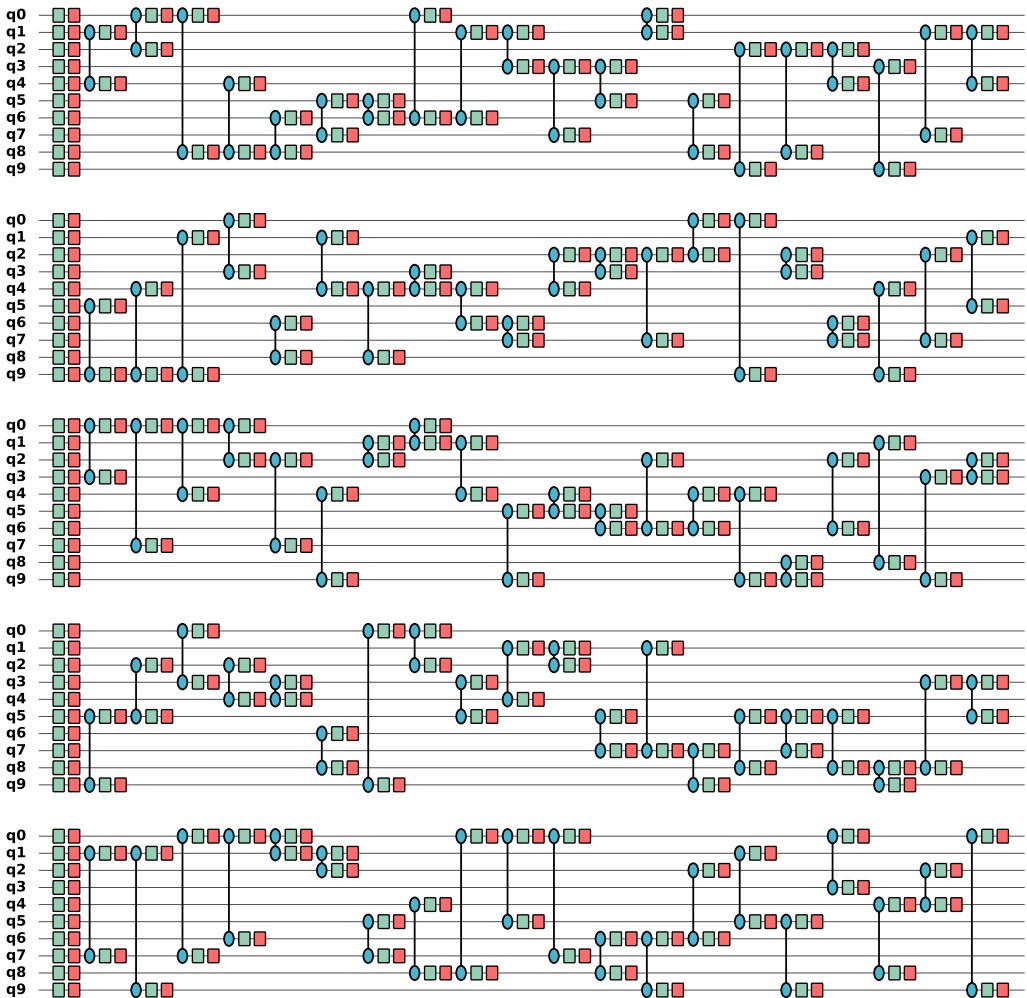

Figure 17: Loading circuits generated from AQER with $T = 20$ for the SST-2 dataset. Green and red blocks denote $R_Y$ and $R_Z$ gates, respectively. Blue circles connected by lines denote $R_{ZZ}$ gates.

**Cost of computing reduced density matrices and entropies.**    Let the state vector in the computational basis be written as

$$\boldsymbol{v} = \sum_{i_1=0}^{1} \cdots \sum_{i_N=0}^{1} v_{i_1,\ldots,i_N} \, \boldsymbol{e}_{i_1,\ldots,i_N}. \tag{70}$$

Then, the reduced density matrix (RDM) of qubits $(1, 2)$ is given by

$$\rho_{j_1,j_2;k_1,k_2} = \sum_{i_3,\ldots,i_N} v_{j_1,j_2,i_3,\ldots,i_N} v^{*}_{k_1,k_2,i_3,\ldots,i_N}, \qquad j_1, j_2, k_1, k_2 \in \{0,1\}. \tag{71}$$

Computing this RDM requires summing over all $2^{N-2}$ assignments of $(i_3, \ldots, i_N)$, and hence costs $\mathcal{O}(2^N) = \mathcal{O}(d)$ operations. The same complexity applies to RDMs of any constant number of qubits and any choice of qubit indices. Once the RDMs are available, the single-qubit entropies and the entanglement measure used in AQER can be obtained by diagonalizing constant-size matrices, which incurs only negligible additional cost. Thus, the cost of computing the RDMs and corresponding entropies for a constant number of qubits is $\mathcal{O}(d)$.

**Step I: greedy entanglement reduction.**    In Step I, AQER greedily builds a circuit with $5T$ gates by iteratively adding single- and two-qubit gates that reduce the entanglement measure. At each iteration, the algorithm evaluates at most $\mathcal{O}(N^2)$ candidate qubit pairs, and evaluating one candidate

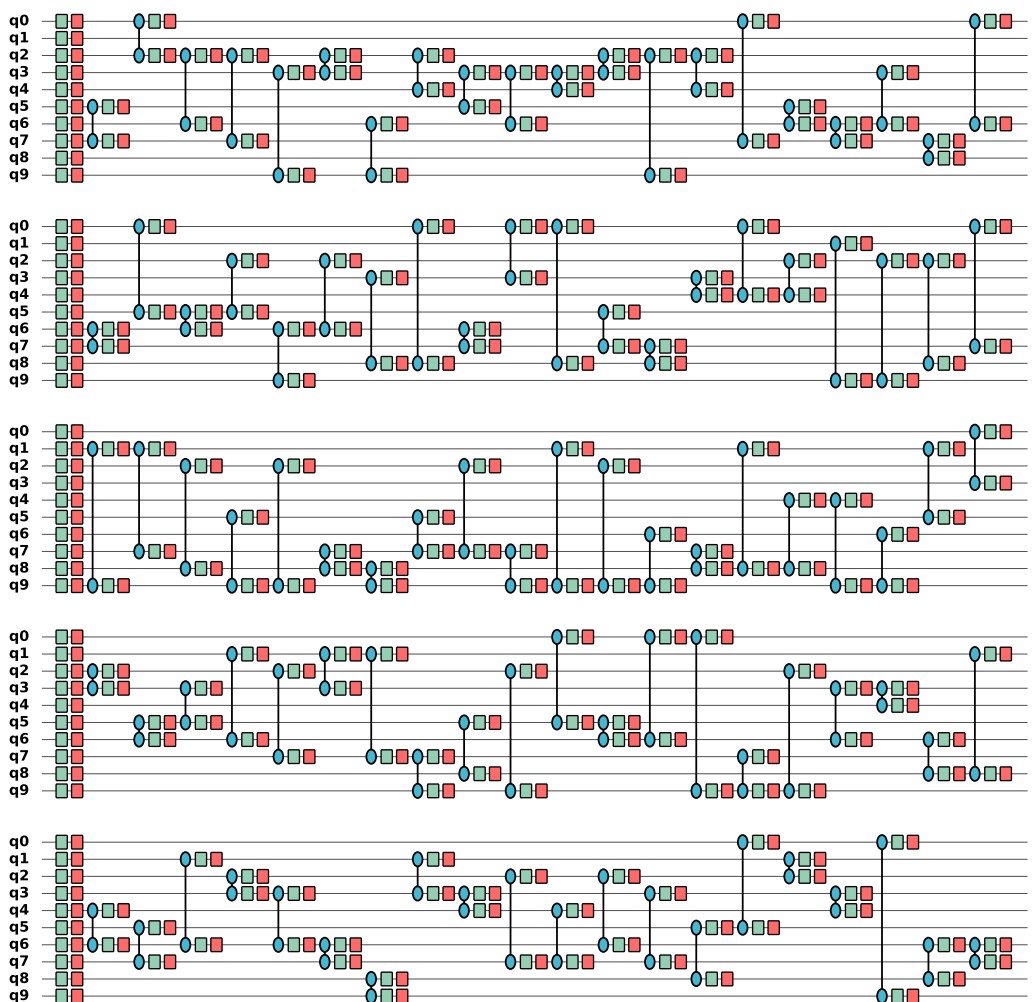

Figure 18: Loading circuits generated from AQER with $T = 20$ for the S-RQC dataset. Green and red blocks denote $R_Y$ and $R_Z$ gates, respectively. Blue circles connected by lines denote $R_{ZZ}$ gates.

requires $\mathcal{O}(d)$ time. Thus, the total cost of Step I is

$$\mathcal{O}(d\,T\,N^2). \tag{72}$$

**Step II: product state approximation.** In Step II, AQER computes the single-qubit RDMs for all $N$ qubits and uses these to initialize the parameters of single-qubit rotations. The cost of computing $N$ single-qubit RDMs is $\mathcal{O}(dN)$, and the subsequent calculation of the $2N$ gate parameters requires only $\mathcal{O}(N)$ operations. Therefore, the complexity of Step II is

$$\mathcal{O}(dN). \tag{73}$$

**Step III: gradient-based paramter fine-tuning.** Step III performs $T_3$ iterations of gradient-based optimization over all trainable parameters in the AQER circuit. The total number of parameters is

$$P = 5T + 2N, \tag{74}$$

where $5T$ comes from the single- and two-qubit rotations in the $T$ iterations of Step I, and $2N$ comes from additional single-qubit rotations determined in Step II.

To estimate the complexity of one iteration in Step III, we consider the cost of computing the gradient of the loss function with respect to all $P$ parameters. Using either the parameter-shift rule or automatic differentiation on the state-vector simulation, computing each partial derivative involves two numbers

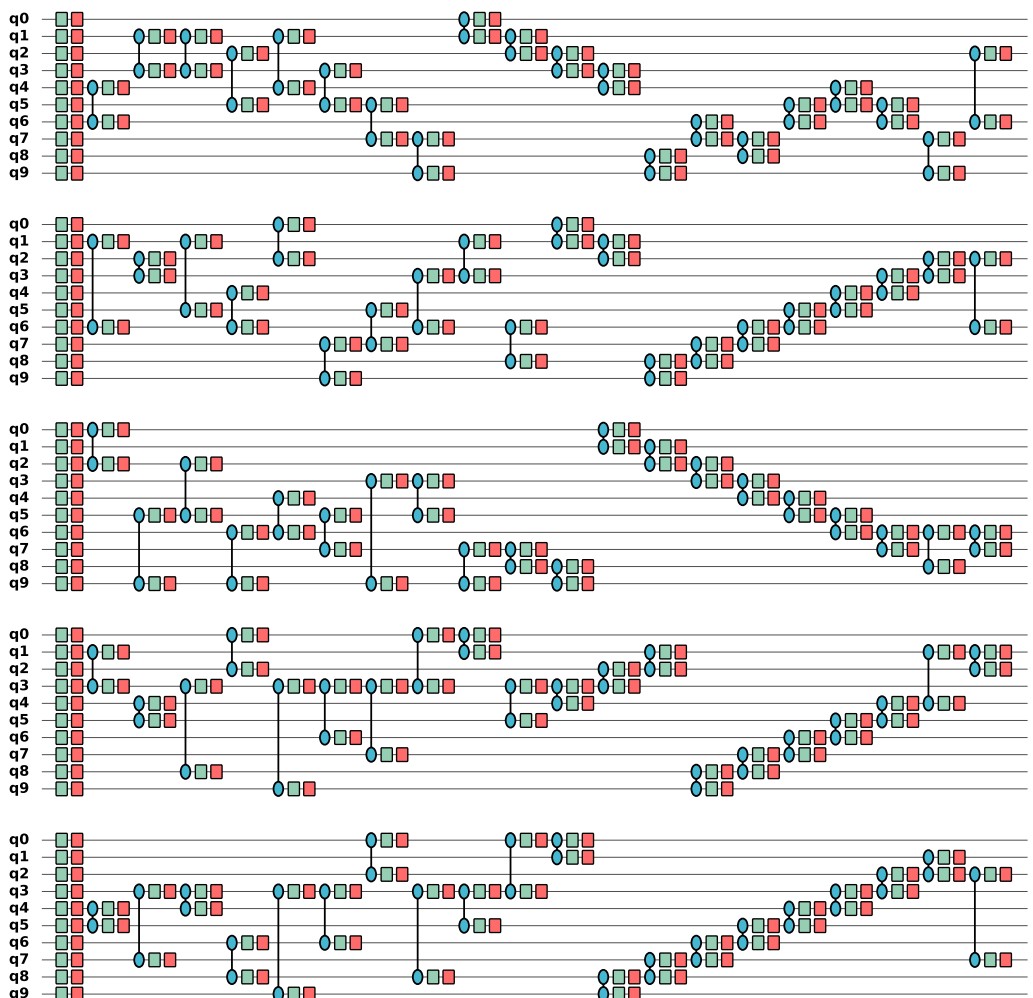

Figure 19: Loading circuits generated from AQER with $T = 20$ for the GS-TFIM dataset ($N = 10$). Green and red blocks denote $R_Y$ and $R_Z$ gates, respectively. Blue circles connected by lines denote $R_{ZZ}$ gates.

of forward evaluations of the whole circuit. Each such forward evaluation costs $\mathcal{O}(d(T + N))$, so the cost of computing one partial derivative is $\mathcal{O}(d(T + N))$. Since there are $P = \mathcal{O}(T + N)$ parameters, a full gradient evaluation costs $\mathcal{O}(d(T + N)^2)$. Performing $T_3$ gradient steps in Step III therefore leads to a total cost of

$$\mathcal{O}(d\,T_3(T + N)^2). \tag{75}$$

**Overall complexity for classical data.** Combining Eqs. (72), (73) and (75), the overall computational complexity of AQER in the classical-data setting is

$$\mathcal{O}\big(dTN^2 + dN + dT_3(T + N)^2\big), \tag{76}$$

which can be summarized as

$$\mathcal{O}\big(d\,T_3\,T^2\,\text{polylog}(d)\big). \tag{77}$$

This scaling is linear in the Hilbert-space dimension $d$ (as unavoidable for exact state-vector simulation), and polynomial in the Step I iteration count $T$ and the number of gradient steps $T_3$.

### G.2 QUANTUM DATA

We now consider the quantum data setting, where the target state $\rho$ is available on a quantum device, and no explicit classical description of $\rho$ is assumed. In this case, all RDMs, entropies and gradients

required by AQER are estimated from a finite number of measurement shots. For convenience, we denote by $M$ the number of shots used to estimate each expectation value or each entry of an RDM.

**Step I and Step II for quantum data.** As in the classical case, Step I adds $\mathcal{O}(T)$ gates that reduce the entanglement measure. At each iteration, the number of candidate gates is at most $N^2$. The cost per candidate is $\mathcal{O}(MT)$, and therefore the total cost of Step I in the quantum-data setting scales as

$$\mathcal{O}(MT^2N^2). \tag{78}$$

In Step II, AQER estimates $N$ single-qubit RDMs. This requires $\mathcal{O}(MN)$ shots and $\mathcal{O}(N)$ classical operations, so the complexity of Step II is

$$\mathcal{O}(MN). \tag{79}$$

**Step III: gradient-based parameter fine-tuning.** For gradient-based optimization with quantum data, we can use the parameter-shift rule to calculate the gradient. For each parameter, estimating one partial derivative costs $\mathcal{O}(M(T+N))$ operations. As before, the total number of parameters is $P = 5T + 2N = \mathcal{O}(T+N)$. Therefore, one full gradient evaluation in Step III requires

$$\mathcal{O}(M(T+N)^2) \tag{80}$$

operations. Performing $T_3$ gradient steps then yields the total complexity of Step III:

$$\mathcal{O}(MT_3(T+N)^2). \tag{81}$$

**Overall complexity for quantum data.** Combining Eqs. (78), (79) and (81), the overall computational complexity of AQER for quantum data is

$$\mathcal{O}(MT_3T^2\text{poly}(N)). \tag{82}$$

## H THE EFFICIENT LOADING OF IQP CIRCUIT STATES VIA AQER

In this section, we analyze the performance of AQER on classical vectors or quantum states arising from Instantaneous Quantum Polynomial (IQP) circuits. We first show in Theorem H.1 that, when the IQP angles lie on a known discrete grid and we have classical access to the full state vector, AQER can exactly reconstruct a loading circuit in at most $|E|$ iterations of Step I. We then relax the discrete-grid assumption and prove in Theorem H.2 that for generic IQP states with i.i.d. continuous angles, AQER can, with high probability, find a loading circuit that reduces the entanglement measure $\mathcal{S}$ to at most $\varepsilon$ using at most $|E|$ iterations of Step I in AQER. Finally, Theorem H.3 establishes an analogous exact-loading guarantee for a discrete IQP state family with fixed angle $\pi/8$ and unknown interaction graph, where we only have quantum access to the IQP state. Let $D$ be the degree of the graph $E$. In this case, AQER still recovers an exact loading circuit with probability at least $1 - \delta$ using at most $|E|$ iterations of Step I and a total quantum access number that is polynomial in $N$, $\log(1/\delta)$, $2^D$, and $|E|$.

**Theorem H.1** (Exact loading of discrete IQP state vectors). *Let $N \geq 2$ be the number of qubits, and let $K \in \mathbb{N}$ be a known grid size. We consider a fully-connected IQP circuit state*

$$|v(\boldsymbol{\omega})\rangle = H^{\otimes N} \left( \prod_{\{i,j\} \in E} e^{-i\omega_{ij}Z_iZ_j/2} \right) H^{\otimes N} |0^N\rangle, \tag{83}$$

*where $E$ is the unknown interaction graph of the IQP circuit, and each parameter is unknown and is restricted to the discrete grid*

$$\omega_{ij} = \frac{k_{ij}\pi}{2K+1}, \qquad k_{ij} \in \{-2K, -2K+1, \ldots, 2K-1, 2K\}. \tag{84}$$

*Assume we have classical access to the state vector of $|v(\boldsymbol{\omega})\rangle$. Then, up to an overall global phase, the AQER algorithm can reconstruct an exact loading circuit for the IQP state using $T$ iterations of Step I, where $T \leq |E|$.*

*Proof.* For the case where the target state is known to be an IQP state, we restrict the optimization in Step I of AQER as follows. For convenience, let $\mathcal{I}_t \subseteq [N]$ denote the set of qubits involved in the new two-qubit gate block added at the $t$-th iteration of Step I, and define the cumulative set $\mathcal{J}_t := \bigcup_{t'=1}^{t} \mathcal{I}_{t'}$. At iteration $t+1$, consider a candidate two-qubit gate block acting on a pair $\{i,j\}$. For each qubit $q \in \{i,j\}$, if $q \in \mathcal{J}_t$, we restrict the corresponding single-qubit part of the block to be the identity via $R_Z(0)R_Y(0) = I$. If $q \notin \mathcal{J}_t$, we set the single-qubit part to be $R_Z(\pi)R_Y(-\pi/2) = -iH$, which implements a Hadamard up to a global phase. For the entangling gate $R_{ZZ}$ on $\{i,j\}$, we restrict its rotation angle to the grid $\alpha_{ij} = a_{ij}\pi/(2K+1)$ with $a_{ij} \in \{-2K, -2K+1, \ldots, 2K-1, 2K\}$. Therefore, the optimization of the entanglement measure is performed by a grid search over the choice of the qubit pair $\{i,j\}$ and the discrete values of $\alpha_{ij}$.

Since the entanglement measure $\mathcal{S}$ is defined as the sum of single-qubit Renyi entropies, it is invariant under arbitrary single-qubit unitaries. Under the optimization rule for Step I described above, optimizing $\mathcal{S}$ for the target IQP state is therefore equivalent to optimizing $\mathcal{S}$ for the residual IQP state $|v'(\boldsymbol{\omega})\rangle$ defined as follows,

$$|v'(\boldsymbol{\omega})\rangle := \left( \prod_{\{i,j\}\in E} e^{-i\omega_{ij}Z_iZ_j/2} \right) H^{\otimes N} |0^N\rangle = \left( \prod_{\{i,j\}\in E} e^{-i\omega_{ij}Z_iZ_j/2} \right) |+\rangle^{\otimes N}, \qquad (85)$$

where $|+\rangle = H|0\rangle = (|0\rangle + |1\rangle)/\sqrt{2}$.

Next, we focus on the first iteration of Step I. Since $|E| > 0$, there exists a candidate gate block on qubits $\{p,q\} \in E$, and the optimization problem is

$$\arg\min_{\alpha_{pq}} \mathcal{S}\left( e^{-i\alpha_{pq}Z_pZ_q/2} |v'(\boldsymbol{\omega})\rangle \right)$$

$$= \arg\min_{\alpha_{pq}} \mathcal{S}\left( e^{-i\alpha_{pq}Z_pZ_q/2} \left( \prod_{\{i,j\}\in E} e^{-i\omega_{ij}Z_iZ_j/2} \right) |+\rangle^{\otimes N} \right)$$

$$= \arg\min_{\alpha_{pq}} \mathcal{S}\left( e^{-i\alpha_{pq}Z_pZ_q/2 - i\sum_{\{i,j\}\in E}\omega_{ij}Z_iZ_j/2} |+\rangle^{\otimes N} \right)$$

$$= \arg\min_{\alpha_{pq}} \mathcal{S}\left( e^{-i\sum_{\{i,j\}\in E}\beta_{ij}(\alpha_{pq})Z_iZ_j/2} |+\rangle^{\otimes N} \right), \qquad (86)$$

where we define the effective angles

$$\beta_{ij}(\alpha_{pq}) := \begin{cases} \omega_{ij}, & \{i,j\} \neq \{p,q\}, \\ \omega_{ij} + \alpha_{pq}, & \{i,j\} = \{p,q\}. \end{cases} \qquad (87)$$

The entanglement measure in Eq. (86) is defined in terms of single-qubit Renyi-2 entropies $\mathcal{S}(\rho) = \sum_{n=1}^{N} \mathcal{S}_n(\rho)$, which depend only on the single-qubit reduced density matrices (RDMs): $\mathcal{S}_n(\rho) = \mathcal{S}(\rho_n) = -\log_2 \text{Tr}[\rho_n^2]$, where $\rho_n$ is the RDM of an $N$-qubit state $\rho$ on the $n$-th qubit. Therefore, it suffices to compute the single-qubit RDMs of the state in Eq. (86). We remark that any single-qubit RDM has the Bloch decomposition $\rho_n = \frac{1}{2}(I + x_n X_n + y_n Y_n + z_n Z_n)$, where $x_n = \text{Tr}(\rho_n X_n)$, $y_n = \text{Tr}(\rho_n Y_n)$, and $z_n = \text{Tr}(\rho_n Z_n)$ are the expectation values of the corresponding Pauli operators on qubit $n$. Thus, computing the single-qubit RDMs reduces to evaluating $x_n, y_n$ and $z_n$. In particular, for the state in Eq. (86), we have

$$x_n(\alpha_{pq}) = \langle +|^{\otimes N} e^{i\sum_{\{i,j\}\in E}\beta_{ij}(\alpha_{pq})Z_iZ_j/2} X_n e^{-i\sum_{\{i,j\}\in E}\beta_{ij}(\alpha_{pq})Z_iZ_j/2} |+\rangle^{\otimes N}$$

$$= \langle +|^{\otimes N} e^{i\sum_{\{i,j\}\in \mathcal{N}(n)}\beta_{ij}(\alpha_{pq})Z_iZ_j/2} X_n e^{-i\sum_{\{i,j\}\in \mathcal{N}(n)}\beta_{ij}(\alpha_{pq})Z_iZ_j/2} |+\rangle^{\otimes N} \qquad (88)$$

$$= \prod_{\{i,j\}\in \mathcal{N}(n)} \cos\big(\beta_{ij}(\alpha_{pq})\big) \langle +|^{\otimes N} X_n |+\rangle^{\otimes N} \qquad (89)$$

$$= \prod_{\{i,j\}\in \mathcal{N}(n)} \cos\big(\beta_{ij}(\alpha_{pq})\big). \qquad (90)$$

In Eq. (88), we define $\mathcal{N}(n) \subseteq E$ as the set of edges incident on $n$, i.e., each $\{i,j\} \in \mathcal{N}(n)$ satisfies $n \in \{i,j\}$, and use the fact that for $\{i,j\} \notin \mathcal{N}(n)$ the operator $Z_iZ_j$ commutes with $X_n$ and hence

cancels in the conjugation without affecting the expectation value. Eq. (89) then follows from the Heisenberg-picture identity for a single incident edge, for example $e^{i\beta_{mn}Z_mZ_n/2}X_n e^{-i\beta_{mn}Z_mZ_n/2} = X_n \cos\beta_{mn} - Z_mY_n \sin\beta_{mn}$, together with the fact that any term containing $Z_m$ has zero expectation on $|+\rangle^{\otimes N}$ because $\langle+|Z|+\rangle = 0$. Eq. (90) then follows from $\langle+|I|+\rangle = \langle+|X|+\rangle = 1$. A similar analysis applied to $Y_n$ and $Z_n$, which yields $y_n = z_n = 0$.

The entanglement measure is therefore a function of $\alpha_{pq}$ and takes the form

$$S(\alpha_{pq}) = -\sum_{n=1}^{N} \log_2\left(\mathrm{Tr}[\rho_n(\alpha_{pq})^2]\right) = -\sum_{n=1}^{N} \log_2\left(\frac{1 + x_n(\alpha_{pq})^2}{2}\right), \tag{91}$$

where $x_n(\alpha_{pq})$ is given by Eq. (90). By construction, the effective angles $\beta_{ij}(\alpha_{pq})$ in Eq. (87) depend on $\alpha_{pq}$ only when $\{i,j\} = \{p,q\}$. Consequently, only $x_p(\alpha_{pq})$ and $x_q(\alpha_{pq})$ carry the dependence on $\alpha_{pq}$:

$$x_p(\alpha_{pq}) = \cos\left(\beta_{pq}(\alpha_{pq})\right) C_p, \qquad C_p := \prod_{\{p,m\}\in\mathcal{N}(p)\setminus\{\{p,q\}\}} \cos\beta_{pm},$$

$$x_q(\alpha_{pq}) = \cos\left(\beta_{pq}(\alpha_{pq})\right) C_q, \qquad C_q := \prod_{\{q,m\}\in\mathcal{N}(q)\setminus\{\{p,q\}\}} \cos\beta_{qm},$$

where $C_p$ and $C_q$ are constants with respect to $\alpha_{pq}$ and $C_p, C_q \neq 0$ by the choice of the angle grid. Plugging these expressions into Eq. (91) and grouping the constant terms, we can write

$$S(\alpha_{pq}) = \mathrm{const} - \log_2\left(\frac{1 + C_p^2 \cos^2\beta_{pq}(\alpha_{pq})}{2}\right) - \log_2\left(\frac{1 + C_q^2 \cos^2\beta_{pq}(\alpha_{pq})}{2}\right),$$

so that minimizing $S(\alpha_{pq})$ over the grid $\alpha_{pq} \in \{a_{pq}\pi/(2K+1)\}$ is equivalent to minimizing the function

$$g(\alpha_{pq}) := -\log_2\left(\frac{1 + C_p^2 \cos^2\beta_{pq}(\alpha_{pq})}{2}\right) - \log_2\left(\frac{1 + C_q^2 \cos^2\beta_{pq}(\alpha_{pq})}{2}\right). \tag{92}$$

It is easy to verify that $\alpha_{pq}^* = -\omega_{pq}$, i.e. $\beta_{pq}(\alpha_{pq}^*) = \omega_{pq} + \alpha_{pq}^* = 0$ is the unique optimal value of the above objective, and it can be found by a grid search over $\alpha_{ij} = a_{ij}\pi/(2K+1)$ with $a_{ij} \in \{-2K, -2K+1, \ldots, 2K-1, 2K\}$.

Having identified, for a fixed edge $\{p,q\} \in E$, the unique optimal choice $\alpha_{pq}^* = -\omega_{pq}$ such that $\beta_{pq}(\alpha_{pq}^*) = \omega_{pq} + \alpha_{pq}^* = 0$, we now iterate this construction. At the $t$-th iteration of Step I, let $\beta_{ij}^{(t)}$ denote the effective angles on the IQP edges and define the residual edge set

$$F^{(t)} := \{\{i,j\} \in E : \beta_{ij}^{(t)} \neq 0\}. \tag{93}$$

By the discussion above, if $F^{(t)}$ is nonempty we can pick any $\{p,q\} \in F^{(t)}$ and minimize $S$ over the grid of angles $\alpha_{pq}$. Uniqueness of the minimizer implies that the optimal choice $\alpha_{pq}^{(t)}$ satisfies

$$\beta_{pq}^{(t+1)} = \beta_{pq}^{(t)} + \alpha_{pq}^{(t)} = 0, \tag{94}$$

while all other angles remain unchanged, i.e., $\beta_{ij}^{(t+1)} = \beta_{ij}^{(t)}$ for all $\{i,j\} \neq \{p,q\}$. Therefore the residual edge set strictly shrinks at each iteration,

$$|F^{(t+1)}| = |F^{(t)}| - 1. \tag{95}$$

Since $|F^{(0)}| \leq |E|$, after $T = |F^{(0)}| \leq |E|$ iterations we reach a configuration with $F^{(T)} = \varnothing$, i.e., $\beta_{ij}^{(T)} = 0$ for all $\{i,j\} \in E$. In this case all two-qubit entangling rotations are cancelled, and the resulting state has vanishing entanglement measure $S = 0$. By definition of $S$ as the sum of single-qubit Renyi entropies, this implies that the state obtained from $|v(\boldsymbol{\omega})\rangle$ after $T$ iterations of Step I is a product state, which we denote by $|v_{\mathrm{prod}}\rangle$.

Finally, by Lemma B.2, we can construct a single-qubit circuit composed of $R_Z$ and $R_Y$ rotations that maps $|v_{\mathrm{prod}}\rangle$ exactly to the computational basis state $|0^N\rangle$ up to a global phase. Let $U_{\mathrm{I}}$ be the product of all Step I gate blocks obtained by the above procedure and $U_{\mathrm{II}}$ be the single-qubit circuit from Lemma B.2. Then

$$U_{\mathrm{II}}U_{\mathrm{I}}|v(\boldsymbol{\omega})\rangle = |0^N\rangle \tag{96}$$

up to a global phase. Taking adjoints and reversing the order of gates, we obtain an explicit loading circuit

$$U_{\text{load}} = (U_{\text{II}}U_{\text{I}})^{\dagger} = U_{\text{I}}^{\dagger}U_{\text{II}}^{\dagger} \tag{97}$$

such that $U_{\text{load}}|0^N\rangle = |v(\boldsymbol{\omega})\rangle$ up to a global phase. This completes the proof of Theorem H.1.

$\square$

**Theorem H.2** (Approximate loading for generic IQP state vectors)**.** *We follow the notations in Theorem H.1 and consider the IQP circuit state*

$$|v(\boldsymbol{\omega})\rangle = H^{\otimes N} \left( \prod_{\{i,j\}\in E} e^{-i\omega_{ij}Z_iZ_j/2} \right) H^{\otimes N}|0^N\rangle, \tag{98}$$

*where the angles $\{\omega_{ij}\}_{\{i,j\}\in E}$ are i.i.d. random variables sampled from $[-\pi, \pi]$ uniformly. Let $D$ be the maximum degree of $E$. We assume the classical access to the state vector of $|v(\boldsymbol{\omega})\rangle$. Then for any $\delta, \epsilon > 0$, AQER uses at most $T \leq |E|$ iterations of Step I by searching the grid with size $K = \left\lceil \frac{\pi}{2}\sqrt{\frac{DN}{\epsilon}} \right\rceil$ to produce a circuit $U_{\text{I}}$ satisfying $\mathcal{S}(U_{\text{I}}|v(\boldsymbol{\omega})\rangle) \leq \varepsilon$ with probability at least $1 - \delta$.*

*Proof.* First, we demonstrate that with high probability all parameters $\omega_{ij}$ satisfy a lower bound on the cosine value. Since each $\omega_{ij}$ is sampled from $[-\pi, \pi]$ uniformly, we have

$$\Pr\left[ |\cos\omega_{ij}| \geq \frac{\delta}{|E|} \right] = 1 - \frac{2}{\pi}\arcsin\left(\frac{\delta}{|E|}\right) \geq 1 - \frac{\delta}{|E|}. \tag{99}$$

By a union bound over all $\{i,j\} \in E$, it follows that with probability at least $1 - \frac{\delta}{|E|} \cdot |E| = 1 - \delta$ we have

$$|\cos\omega_{ij}| \geq \frac{\delta}{|E|} \quad \text{for all } \{i,j\} \in E. \tag{100}$$

In the following, we condition on this high-probability event.

Next, we follow the same optimization procedure of Step I as in the proof of Theorem H.1. Specifically, the two-qubit gate on $\{i,j\}$ has angle $\alpha_{ij}$ restricted to the grid

$$\mathcal{G}_K := \left\{ \frac{a\pi}{2K+1} : a \in \{-2K, -2K+1, \ldots, 2K-1, 2K\} \right\},$$

where $K = \left\lceil \frac{\pi}{2}\sqrt{\frac{DN}{\epsilon}} \right\rceil$. We also impose that at most one two-qubit gate block is added for each pair $\{i,j\} \subseteq [N]$. Under this restriction, Step I has at most one update per edge, so the total number of iterations satisfies $T \leq |E|$. For each $\{p,q\} \in E$, the grid search in the optimization problem of Eq. (92) finds a value $\alpha_{pq}^{\star}$ whose distance to $-\omega_{pq}$ is at most $\pi/(2K+1)$. Thus, after $T = |E|$ iterations, the remaining residual state can be written as

$$|v(\boldsymbol{\beta})\rangle = e^{-i\sum_{\{i,j\}\in E}\beta_{ij}Z_iZ_j/2}|+\rangle^{\otimes N}, \tag{101}$$

where each effective angle satisfies $|\beta_{ij}| \leq \pi/(2K+1)$.

The entanglement measure of the state $|v(\boldsymbol{\beta})\rangle$ can be obtained via a calculation analogous to Eqs. (88)–(91). In particular,

$$\mathcal{S}(|v(\boldsymbol{\beta})\rangle) = -\sum_{n=1}^{N} \log_2\left(\frac{1+x_n^2}{2}\right)$$

$$\leq \sum_{n=1}^{N}(1-x_n^2) \tag{102}$$

$$= N - \sum_{n=1}^{N} \prod_{\{i,j\}\in\mathcal{N}(n)} \cos^2\beta_{ij} \tag{103}$$

$$\leq N - N\cos^{2D}\left(\frac{\pi}{2K+1}\right) \tag{104}$$

$$\leq N - N \left[ 1 - \left( \frac{\pi}{2K+1} \right)^2 \right]^D \tag{105}$$

$$\leq N - N \left[ 1 - D \left( \frac{\pi}{2K+1} \right)^2 \right] \tag{106}$$

$$= DN \left( \frac{\pi}{2K+1} \right)^2 \leq \epsilon. \tag{107}$$

Here Eq. (102) follows from the inequality $-\log_2 \frac{2-x}{2} \leq x$ for $x \in [0,1]$ by setting $x = 1 - x_n^2$. Eq. (103) follows from Eq. (90), which gives $x_n^2 = \prod_{\{i,j\} \in \mathcal{N}(n)} \cos^2 \beta_{ij}$. Eq. (104) is derived using $|\beta_{ij}| \leq \pi/(2K+1)$ and $|\mathcal{N}(n)| \leq D$, so that each product is lower bounded by $\cos^{2D}(\pi/(2K+1))$. Eq. (105) uses $\cos^2 x = 1 - \sin^2 x \geq 1 - x^2$ for $|x| \leq \pi/2$. Eq. (106) is obtained from the Bernoulli inequality $(1-x)^D \geq 1 - Dx$ for $x \in [0,1]$. Finally, Eq. (107) follows from the choice $K = \left\lceil \frac{\pi}{2} \sqrt{\frac{DN}{\epsilon}} \right\rceil$, which ensures $DN \left( \frac{\pi}{2K+1} \right)^2 \leq \epsilon$. This completes the proof of Theorem H.2.

$\square$

**Theorem H.3** (Exact loading of discrete IQP states). *Let $N \geq 2$ be the number of qubits. We consider a fully-connected IQP circuit state*

$$|v(E)\rangle = H^{\otimes N} \left( \prod_{\{i,j\} \in E} e^{-i\pi Z_i Z_j / 8} \right) H^{\otimes N} |0^N\rangle, \tag{108}$$

*where $E$ is the unknown interaction graph of the IQP circuit, with the maximum degree $D$. Assume we have quantum access to the state $|v(\boldsymbol{\omega})\rangle$. Then, up to an overall global phase, the AQER algorithm can reconstruct an exact loading circuit for the IQP state using $T$ iterations of Step I, where $T \leq |E|$ with probability at least $1 - \delta$ by using at most $\mathcal{O}\left( |E| N^2 2^D \log \frac{N^2 |E|}{\delta} \right)$ calls to the given IQP state.*

*Proof.* We follow the Step I optimization as in Theorem H.1, except that we restrict the grid search for each pair $\{p,q\}$ to the two angles $\alpha_{pq} \in \{0, -\pi/4\}$. We remark that we now only have quantum (rather than classical vector) access to the IQP state. Our goal is to guarantee that, in each iteration, the grid search selects a qubit pair in $E$ with probability at least $1 - \delta/|E|$, where the failure probability comes solely from the finite number of measurement shots. Consequently, after $T = |E|$ iterations of Step I, all qubit pairs in $E$ are cancelled with probability at least $1 - \delta$.

Next, we focus on the first iteration. Similar to the proof of Theorem H.1, the residual state can be written as
$$|v'(E)\rangle = e^{-i \sum_{\{i,j\} \in E} \beta_{ij}(\alpha_{pq}) Z_i Z_j / 2} |+\rangle^{\otimes N}. \tag{109}$$
For any candidate pair $\{p,q\}$ and any choice of $\alpha_{pq}$, the single-qubit Bloch coefficients satisfy

$$x_n(\alpha_{pq}) = \prod_{\{i,j\} \in \mathcal{N}(n)} \cos(\beta_{ij}(\alpha_{pq})), \qquad y_n(\alpha_{pq}) = 0, \qquad z_n(\alpha_{pq}) = 0.$$

In Theorem H.1 we showed that Step I optimizes the entanglement measure

$$\mathcal{S}(\alpha_{pq}) = -\sum_{n=1}^{N} \log_2 \left( \frac{1 + x_n(\alpha_{pq})^2}{2} \right), \tag{110}$$

and that a decrease in $\mathcal{S}$ is achieved via an increase in $|x_n(\alpha_{pq})|$. Since $|\mathcal{N}(n)| \leq D$ and $\beta_{ij}(\alpha_{pq}) \in \{0, \pm\pi/4\}$, we have

$$x_n(\alpha_{pq}) \in \left\{ \left( \frac{\sqrt{2}}{2} \right)^d : 0 \leq d \leq D \right\} \subseteq [2^{-D/2}, 1].$$

Thus, for each $\{p,q\} \subseteq [N]$, by using

$$\mathcal{O}\left( 2^D \log \frac{N^2 |E|}{\delta} \right)$$

quantum shots, we can estimate the relevant $x_p(\alpha_{pq})$ and $x_q(\alpha_{pq})$ with additive error smaller than a constant multiple of $2^{-D/2}$, and hence determine whether both $|x_p|$ and $|x_q|$ increase when switching $\alpha_{pq}$ from 0 to $-\pi/4$ with probability at least $1 - \delta/(N^2|E|)$ if $\{p,q\} \in E$. By going over all $\{p,q\} \subseteq [N]$, we can then identify at least one $\{p,q\} \in E$ with probability at least $1 - \delta/|E|$. Repeating this procedure for all $T = |E|$ iterations, a union bound implies that the overall success probability is at least $1 - \delta$. The total number of quantum shots is

$$\mathcal{O}\left(2^D \log \frac{N^2|E|}{\delta}\right) \cdot N^2 \cdot |E| = \mathcal{O}\left(|E|N^2 2^D \log \frac{N^2|E|}{\delta}\right).$$

This completes the proof of Theorem H.3.

$\square$

## I  THE USE OF LARGE LANGUAGE MODELS

In this work, large language models (LLM) were used only as a general-purpose tool for minor language polishing and expression refinement. The LLM did not contribute to any scientific ideas, experiments, or results, and all content generated under its assistance was carefully reviewed and edited by the authors. The authors take full responsibility for all contents of the paper, including those influenced by LLM-assisted text.

