# OpenReview forum: "AQER: A Scalable and Efficient Data Loader for Digital Quantum Computers"
_ICLR.cc/2026/Conference — ICLR 2026 Poster_

### Official Review · Reviewer_eQiP · 2025-10-27

**Soundness:** 2
**Presentation:** 3
**Contribution:** 2
**Rating:** 4
**Confidence:** 4

**Summary:**

AQER relates approximate-state-loading error to entanglement entropy S and derives corresponding information-theoretic bounds

**Strengths:**

1.	Provides information-theoretic bounds that relate approximation error directly to a locally measurable entanglement proxy, guiding circuit construction for state preparation.
2.	Consistently outperforms MPS, HEC and AQCE at equal two-qubit gate budgets on both classical and quantum data sets.

**Weaknesses:**

1.	Theorem 3.1 is general, but the evaluated states appear to be low- or modestly entangled; tougher, higher-entanglement regimes are not covered.
2.	No scaling analysis of gradient variance with qubit number; trainability claim remains heuristic.
3.	The paper claims code availability but does not provide the anonymous GitHub link in the main text, which could hinder reproducibility efforts.

**Questions:**

1.	Could 2-D random circuits or critical 2-D spin systems be included to probe performance when entanglement is less easily reduced?
2.	Could the authors provide gradient-magnitude statistics across N=10–50 qubits?

---

> ### Author Response · Authors · 2025-11-22
> **Author Response to Reviewer eQiP (1)**
>
> **Q1.** Could 2-D random circuits or critical 2-D spin systems be included to probe performance when entanglement is less easily reduced?
>
> **Response to Q1.** We thank the reviewer for this valuable suggestion. In the revision, we have added experiments on 2D models, including (1) random circuit states and (2) ground states of 2D XXZ (Heisenberg) models at the critical points.
>
> *1. For 2D random circuit states, AQER effectively reduces both the entanglement measure and the infidelity as $T$, the number of iterations in Step I of AQER increases. 2D random circuit states are generated by shallow nearest-neighbor CZ circuits with random single-qubit rotations on a rectangular lattice.* In particular, the 2D random circuit state is defined as the output of a depth-4 circuit composed of alternating single-qubit and CZ layers:
>
> - In each single-qubit layer, every qubit is acted on by a randomly chosen rotation from $\{R_X, R_Y, R_Z\}$ with uniformly sampled parameters.
> - In each CZ layer, CZ gates are applied to nearest neighbors along horizontal or vertical directions according to one of four distinct tilings of the 2D grid (two horizontal and two vertical patterns).
>
> These four CZ patterns are cycled over the four layers so that, across the depth-4 circuit, all nearest-neighbor couplings are activated.
>
> In the experiment, we consider $4\times 4$, $4\times 5$, and $5\times 5$ structures. The entanglement measure $\mathcal{S}$ with different $T$ values is given in the table below.
>
> | Size\T | 10 | 20 | 40 | 60 | 80 | 100 |
> | --- | --- | --- | --- | --- | --- | --- |
> | 4x4 | 1.6 | 0.34 | 0.19 | 0.21 | 0.23 | 0.22 |
> | 4x5 | 4.0 | 0.99 | 0.35 | 0.30 | 0.23 | 0.28 |
> | 5x5 | 6.7 | 2.9 | 1.3 | 0.81 | 0.71 | 0.62 |
>
> For all cases, the entanglement measure decreases by about one order of magnitude when $T$ increases from 10 to 100. The corresponding infidelities behave as:
>
> | Size\T | 10 | 20 | 40 | 60 | 80 | 100 |
> | --- | --- | --- | --- | --- | --- | --- |
> | 4x4 | 0.18 | 0.029 | 0.013 | 0.014 | 0.014 | 0.014 |
> | 4x5 | 0.46 | 0.11 | 0.025 | 0.022 | 0.019 | 0.019 |
> | 5x5 | 0.66 | 0.31 | 0.14 | 0.088 | 0.066 | 0.051 |
>
> As shown in the table above, AQER consistently reduces the infidelity as $T$ increases. For example, for the $4\times 4$ case, the infidelity drops from 0.18 at $T=10$ to 0.014 at $T=100$; for the $5\times 5$ structure, it decreases from 0.66 to 0.051. Thus, AQER achieves substantial improvements in both $\mathcal{S}$ and infidelity.
>
> *2. For 2D XXZ ground states at the critical point, AQER reduces both the entanglement measure and the infidelity effectively as $T$ increases.* In particular, we consider the spin-$\frac{1}{2}$ XXZ model on rectangular lattices, with Hamiltonian
> $H_{\mathrm{XXZ}} \;=\; \sum_{\langle i,j\rangle}
> \Bigl[
> J_{xy} \bigl( \sigma_i^x \sigma_j^x + \sigma_i^y \sigma_j^y \bigr) +J_z \,\sigma_i^z \sigma_j^z
> \Bigr]$,
>
> where $\langle i,j\rangle$ runs over nearest-neighbor pairs on the 2D grid and $\sigma^{x,y,z}$ denote Pauli matrices. We focus on the critical point $J_{xy} = J_z$. The entanglement measure $\mathcal{S}$ with different $T$ values is given in the table below.
>
> | Size\T | 10 | 20 | 40 | 60 | 80 | 100 |
> | --- | --- | --- | --- | --- | --- | --- |
> | 4x4 | 8.7 | 6.7 | 4.2 | 3.9 | 3.7 | 3.5 |
> | 4x5 | 12 | 9.6 | 7.0 | 5.9 | 5.4 | 5.1 |
> | 5x5 | 18 | 14 | 7.7 | 6.8 | 6.6 | 6.5 |
>
> For cases, the entanglement measure decreases steadily with increased $T$, typically by a factor of about 2-3 when $T$ increases from 10 to 100 (e.g., from 8.7 to 3.5 on $4\times 4$, and from 18 to 6.5 on $5\times 5$). The corresponding infidelity behave as:
>
> | Size\T | 10 | 20 | 40 | 60 | 80 | 100 |
> | --- | --- | --- | --- | --- | --- | --- |
> | 4x4 | 0.78 | 0.57 | 0.32 | 0.25 | 0.20 | 0.19 |
> | 4x5 | 0.89 | 0.77 | 0.56 | 0.37 | 0.33 | 0.28 |
> | 5x5 | 0.93 | 0.87 | 0.53 | 0.42 | 0.40 | 0.36 |
>
> Similar to the random circuit experiments, AQER consistently reduces the infidelity as $T$ increases.

---

> ### Author Response · Authors · 2025-11-22
> **Author Response to Reviewer eQiP (2)**
>
> **Q2&W2.** No scaling analysis of gradient variance with qubit number; trainability claim remains heuristic. Could the authors provide gradient-magnitude statistics across N=10–50 qubits?
>
> **Response to Q2&W2.** We thank the reviewer for this suggestion and address the question in two parts.
>
> *1. The gradient norm remains large as the number of qubits increases.* We have conducted the gradient-magnitude statistics for the TFIM model with system sizes of $N = 10,20,30,40,50$ qubits:
>
> | N | 10 | 20 | 30 | 40 | 50 |
> | --- | --- | --- | --- | --- | --- |
> | T=10 | 0.080 | 0.029 | 0.024 | 0.026 | 0.047 |
> | T=20 | 0.063 | 0.061 | 0.034 | 0.032 | 0.045 |
> | T=40 | 0.052 | 0.13 | 0.086 | 0.051 | 0.054 |
> | T=60 | 0.052 | 0.12 | 0.12 | 0.094 | 0.090 |
> | T=100 | 0.057 | 0.12 | 0.14 | 0.13 | 0.13 |
>
> As shown in the table, the initial gradient norms remain greater than 0.02 across all system sizes and **do not exhibit an exponential decay with $N$**. This behavior provides numerical evidence that barren plateau effects are mitigated in our experiments. The result is also shown in Figure. 10 in Appendix F in the revision.
>
> *2. The trainability claim is supported by the optimization curve statistics.* In fact, complementary numerical evidence supporting the trainability claim for the AQER Step III was already presented in the original submission. As shown in Figure 4(a) in the main text, the optimization curves for $N=50$ qubits display a clear and steady decrease of the loss throughout training. This effective loss reduction is consistent with the newly added gradient statistics and jointly supports the conclusion that our method does not suffer from barren plateaus in the considered large-qubit regime.

---

> ### Author Response · Authors · 2025-11-22
> **Author Response to Reviewer eQiP (3)**
>
> **W1.** Theorem 3.1 is general, but the evaluated states appear to be low- or modestly entangled; tougher, higher-entanglement regimes are not covered.
>
> **Response to W1.** We thank the reviewer for this comment and address the concern in two aspects.
>
> *1. Neither Theorem 3.1 nor AQER is limited to low- or modestly entangled cases.* We thank the reviewer for highlighting the generality of Theorem 3.1. In particular, Theorem 3.1 shows that, given a quantum circuit that reduces the entanglement measure of the target state into the small-S regime, then the infidelity of AQL scales linearly with S. This is precisely why Step I of AQER is designed to explicitly reduce the entanglement measure. At the same time, **this does not mean that AQER only works well for low-entangled target states**. Intuitively, if the entanglement measure $\mathcal{S}$ has large initial values but can be driven down quickly with a moderate number of two-qubit gates, then Theorem 3.1 provides an efficient theoretical guarantee on the resulting approximation error of AQER.
>
> *2. For certain highly entangled target states, AQER is efficient.* To illustrate this, we have added experiments for the GHZ state in the revision. The $N$-qubit GHZ state is highly entangled in the sense that each single-qubit marginal is maximally mixed and the initial $\mathcal{S}=N$. Nevertheless, AQER can still reduce its entanglement efficiently. For $N=10$, the entanglement measure $\mathcal{S}$ decays quickly and drops below 0.03 after only 9 iterations. Correspondingly, the infidelity is reduced to lower than 2e-4 with only $T=9$ two-qubit gates:
>
> | $T$ | 1 | 2 | 3 | 4 | 5 | 6 | 7 | 8 | 9 |
> | --- | --- | --- | --- | --- | --- | --- | --- | --- | --- |
> | $\mathcal{S}$ | 9.0 | 8.0 | 7.0 | 6.0 | 5.0 | 4.0 | 3.0 | 2.0 | 0.025 |
> | infidelity | 0.5 | 0.5 | 0.5 | 0.5 | 0.5 | 0.5 | 0.5 | 0.5 | 1.2e-4 |
>
> The value of $\mathcal{S}$ and the infidelity with increasing $T$ is also shown in Figure 12 in Appendix F in the revision. This example demonstrates that **a state can be highly entangled while still being structurally simple such that AQER can achieve low infidelity efficiently.**
>
> **W3.** The paper claims code availability but does not provide the anonymous GitHub link in the main text, which could hinder reproducibility efforts.
>
> **Response to W3.** We thank the reviewer for pointing this out. In the original submission, the code is already available at an anonymous GitHub repository, and the hyperlink is attached to the word "GitHub" in line 107 in the Introduction in the original version. However, the hyperlink may be overlooked since it is not highlighted. In the revision, we have highlighted this hyperlink in blue so that readers can easily notice and access the code.

---

> ### Author Response · Authors · 2025-11-26
> **Comment for Reviewer eQiP**
>
> Dear Reviewer eQiP:
>
> Thanks a lot for your helpful and thoughtful review of our paper! We have tried our best to address the mentioned concerns and have carefully revised the manuscript. Are there unclear explanations or remaining problems? We would be very grateful for further feedback and will do our best to address them.
>
> Best regards,
>
> The authors

---

> > ### Comment · Reviewer_eQiP · 2025-11-27
> > **Reply to authors**
> >
> > I appreciate the authors’ timely response. However, after examining the supplementary numerical results and the main theorem, I find that the proposed method continues to lack rigorous theoretical guarantees regarding scalability. Therefore, I will maintain my current score.

---

> ### Author Response · Authors · 2025-12-03
> **Summary of Reviewer eQiP‘s Official Review and Our Response**
>
> First, we would like to thank the reviewer for investing substantial time in reviewing our work and for the constructive comments that helped improve the paper. In the original official review, the reviewer summarized two main strengths:
>
> (1) **AQER** **provides information-theoretic bounds** that relate approximation error directly to a locally measurable entanglement proxy, **guiding circuit construction** for state preparation;
>
> (2) AQER consistently outperforms MPS, HEC, and AQCE at equal two-qubit gate budgets on both classical and quantum data sets.
>
> The reviewer also pointed out several weaknesses and questions that helped us further strengthen the paper:
>
> (W1/Q1) the evaluated states in the original submission appeared to be low- or modestly entangled, and tougher, higher-entanglement regimes were not covered;
>
> (W2/Q2) there was no scaling analysis or gradient statistics with qubit number, so the trainability claim remained heuristic. We addressed these points as follows.
>
> For W1/Q1, we added new experiments on **2D random circuits** and **2D XXZ (Heisenberg) ground states at the critical point**, showing that as the number of AQER iterations $T$ increases, the entanglement measure $\mathcal{S}$ and the infidelity both systematically decrease across different lattice sizes, demonstrating effectiveness in more challenging, higher-entanglement regimes. We also added a **GHZ-state example**, where the initial entanglement measure is $\mathcal{S}=N$ (maximally entangled single-qubit marginals), and showed that AQER can drive $\mathcal{S}$ close to zero and reach infidelity $\sim 10^{-4}$ with only $T=9$ two-qubit gates, illustrating that certain highly entangled yet structured states can still be handled efficiently.
>
> For W2/Q2, we provided **gradient-magnitude statistics** for the TFIM benchmark at $N=10,20,30,40,50$, showing that the initial gradient norms remain above 0.02 and do not decay exponentially with $N$. This offers numerical evidence that Step III does not suffer from barren plateaus in the considered regime.

---

> ### Author Response · Authors · 2025-12-03
> **Response to the newly raised comments**
>
> Here, we briefly clarify our theoretical guarantees on scalability in three parts in light of reviewer eQiP’s final comment that the method “continues to lack rigorous theoretical guarantees regarding scalability.”
>
> *1. What our main theorem guarantees.* Theorem 3.1 provides a general, non-asymptotic upper bound that holds for all AQL methods, including AQER: if the entanglement measure $\mathcal{S}$ can be reduced below a target value $S$ in $T$ steps, then the resulting infidelity is bounded by two explicit functions $f _ 1(S)$ and $f _ 2 (S)$, both scaling as $\mathcal{O}(S)$ as $S \rightarrow 0$. This is a rigorous complexity statement: it reduces the question of scalability to a quantitative condition on **how fast $\mathcal{S}$ can be reduced**. In this framework, gradient norms *per se* are not the right object for certifying scalability in the small-$S$ regime: once $\mathcal{S}$ has been driven to a small value, Theorem 3.1 already implies that the infidelity is small, and the optimization has moved far away from the high-loss “barren plateau” region $1 - 2^{-\mathcal{O}(N)}$. As the infidelity approaches zero, it is natural and expected that the gradient norm also decreases, which is a signature of convergence rather than untrainability. What matters for scalability is therefore whether AQER can reduce $\mathcal{S}$ (and hence the infidelity) to a small target within a number of steps $T$ that scales at most polynomially in the problem size.
>
> *2. Explicit polynomial-time guarantees for nontrivial state families.* As mentioned in the second round response to reviewer Ki6d, we further strengthened the theory by identifying a broad and complexity-theoretically nontrivial family of target states, i.e., $N$-qubit IQP circuits with bounded degree and generic or fixed non-Clifford angles, for which we can **explicitly** bound the required number of AQER steps. In Appendix H we prove that, for these IQP families, AQER reduces the entanglement measure $\mathcal{S}$ to zero (or below any prescribed $\varepsilon>0$) in at most $T = |E|$ iterations of Step I, where $E$ is the interaction graph of the IQP circuit so $|E| \leq N(N-1)/2$. For the quantum data case, the total number of quantum calls scales as $\text{poly}(N,|E|,2^{D},\log(1/\delta))$ , where $D$ is the maximum degree of $E$. This provides a concrete polynomial (or quasi-polynomial) runtime bound for bounded or polylogarithmic $D$, for a class of states that is believed to be classically hard to sample from under standard assumptions [1].
>
> *3. Why stronger, fully general scalability guarantees are impossible in the AQL setting.* At the same time, existing complexity-theoretic results [2-5] show that one cannot hope for a universal polynomial-time guarantee over all moderately or highly entangled states. Generic volume-law states drawn from, e.g., deep random circuits typically require exponentially many gates to prepare and are believed to have output distributions that are average-case hard to learn or even approximate under standard assumptions. For such worst-case or average-case-hard families, no AQL method, not just AQER, can provably offer polynomial-time guarantees with a polynomial gate budget. In this sense, requiring a global polynomial-time scalability guarantee for arbitrary highly entangled target states is incompatible with what is known about the underlying computational complexity of the AQL problem.
>
> Taken together, we believe that the current version of the paper does provide rigorous and meaningful scalability guarantees at the level that is realistically achievable for this problem: (i) a general theorem that reduces scalability to a quantitative condition on entanglement reduction; (ii) explicit polynomial-time guarantees for a nontrivial, classically hard family of target states; and (iii) a complexity-theoretic explanation of why stronger, fully general guarantees cannot be expected for any AQL method.
>
> [1] Michael J. Bremner, Richard Jozsa, Dan J. Shepherd. Classical simulation of commuting quantum computations implies collapse of the polynomial hierarchy. Proc. A, 2011.
>
> [2] Patrick Hayden, Debbie W. Leung, Andreas Winter. Aspects of generic entanglement. Comm. Math. Phys. Vol. 265, No. 1, pp. 95-117, 2006.
>
> [3] Michał Oszmaniec, Marcin Kotowski, Michał Horodecki, Nicholas Hunter-Jones. Saturation and recurrence of quantum complexity in random local quantum dynamics. Phys. Rev. X 14, 041068, 2024.
>
> [4]. Haimeng Zhao, Laura Lewis, Ishaan Kannan, Yihui Quek, Hsin-Yuan Huang, Matthias C. Caro. Learning Quantum States and Unitaries of Bounded Gate Complexity. PRX Quantum 5, 040306. 2024.
>
> [5] Alexander Nietner, Marios Ioannou, Ryan Sweke, Richard Kueng, Jens Eisert, Marcel Hinsche, Jonas Haferkamp. On the average-case complexity of learning output distributions of quantum circuits. Quantum, 2025.

---

### Official Review · Reviewer_Ki6d · 2025-10-29

**Soundness:** 2
**Presentation:** 3
**Contribution:** 2
**Rating:** 4
**Confidence:** 3

**Summary:**

The paper introduces a unified framework for analyzing the accuracy of approximate quantum loaders (AQLs), by bounding the achievable infidelity in terms of an entanglement measure S (the sum of single-qubit Rényi-2 entropies after applying the inverse loader to the target state). Building on this, the authors propose AQER, a data loader that constructs an encoding circuit by first reducing the target state’s entanglement, then applying single-qubit product-state corrections with closed-form parameters, and finally refining all parameters jointly. Experiments on classical datasets (MNIST, CIFAR-10, SST-2) and quantum datasets (random quantum circuits and Ising model ground states) reaching up to 50 qubits, show lower infidelity than MPS-, HEC-, and AQCE-based baselines for the same or fewer two-qubit gates.

**Strengths:**

- The paper presents a clear, useful bound for all AQLs, formulated in terms of the entanglement measure S.


- The AQER algorithm is intentionally designed to minimize the S, which the theory connects directly to infidelity.


- Once target states are available, training AQER relies only on local measurements.


- Experiments provide encouraging evidence that supports the paper’s central claims

**Weaknesses:**

Although the above strength, I do not think the current version of the paper matches the ICLR criteria for the following reason:

- My main concern is the computational cost required to reach low infidelity in the general case. The theoretical bound in the paper is meaningful only in the limit S goes to 0. For target states with moderate or high entanglement, achieving a small S requires deeper circuits U, and it is unclear whether this can still be done with circuits of polynomial size.


- Also, the upper bound is stated in terms of the existence of a product state with certain properties. This does not guarantee that a generic product-state ansatz will satisfy the same bound in practice.

- Another concern I have is that computing the entanglement measure requires access to the target state. For quantum data, this may be possible if multiple copies of the state are available, though the number of required copies is not made explicit. For classical data, however, one must first classically construct the full target state, which is not tractable at large numbers of qubits. In fact, the classical-data experiments in the paper are limited to around 10 qubits.


- Even for quantum states, as far as I understand, the simulations proposed in the paper rely on tensor-network methods, which inherently assume limited entanglement. This suggests that highly entangled quantum data are effectively out of scope for the demonstrated results.


- Finally, it is not clear that the optimization described in Eq. (2) can be solved in polynomial time.

**Questions:**

- Can the authors specify what happens, in terms of the algorithm’s computational complexity, if the target states have a high level of entanglement?


- Can the authors provide theoretical guarantees that the optimization step in Eq. (2) is efficient?


- How is the method applicable to high-dimensional classical data that require a large number of qubits?

---

> ### Author Response · Authors · 2025-11-22
> **Author Response to Reviewer Ki6d (1)**
>
> **Q1.** Can the authors specify what happens, in terms of the algorithm’s computational complexity, if the target states have a high level of entanglement?
>
> **Response to Q1.** We thank the reviewer for this question. In the following, we address the question in two parts.
>
> *1. Computational complexity of AQER for vector-stored classical data.* Let $d$ be the dimension of the input vector and $N = \lceil \log_2 d \rceil$ be the qubit number of the target state. We follow notations in the main text and denote by  $T$ and $T_3$ the numbers of iterations in Steps I and III of AQER, respectively. Then **the computational complexity of AQER for classical data is**  $\mathcal{O}(dT_3 T^2 \text{polylog}(d))$.  When $T, T_3 \leq \text{poly}(N)=\text{polylog}(d)$, the leading term in the complexity becomes $\mathcal{O}(d)$, which matches the cost of storing and accessing all $d$ entries of the vector. In practice, the dimension $d$ of the classical data vector is itself limited by both the available classical memory used to store the data (which scales as $\mathcal{O}(d)$) and the feature dimension of classical data in practice. For example, MNIST images have $d = 28 \times 28 = 784$ ($N \approx 10$ qubits), and even a high-resolution $1024 \times 1024$ image has $d \approx 10^6$ ($N \approx 20$ qubits). Thus, the dimension $d$ would not be extremely large, and the qubit number $N=\mathcal{O}(\log d)$ would only have modest scalings.
>
> For completeness, we provide a sketch of the proof for the computational complexity of AQER.
>
> First, each single- or two-qubit gate used in AQER ($R _ Z$, $R _ Y$, and $R _ {ZZ}$ rotations) acts as a diagonal or 2-sparse matrix. Thus, applying any such gate to the state vector can be implemented as a matrix-vector multiplication in $\mathcal{O}(d)$ time. Besides, computing the reduced density matrix (RDM) of a constant number of qubits also costs $\mathcal{O}(d)$ time: for example, suppose the state vector is $\boldsymbol{v}=\sum _ {i _ 1=0}^{1} \cdots \sum _ {i _ N=0}^{1} v _ {i _ 1, \cdots, i _ N} \boldsymbol{e} _ {i _ 1, \cdots, i _ N}$  in the computational basis, then the RDM at $(i _ 1, i _ 2)$ is $\rho _ {j _ 1, j _ 2, k _ 1, k _ 2}=\sum _ {i _ 3,\cdots,i _ N} v _ {j _ 1, j _ 2, i _ 3, \cdots, i _ N} v _ {k _ 1, k _ 2, i _ 3, \cdots, i _ N}^*$. The above example can be generalized to one or more qubits cases with different qubit indices. Single-qubit entropies and the entanglement measure are then obtained from these RDMs at negligible additional cost. Therefore, each iteration of Step I costs $\mathcal{O}(d \cdot \text{poly}(N))$ time, and the total complexity of Step I is $\mathcal{O}(d T \text{poly}(N))$. Step II has the complexity $\mathcal{O}(d N)$, which covers the cost of computing $N$ single-qubit RDMs and the calculation of $2N$ gate parameters. Step III consists of $T _ 3$ iterations of full gradient evaluation and parameter update. The gradient has the dimension $5T+2N$. Using either the parameter-shift rule or automatic differentiation, the complexity of computing each partial derivative is $\mathcal{O}(d (T+N))$. Hence, a full gradient evaluation costs $\mathcal{O}(d (T+N)^2)$, and the total complexity of Step III is $\mathcal{O}(d T _ 3 (T+N)^2)=\mathcal{O}(dT _ 3 T^2 \text{polylog}(d))$. The detailed computational analysis of AQER for classical data is included in the revision.

---

> ### Author Response · Authors · 2025-11-22
> **Author Response to Reviewer Ki6d (2)**
>
> *2. Computational complexity of AQER for quantum data.* In the case of quantum data, one has access to a target state but not to its explicit classical description. RDMs and gradients are estimated from a finite number of measurement shots. Let $M$ be the number of measurement shots used to estimate each quantity required by AQER. Then **the computational complexity of AQER for quantum data is $\mathcal{O}(M T_3 T^2 \text{poly}(N))$**. When $T_3, T=\text{poly}(N)$, the complexity is $\mathcal{O}(M\text{poly}(N))$, which is efficient that scales polynomially to the qubit number and linearly to the number of measurement shots. In this regime, AQER is conceptually significant as a bridge from states with quantum access to explicit circuit representations, and practically valuable in that it provides an efficient and portable data-loading procedure for downstream algorithms. Specifically, AQER provides a systematic way to turn quantum access to a target state into an explicit preparation circuit, where the circuit information (the sequence of qubit pairs, gate types, and continuous parameters) is fully specified as a classical object. This circuit description can be stored, transmitted to other quantum processors, and reused as a data-loading subroutine for downstream tasks. When the entanglement measure can be reduced efficiently, both the generation of the preparation circuit and the data-loading cost for downstream algorithms remain efficient. This helps ensure that the data-loading step does not affect potential quantum advantages in subsequent quantum algorithms. Next, we discuss the complexity of AQER for highly entangled states in two regimes: (i) the worst case and (ii) a benign, structured case.
>
> **(i). In the worst case, the preparation of highly entangled states is computationally expensive for any method including AQER.** Specifically, states sampled from the Haar measure is almost surely highly entangled [1], and its gate complexity to achieve an $\varepsilon$-approximation is exponentially large in the qubit number [2]. Therefore, in the worst case, the gate complexity of preparing highly entangled states is exponential, which is independent of the particular algorithmic scheme. In this regime, no algorithm, including AQER, can fundamentally avoid the exponential complexity. However, this does not imply that AQER is inefficient for all highly entangled states.
>
> **(ii). For certain highly entangled target states, AQER is efficient.** To illustrate this, we have added experiments for the GHZ state in the revision. The $N$-qubit GHZ state is highly entangled in the sense that each single-qubit marginal is maximally mixed and the initial $\mathcal{S}=N$. Nevertheless, AQER can still reduce its entanglement efficiently. For $N=10$, the entanglement measure $\mathcal{S}$ decays quickly and drops below 0.03 after only 9 iterations. Correspondingly, the infidelity is reduced to lower than 2e-4 with only $T=9$ two-qubit gates:
>
> | $T$ | 1 | 2 | 3 | 4 | 5 | 6 | 7 | 8 | 9 |
> | --- | --- | --- | --- | --- | --- | --- | --- | --- | --- |
> | $\mathcal{S}$ | 9.0 | 8.0 | 7.0 | 6.0 | 5.0 | 4.0 | 3.0 | 2.0 | 0.025 |
> | infidelity | 0.5 | 0.5 | 0.5 | 0.5 | 0.5 | 0.5 | 0.5 | 0.5 | 1.2e-4 |
>
> The value of $\mathcal{S}$ and infidelity with increasing $T$ is plotted in Figure 12 in Appendix F. This example demonstrates that **a state can be highly entangled while still being structurally simple such that AQER can achieve low infidelity efficiently.**
>
> [1] Patrick Hayden, Debbie W. Leung, Andreas Winter. Aspects of generic entanglement. Comm. Math. Phys. Vol. 265, No. 1, pp. 95-117, 2006.
>
> [2] Michał Oszmaniec, Marcin Kotowski, Michał Horodecki, Nicholas Hunter-Jones. Saturation and recurrence of quantum complexity in random local quantum dynamics. Phys. Rev. X 14, 041068, 2024.

---

> ### Author Response · Authors · 2025-11-22
> **Author Response to Reviewer Ki6d (3)**
>
> **Q2.** How is the method applicable to high-dimensional classical data that require a large number of qubits?
>
> **Response to Q2.** We thank the reviewer for this question. *AQER can be applied to classical data as long as the complexity of storing and accessing the data itself is acceptable.* In the case of classical data that is stored explicitly as a vector $\boldsymbol{v} \in \mathbb{C}^d$, the computational complexity of AQER is $\mathcal{O}(dT_3 T^2 \text{polylog}(d))$, as provided in our Response to Q1. Here $T$ and $T_3$ denote the number of iterations in Step I and Step III of AQER, respectively. Next, we discuss the size of $T$ and $T_3$ separately.
>
> - The size of $T$. We do not expect that $T$ is small for completely arbitrary classical vectors. In fact, for a generic random vector, the state preparation circuit is known to require $\Omega(d)$ gates in the worst case [3], so no generic encoding scheme including AQER can be efficient (with $T=\text{poly}(N)$) in that regime. However, high-dimensional data of practical interest are typically highly structured. In our experiments, e.g., Figure 3(a) in the main text and Figure 6 in the appendix of the original submission, we observe that for datasets such as MNIST and CIFAR-10, the entanglement measure decreases rapidly as $T$ increases, and good approximation quality is achieved at relatively small $T$. For example, Table 1 in the main text shows that for MNIST, $T=40$ already yields the infidelity below 0.1. Therefore, we focus on the approximate loading of such practically relevant classical data, where $T$ grows at most polynomially in $N= \mathcal{O}(\log d)$.
> - The size of $T_3$. Once the entanglement measure $\mathcal{S}$ has been driven down by Step I, Theorem 3.1 in the main text already guarantees a corresponding upper bound on the infidelity, so a small $T_3$ is sufficient for benign performance. For example, as shown in Figure 7 in the Appendix of the original submission, for the MNIST task, the infidelity with $T_3=40$ and $T=100$ is smaller than that with $T_3=2000$ and $T=40$.
>
> In these regimes where $T, T_3  \leq \text{poly}(N)=\text{polylog}(d)$, the leading term in the complexity becomes $\mathcal{O}(d)$, which matches the cost of storing and accessing all $d$ entries of the vector. Thus, AQER does not introduce any additional polynomial factor in $d$ beyond the linear cost of loading the data classically. We remark that the dimension $d$ of the classical data vector is itself limited by the available classical memory used to store the data (which scales as $\mathcal{O}(d)$). So the resulting qubit number $N=\mathcal{O}(\log d)$ would only have modest scalings.
>
> [3] Martin Plesch, Časlav Brukner. Quantum-state preparation with universal gate decompositions. Phys. Rev. A 83, 032302, 2011.

---

> ### Author Response · Authors · 2025-11-22
> **Author Response to Reviewer Ki6d (4)**
>
> **Q3&W5.** Finally, it is not clear that the optimization described in Eq. (2) can be solved in polynomial time. Can the authors provide theoretical guarantees that the optimization step in Eq. (2) is efficient?
>
> **Response to Q3&W5.** We thank the reviewer for raising this question. *In principle, Eq. (2) can be optimized to $\mathcal{O}(\varepsilon)$-approximate global minimum in polynomial time.* Next, we provide a sketch of the proof.
>
> Recall that in the $t$-th iteration of Step I of AQER, we aim to find the qubit pair $\mathcal{I} _ t$ and the parameter $\boldsymbol{\alpha} _ t$ of the new gate block $V _ {\mathcal{I} _ t}(\boldsymbol{\alpha} _ t)$, such that the entanglement measure $\mathcal{S}$ of the resulting state is minimized. This is achieved by solving the optimization problem in Eq. (2), which is given as $(\mathcal{I} _ t, \boldsymbol{\alpha} _ t) = \arg \min _ { {\mathcal{I}}, {\boldsymbol{\alpha}} } \mathcal{S}(V _ {\mathcal{I}}(\boldsymbol{\alpha})| \boldsymbol{v} _ {t-1}\rangle)$. There are two types of variables in the optimization:
>
> 1. The qubit pair  $\mathcal{I} = (j,k)$, for which there are at most $\mathcal{O}(N^2)$ possible choices;
> 2. The gate parameters $\boldsymbol{\alpha} \in \mathbb{R}^5$ for a given pair.
>
> The optimization over different qubit pairs can be implemented by scanning all configurations. For a fixed $\mathcal{I}=(j,k)$, the optimization problem becomes $\arg \min _ {\boldsymbol{\alpha} \in \Omega} \mathcal{S}(V _ {\mathcal{I}}(\boldsymbol{\alpha})| \boldsymbol{v} _ {t-1}\rangle),$  where $\Omega = [0,2\pi]^5$. We remark that inserting the new gate block $V _ {\mathcal{I}}(\boldsymbol{\alpha})$ only changes the single-qubit entropies of two qubits $\mathcal{I}=(j,k)$, and the other $N-2$ single-qubit entropies **remain unchanged**. So we can decompose the whole entanglement measure as $\mathcal{S} = C + s_{\mathcal{I}}(\boldsymbol{\alpha})$, where $C$ is independent of $\boldsymbol{\alpha}$ and
> $s _ {\mathcal{I}}(\boldsymbol{\alpha}) = \mathcal{S} _ {\{j\}}(V(\boldsymbol{\alpha})| \boldsymbol{v} _ {t-1}\rangle) + \mathcal{S} _ {\{k\}}(V(\boldsymbol{\alpha})| \boldsymbol{v} _ {t-1}\rangle) = \mathcal{S}\bigl(V _ {\mathcal{I}}(\boldsymbol{\alpha}) \,\rho _ {\mathcal{I}}\, V _ {\mathcal{I}}(\boldsymbol{\alpha})^\dagger\bigr)$.
> Here, $\rho _ {\mathcal{I}}$ is the two-qubit reduced density matrix (RDM) of the state $|\boldsymbol{v} _ {t-1} \rangle \langle \boldsymbol{v} _ {t-1}|$ on the qubit pair $\mathcal{I}=(j,k)$. In the quantum data setting, the RDM $\rho _ {\mathcal{I}}$ can be obtained by standard two-qubit tomography using a number of copies polynomial in $\mathcal{O}(1/\varepsilon^2)$. In the classical data setting, it can be exactly computed from the classical representation of the state, as detailed in the response to W3.
>
> Given the matrix formulation of $\rho _ {\mathcal{I}}$, the remaining problem is to optimize the function $s _ {\mathcal{I}}$ in classical ways. We remark that $s _ {\mathcal{I}}$ is globally Lipschitz with a Lipschitz constant $C _ 1$. Thus, a naive **grid search** over $\Omega = [0,2\pi]^5$ with grid $\delta = \mathcal{O}(\varepsilon)$ finds an $\mathcal{O}(\varepsilon)$-approximate minimum. Any optimum point lies within a distance at most $\frac{\sqrt{5}}{2} \delta$ from some grid point. Due to the Lipschitz continuity, the value of this grid point is $\mathcal{O}(\varepsilon)$-close to the minimum. By considering the number of grid points $\mathcal{O}(1/\delta^5) = \mathcal{O}(1/\varepsilon^5)$ and $\mathcal{O}(N^2)$ choices of qubit pairs, we obtain an overall complexity $\mathcal{O}(N^2/\varepsilon^5)$.
>
> Finally, we emphasize that our algorithm does not require converging to the global minimum of Eq. (2) at each iteration. By Theorem 3.1, the infidelity is controlled by the value of $\mathcal{S}$, so any decrease in $\mathcal{S}$ directly tightens the infidelity bound. Thus, it suffices to find parameters that yield a non-trivial reduction of $\mathcal{S}$, rather than the exact global optimum. In our experiments, We run the built-in Nelder–Mead optimizer in Python or Julia on this 5-dimensional parameter space and begin the optimization from $\boldsymbol{\alpha}=\boldsymbol{0}$. The converged value of $\mathcal{S}$ is then equal to or lower than its value at $\boldsymbol{\alpha}=\boldsymbol{0}$, which equals to the value of $\mathcal{S}$ before adding the new gate block: $\mathcal{S}\bigl(V _ {\mathcal{I}}(\boldsymbol{0})|\boldsymbol{v} _ {t-1}\rangle\bigr) = \mathcal{S}(|\boldsymbol{v} _ {t-1}\rangle)$.

---

> ### Author Response · Authors · 2025-11-22
> **Author Response to Reviewer Ki6d (5)**
>
> **W1.** My main concern is the computational cost required to reach low infidelity in the general case. The theoretical bound in the paper is meaningful only in the limit S goes to 0. For target states with moderate or high entanglement, achieving a small S requires deeper circuits U, and it is unclear whether this can still be done with circuits of polynomial size.
>
> **Response to W1.** We thank the reviewer for this comment and address the concern in four aspects.
>
> *1. The concern about the computational cost in the general case is unnecessary in our setting.* The computational cost of reaching low infidelity for general target states is known to be exponentially large, regardless of the specific algorithmic method employed. Our work does **not** claim to overcome this inherent barrier. Instead, **our contribution on the algorithm side is a more gate-efficient loader for meaningful and practically relevant families of states.** These states include quantum encodings of classical datasets, states from circuits with polynomial numbers of gates, and ground state of quantum many-body systems. The efficient preparation of these states is both meaningful and valuable, which is crucial for downstream tasks such as classification and quantum property estimation. In these cases, AQER achieves lower or comparable infidelity using significantly fewer two-qubit gates than prior AQL approaches.
>
> *2. Theorem 3.1 is not only meaningful in the limit $S \rightarrow 0$.* Theorem B.4 (the full version of Theorem 3.1) in the appendix of the original submission already provides the exact formulations of both the lower and upper bounds, which remain meaningful beyond the $S \rightarrow 0$. In particular, for an $N$-qubit state $|\boldsymbol{v} _ {\text{target}}\rangle$ and a circuit $U$ with entanglement measure $\mathcal{S}(U^{\dagger}(|\boldsymbol{v} _ {\text{target}}\rangle) = S$, we prove that for all product states,
>
>  $1 - \big| \langle \boldsymbol{v} _ {\text{target}} | U | \psi _ {\text{product}} \rangle \big|^2 \ge f _ 1(S) := \frac{1 - \sqrt{2^{1 - S/N} - 1}}{2}$ ,
>
> and there exists a product state, such that
>
>  $1 - \big| \langle \boldsymbol{v} _ {\text{target}} | U | \psi _ {\text{product}}^{\prime} \rangle \big|^2 \le f _ 2(S) := \frac{1}{2} \left( 1 - \sqrt{2^{1-S+\lfloor S \rfloor} -1} + \lfloor S \rfloor \right)$ .
>
> Functions $f _ 1$ and $f _ 2$ remain meaningful beyond the limit $S \rightarrow 0$:
>
> - $f _ 1(S)$ is non-trivial over the full physical range $S \in [0, N]$;
> - $f _ 2(S)$ is meaningful for $S \in [0, 1]$, where it yields a constant upper bound that is strictly smaller than 1 (in particular, $f _ 2(1) = 1/2$).
>
> In the original version, we only presented the Taylor expansions of $f _ 1$ and $f _ 2$ to highlight the linear scaling of the error when $S\rightarrow 0$. In the revision, we have made the exact forms $f _ 1$ and $f _ 2$ explicit in the statement.

---

> ### Author Response · Authors · 2025-11-22
> **Author Response to Reviewer Ki6d (6)**
>
> *3. In the worst case, the encoding circuit produced by AQER can be exponentially large for certain moderately or highly entangled states.* This result comes from the fact that generating the encoding circuits for states with moderate-to-high entanglement is computationally hard in the worst case. In fact, a recent study [4] has shown that there exist $N$-qubit states prepared by circuits with $G$ two-qubit gates, for which any quantum learner that outputs a preparation circuit must runs in time $\exp(\mathcal{\Omega}(\min(G,N)))$. Since a wide belief [5] that a quantum learner is more powerful than classical learner, we expect that the complexity of all AQL methods, including AQER, will fall into this regime. For the case $G=cN$ with $0<c<1$, the entanglement of such states is moderate in the sense that not all qubits are entangled together, while the computational complexity of generating a preparation circuit is already exponential. Since the computational complexity of AQER scales quadratically to $T$ as provided in the response to Q1, and the size of the AQER encoding circuit is $\mathcal{O}(T+N)$, we conclude that there exist states with moderate-to-high entanglement for which the encoding circuit generated by AQER can be exponentially large. However, this does not imply that AQER is inefficient for all moderately or highly entangled states.
>
> *4. There exist highly entangled target states for which the encoding circuit generated by AQER has linear size in $N$.* To illustrate this, we have added experiments for the GHZ state in the revision. The $N$-qubit GHZ state is highly entangled in the sense that each single-qubit marginal is maximally mixed and the initial $\mathcal{S}=N$. Nevertheless, AQER can still reduce its entanglement efficiently. For $N=10$, the entanglement measure $\mathcal{S}$ decays quickly and drops below 0.03 after only 9 iterations. Correspondingly, the infidelity is reduced to lower than 2e-4 with only $T=9$ two-qubit gates:
>
> | $T$ | 1 | 2 | 3 | 4 | 5 | 6 | 7 | 8 | 9 |
> | --- | --- | --- | --- | --- | --- | --- | --- | --- | --- |
> | $\mathcal{S}$ | 9.0 | 8.0 | 7.0 | 6.0 | 5.0 | 4.0 | 3.0 | 2.0 | 0.025 |
> | infidelity | 0.5 | 0.5 | 0.5 | 0.5 | 0.5 | 0.5 | 0.5 | 0.5 | 1.2e-4 |
>
> The value of $\mathcal{S}$ and infidelity with increasing $T$ is plotted in Figure 12 in Appendix F. This example demonstrates that **a state can be highly entangled while still being structurally simple such that AQER can achieve low infidelity with an encoding circuit of linear size in $N$.**
>
> [4]. Haimeng Zhao, Laura Lewis, Ishaan Kannan, Yihui Quek, Hsin-Yuan Huang, Matthias C. Caro. Learning Quantum States and Unitaries of Bounded Gate Complexity. PRX Quantum 5, 040306. 2024.
>
> [5]. Casper Gyurik, Vedran Dunjko. Exponential separations between classical and quantum learners. arXiv:2306.16028.

---

> ### Author Response · Authors · 2025-11-22
> **Author Response to Reviewer Ki6d (7)**
>
> **W2.** Also, the upper bound is stated in terms of the existence of a product state with certain properties. This does not guarantee that a generic product-state ansatz will satisfy the same bound in practice.
>
> **Response to W2.** We thank the reviewer for this comment and clarify in two parts that (1) the upper bound does not rely on a generic product-state ansatz, and (2) the product state achieving the bound can be effectively constructed.
>
> *1. We only need a certain product state, rather than a generic product-state ansatz, to guarantee the upper bound.* Concretely, the existence of this certain product state shows that AQL could achieve a certified upper bound on the infidelity. This product state is then used in Step II of AQER to generate the gate parameters, so that for small $\mathcal{S}$, the resulting infidelity error is already small. The upper bound in Theorem 3.1 does not rely on the performance of a generic product-state ansatz. We do not claim (and do not need to claim) that an arbitrary product state will satisfy the same bound.
>
> *2. The product state achieving the upper bound can be realized constructively.* In fact, the proof of Theorem 3.1 in Appendix B is already constructive: it provides an explicit product state that achieves the stated upper bound. Concretely, this state is given by the tensor product of single-qubit states ${R _ Z(\beta _ i)} {R _ Y(\gamma _ i)} | 0\rangle$. Let $\rho^{(i)}$ be the reduced density matrix of $U^{\dagger} |\boldsymbol{v} _ {\text{target}} \rangle$ on the $i$-th qubit. Then, by Lemma B.2, $\beta _ i$ and $\gamma _ i$ can be expressed in terms of the matrix elements $\rho _ {00}, \rho _ {11}, \rho _ {10}$ of $\rho^{(i)}$ as
> $\beta _ i = \arg(\rho _ {10})$  and $\gamma _ i = \frac{\pi}{2} - \arcsin\left(\frac{\rho _ {00} - \rho _ {11}}{\sqrt{4 | \rho _ {10}|^2 + (\rho _ {00} - \rho _ {11})^2}}\right)$.
>
> The original statement of Theorem 3.1 may give the impression that the upper bound is only existential. In the revision, we have rephrased Theorem 3.1 to explicitly emphasize that the upper bound is constructive.
>
> **W3.** Another concern I have is that computing the entanglement measure requires access to the target state. For quantum data, this may be possible if multiple copies of the state are available, though the number of required copies is not made explicit. For classical data, however, one must first classically construct the full target state, which is not tractable at large numbers of qubits. In fact, the classical-data experiments in the paper are limited to around 10 qubits.
>
> **Response to W3.** We thank the reviewer for this comment and respond by separating the concern into two parts.
>
> *1. The computation of the entanglement measure is tractable for classical data.* As explained in our Response to Q1, for classical data given as a vector $\boldsymbol{v} \in \mathbb{C}^d$, the cost to obtain the entanglement measure $\mathcal{S}$ is $\mathcal{O}(d \text{polylog}(d))$. The leading term in the complexity is $\mathcal{O}(d)$, which matches the linear cost of storing and accessing all $d$ entries of the vector. In such regimes, $d$ would not be extremely large due to the limitation of classical memory, so computing $\mathcal{S}$ is easily manageable.
>
> *2. Why the qubit number of vector-stored classical data is modest.* In practice, the dimension $d$ of the classical data vector is itself limited by the available classical memory used to store the data (which scales as $\mathcal{O}(d)$). So the resulting qubit number $N=\mathcal{O}(\log d)$ would only have modest scalings. For example, MNIST images have $d = 28 \times 28 = 784$ ($N \approx 10$ qubits), and even a high-resolution $1024 \times 1024$ image has $d \approx 10^6$ ($N \approx 20$ qubits).

---

> ### Author Response · Authors · 2025-11-22
> **Author Response to Reviewer Ki6d (8)**
>
> **W4.** Even for quantum states, as far as I understand, the simulations proposed in the paper rely on tensor-network methods, which inherently assume limited entanglement. This suggests that highly entangled quantum data are effectively out of scope for the demonstrated results.
>
> **Response to W4.** We thank the reviewer for this comment and respond by separating the concern into three parts.
>
> *1. The use of tensor-network simulators is a technical choice for large-qubit numerical experiments, not an intrinsic limitation of AQER.* In our numerical experiments on quantum data, we employ tensor-network (TN) methods to simulate TFIM ground states up to $N=50$ qubits on classical hardware. This choice is due to the limitation of classical simulations (full state vectors become exponentially large), **not by any restriction of the AQER algorithm itself to low-entanglement cases.**
>
> *2. Even for limited entanglement cases, preparing encoding circuits can be provably hard in the worst case.* We remark that when the tensor-network formulation of a moderately or limitedly entangled state is not given explicitly, the preparation of encoding circuits is itself a nontrivial challenge. In fact, recent complexity-theoretic results show that there exist $N$-qubit states prepared by circuits with $G$ two-qubit gates for which any quantum learner that outputs a preparation circuit must run in time $\exp(\Omega(\min(G,N)))$ [4]. In the regime $G = cN$ with $0 < c < 1$, such states have only moderate entanglement in the sense that not all qubits are simultaneously entangled, yet learning a corresponding preparation circuit is already exponentially hard. Together with the widely held belief that quantum learners are more powerful than classical learners [5], this suggests that **no AQL method, including AQER, should be expected to admit polynomial-time guarantees for arbitrary states, no matter whether they are moderately or highly entangled**. Consequently, our goal is not to prove such worst-case guarantees, but rather to focus on the broad class of physically relevant quantum states and quantum encodings of classical data for which **the entanglement measure can be reduced efficiently by adding two-qubit gates**. In this regime, the number of iterations (and thus the computational cost) remains moderate.
>
> *3. For certain highly entangled target states, AQER is efficient.* To illustrate this, we have added experiments for the GHZ state in the revision. The $N$-qubit GHZ state is highly entangled in the sense that each single-qubit marginal is maximally mixed and the initial $\mathcal{S}=N$. Nevertheless, AQER can still reduce its entanglement efficiently. For $N=10$, the entanglement measure $\mathcal{S}$ decays quickly and drops below 0.03 after only 9 iterations. Correspondingly, the infidelity is reduced to lower than 2e-4 with only $T=9$ two-qubit gates:
>
> | $T$ | 1 | 2 | 3 | 4 | 5 | 6 | 7 | 8 | 9 |
> | --- | --- | --- | --- | --- | --- | --- | --- | --- | --- |
> | $\mathcal{S}$ | 9.0 | 8.0 | 7.0 | 6.0 | 5.0 | 4.0 | 3.0 | 2.0 | 0.025 |
> | infidelity | 0.5 | 0.5 | 0.5 | 0.5 | 0.5 | 0.5 | 0.5 | 0.5 | 1.2e-4 |
>
> The value of $\mathcal{S}$ and infidelity with increasing $T$ is plotted in Figure 12 in Appendix F. This example demonstrates that **a state can be highly entangled while still being structurally simple such that AQER can achieve low infidelity efficiently.**
>
> [4]. Haimeng Zhao, Laura Lewis, Ishaan Kannan, Yihui Quek, Hsin-Yuan Huang, Matthias C. Caro. Learning Quantum States and Unitaries of Bounded Gate Complexity. PRX Quantum 5, 040306. 2024.
>
> [5]. Casper Gyurik, Vedran Dunjko. Exponential separations between classical and quantum learners. arXiv:2306.16028.

---

> ### Author Response · Authors · 2025-11-26
> **Comment for Reviewer Ki6d**
>
> Dear Reviewer Ki6d:
>
> Thanks a lot for your helpful and thoughtful review of our paper! We have tried our best to address the mentioned concerns and have carefully revised the manuscript. Are there unclear explanations or remaining problems? We would be very grateful for further feedback and will do our best to address them.
>
> Best regards,
>
> The authors

---

> > ### Comment · Reviewer_Ki6d · 2025-11-27
> >
> > I thank the authors for their reply, which indeed clarifies many aspects of their work.
> >
> > However, I still feel that my main concern has not been fully addressed. The current efficiency guarantees are essentially conditional: the complexity is polynomial whenever the required number of steps T is itself polynomial. What I was asking is whether the authors can provide some form of dependence of the required number of steps T on the level of entanglement (measured, for example, by S) of the target state.
> >
> > I understand that in full generality such a bound may be impossible (e.g., there are highly entangled states that are efficiently preparable, and others that are provably hard). However an analytical characterization, even under additional structural assumptions on the target states (such as states generated by poly-depth circuits or arising from specific data families), of how T depends on S would be very useful.
> >
> > I am asking this in order to better understand the applicability of AQER, i.e., how large is the class of target quantum states for which we can expect the method to run in polynomial time, based on analytical considerations rather than only numerical evidence.
> >
> > Alternatively, the authors could characterize the class of target states for which the algorithm is efficient in terms other than the level of S itself. For instance, it would already be helpful to identify families of states for which one can a priori expect a low value of S, and hence efficient performance of AQER.

---

> ### Author Response · Authors · 2025-12-03
> **Summary of Reviewer Ki6d’s Official Review and Our Response**
>
> First, we would like to thank the reviewer for investing substantial time in reviewing our work and for the constructive comments that helped improve the paper. In the original official review, the reviewer acknowledged four main strengths:
>
> (1) the paper presents a **clear and generally applicable bound for all AQLs** in terms of the entanglement measure $\mathcal{S}$;
>
> (2) the AQER algorithm is explicitly designed to minimize $\mathcal{S}$, **aligning the heuristic with the theory**;
>
> (3) once target states are available, training AQER relies only on local measurements;
>
> (4) the numerical experiments provide encouraging evidence for the main claims.
>
> The reviewer also raised several weaknesses and questions that guided our revisions:
>
> (1) the computational cost required to reach low infidelity in the general case;
>
> (2) the existential formulation of the upper bound in the main theorem;
>
> (3) the practicality of computing of entanglement measure;
>
> (4) whether the scope of AQER is restricted to limited entangled states;
>
> (5) whether the optimziation in Eq. (2) is efficient.
>
> We carefully addressed these points in our detailed response submitted on November 22. In particular,
>
> (1) we provided a detailed complexity analysis and clarified that AQER is efficient when the entanglement measure can be reduced with a polynomial number of iterations. we also discussed existing complexity-theoretic results showing that efficiently approximating generic moderately or highly entangled states lies outside the AQL regime in general, and thus outside the scope of AQER as well.
>
> (2) We made the constructive formulation of Theorem 3.1 explicit in the revision.
>
> (3) We explained how to compute the entanglement measure for both classical and quantum data, and clarified that for vector-stored classical data the cost matches the linear cost of accessing the data.
>
> (4) We emphasized that the efficiency of AQER depends not on whether the target state is highly entangled in an absolute sense, but on how quickly the entanglement measure can be reduced by adding two-qubit gates. To demonstrate this, we added a GHZ state example, where a globally entangled state with maximal entanglement measure can be efficiently handled by AQER.
>
> (5) We gave a proof sketch that the optimization of Eq. (2) can find an $\epsilon$-approximated global minimum in $\text{poly}(\epsilon)$ time.

---

> ### Author Response · Authors · 2025-12-03
> **Response to the newly raised comments**
>
> In the subsequent comment, the reviewer asks for a characterization of families of states (such as poly-depth circuits or other states/vectors under structural assumptions) for which AQER is expected to be efficient. Below, we first explain why generic poly-depth random circuits raised by the reviewer fall outside the scope of AQL and AQER, and then present the Instantaneous Quantum Polynomial (IQP) circuit family as an analytically tractable example where AQER admits rigorous efficiency guarantees.
>
> *1. We do not expect that large-qubit states from random quantum circuits with poly-depths fall within the regime of AQL and AQER.* Recent complexity-theoretic work [1] shows that learning the approximate output distributions of random quantum circuits with only linear depth has **exponentially large** average-case complexity. We remark that an approximate encoding circuit for the target state naturally provides a procedure for generating its output distribution approximately. Thus, these hardness results suggest that, no AQL method can learn approximate encodings for large-qubit states generated from random quantum circuits with linear depths using sub-exponential time or resources. Therefore, highly entangled large-qubit states generated by polynomial-depth random circuits lie outside the regime where either AQL or AQER is expected to be efficient, and in particular we do not expect that $\mathcal{S}$ can be reduced efficiently in these cases.
>
> *2. AQER **efficiently** handles classical vectors and quantum states arising from IQP circuits.* In particular, we consider the all-to-all qubit connectivity. The IQP circuit state is $H^{\otimes N} [ \prod _ {\{i,j\} \in E} e^{-i \omega _ {ij} Z_i Z_j /2} ] H^{\otimes N} {|0\rangle}^{\otimes N}$, where $E$ is the iteraction graph and each $w _ {ij}$ is the angle parameter.
>
> - For the classical data case (state vectors of IQP circuits), we proved that (i) for discrete-angle IQP states (unknown angles on a known finite grid), AQER can reduce the entanglement measure $\mathcal{S}$ to zero in at most $T=|E|$ iterations of Step I, thereby exactly reconstructing the loading circuits; and (ii) for continuous-angle IQP states with i.i.d. angles in $[-\pi,\pi]$, AQER can reduce the entanglement measure $\mathcal{S}$ below any prescribed $\varepsilon>0$ with high probability, using at most $T=|E|$ iterations of Step I. The corresponding theorems and proofs are provided in Appendix H (Theorems H.1 and H.2).
> - For the quantum data case, we consider the fully connected IQP family with fixed non-Clifford angle $\pi/4$ on an unknown graph $E$ of maximum degree $D$, where only quantum access to the state is available. In this setting, we proved that AQER still recovers an exact loading circuit (i.e., reduces $\mathcal{S}$ to zero) with probability at least $1-\delta$ using at most $T=|E|$ iterations of Step I and a number of calls to the given IQP state that scales as $\text{poly}(N,|E|,2^{D},\log(1/\delta))$, which is polynomial or quasi-polynomial for bounded or polylogarithmic $D$. The corresponding theorem and proof are provided in Appendix H (Theorem H.3).
> - We remark that these IQP families are nontrivial. IQP circuits with bounded degree and either random or fixed constant angles are widely believed to be **classically hard** to sample from, as supported by existing complexity-theoretic results [2]. Our results therefore demonstrate that AQER can be provably efficient on a broad class of IQP states that are nontrivial from a complexity-theoretic perspective.
>
> [1] Alexander Nietner, Marios Ioannou, Ryan Sweke, Richard Kueng, Jens Eisert, Marcel Hinsche, Jonas Haferkamp. On the average-case complexity of learning output distributions of quantum circuits. Quantum, 2025.
>
> [2] Michael J. Bremner, Richard Jozsa, Dan J. Shepherd. Classical simulation of commuting quantum computations implies collapse of the polynomial hierarchy. Proc. A, 2011.

---

### Official Review · Reviewer_LZZQ · 2025-10-31

**Soundness:** 1
**Presentation:** 2
**Contribution:** 2
**Rating:** 6
**Confidence:** 4

**Summary:**

This paper introduces AQER, a novel three-stage algorithm for approximate quantum loading (AQL). The core idea is to construct a quantum circuit by greedily minimizing an entanglement measure, the sum of single-qubit Renyi-2 entropies. The authors provide a theoretical bound linking this measure to the final approximation error and demonstrate through extensive simulations on classical and quantum datasets that AQER outperforms several existing methods in both accuracy and gate efficiency.

**Strengths:**

The paper's primary strength is Theorem 3.1, which establishes a formal information-theoretic bound between the approximation infidelity and the proposed measure. This provides a solid theoretical justification for the algorithm's design, moving beyond purely heuristic approaches.

The entanglement-reduction-guided strategy for building the circuit is a novel and intelligent heuristic. It offers a structured method for ansatz construction that aims to position the optimization in a favorable region, which is a key challenge in variational algorithms.

The authors have conducted extensive and compelling numerical experiments across a diverse set of benchmarks, including classical data and quantum many-body states up to 50 qubits. The consistent and significant performance gains shown in Table 1 strongly support the practical effectiveness and scalability of the proposed method.

**Weaknesses:**

1. The paper compellingly argues that AQER mitigates barren plateaus in the final optimization. However, a critical discussion or numerical experiment is missing on the optimization landscape. The algorithm's success hinges on this greedy step being efficient. The paper would be significantly strengthened by a discussion of why the local structure of this cost function makes the optimization in Step I tractable.

2. The experimental comparison relies heavily on the two-qubit gate count, which is a good but incomplete metric. To make a fairer and more robust assessment of AQER's scalability, I suggest the authors consider the gate depth, the number of trainable parameters, measurement overhead, and training cost.

3. The study is performed under ideal, noise-free conditions. Given that the practical utility of any near-term algorithm depends on its noise resilience, the paper would be substantially more impactful with a discussion on this aspect. For instance, how might hardware noise affect the measurement of entanglement and the subsequent greedy circuit construction? Even a qualitative discussion or a small-scale simulation under a simple noise model would add significant value.

**Questions:**

See weakness

---

> ### Author Response · Authors · 2025-11-22
> **Author Response to Reviewer LZZQ (1)**
>
> **W1.** The paper compellingly argues that AQER mitigates barren plateaus in the final optimization. However, a critical discussion or numerical experiment is missing on the optimization landscape. The algorithm's success hinges on this greedy step being efficient.
>
> **Response to W1.** We thank the reviewer for this important comment. In the revision, we have strengthened both the **numerical evidence (1,2)** and the **discussion (3)** to show that **Step III optimization in AQER does not fall into the barren plateau regime.**
>
> *1. The gradient norm remains large as the number of qubits increases in the numerical experiments.* We have conducted the gradient-magnitude statistics for the numerical experiment of the transverse-field Ising model with system sizes $N = 10,20,30,40,50$ qubits:
>
> | N | 10 | 20 | 30 | 40 | 50 |
> | --- | --- | --- | --- | --- | --- |
> | T=10 | 0.080 | 0.029 | 0.024 | 0.026 | 0.047 |
> | T=20 | 0.063 | 0.061 | 0.034 | 0.032 | 0.045 |
> | T=40 | 0.052 | 0.13 | 0.086 | 0.051 | 0.054 |
> | T=60 | 0.052 | 0.12 | 0.12 | 0.094 | 0.090 |
> | T=100 | 0.057 | 0.12 | 0.14 | 0.13 | 0.13 |
>
> As shown in the table, the initial gradient norms remain greater than 0.02 across all system sizes and **do not exhibit an exponential decay with $N$**. This behavior provides numerical evidence that barren plateau effects are mitigated in our experiments. The result is also shown in Figure. 10 in Appendix F in the revision.
>
> *2. The trainability claim is numerically supported by the optimization curve statistics.* In fact, complementary numerical evidence supporting the trainability claim for the AQER Step III was already presented in the original submission. As shown in Figure 4(a) in the main text, the optimization curves for $N=50$ qubits display a clear and steady decrease of the loss throughout training. This effective loss reduction is consistent with the newly added gradient statistics and jointly supports the conclusion that our method does not suffer from barren plateaus in the considered large-qubit regime.
>
> *3. Steps I and II of AQER shape the optimization landscape of Step III into a trainable regime.* As discussed in the Appendix. D.3, Step III of AQER does not start training from a completely random circuit. Instead, Steps I and II gradually build a circuit that already has **nontrivial fidelity with the target state**: Step I selects two-qubit gates that explicitly reduce the entanglement measure, while Step II assembles these gates into an ansatz that is already a reasonably good approximation by theoretical bounds in Theorem 3.1. By the time we reach Step III, the parameters are therefore initialized in a region where the loss is well below 1 and the state is far from a highly random state.

---

> ### Author Response · Authors · 2025-11-22
> **Author Response to Reviewer LZZQ (2)**
>
> **W2.** The paper would be significantly strengthened by a discussion of why the local structure of this cost function makes the optimization in Step I tractable.
>
> **Response to W2.** We thank the reviewer for raising this point. In short, the tractability of Step I comes from the fact that the entanglement measure we use is **local**, so the global optimization decomposes into many small classical optimization problems. **Each of these classical optimizations involves only 5 real parameters, which is tractable.** Next, we explain this more concretely.
>
> Recall that in the $t$-th iteration of Step I of AQER, we aim to find the qubit pair $\mathcal{I} _ t$ and the parameter $\boldsymbol{\alpha} _ t$ of the new gate block $V _ {\mathcal{I} _ t}(\boldsymbol{\alpha} _ t)$, such that the entanglement measure $\mathcal{S}$ of the resulting state is minimized. This is achieved by solving the optimization problem in Eq. (2), which is given as $(\mathcal{I} _ t, \boldsymbol{\alpha} _ t) = \arg \min _ { {\mathcal{I}}, {\boldsymbol{\alpha}} } \mathcal{S}(V _ {\mathcal{I}}(\boldsymbol{\alpha})| \boldsymbol{v} _ {t-1}\rangle)$. There are two types of variables in the optimization:
>
> 1. The qubit pair  $\mathcal{I} = (j,k)$, for which there are at most $\mathcal{O}(N^2)$ possible choices;
> 2. The gate parameters $\boldsymbol{\alpha} \in \mathbb{R}^5$ for a given pair.
>
> The optimization over different qubit pair can be implemented by scanning all configurations. For a fixed qubit pair $\mathcal{I} = (j,k)$, the corresponding two-qubit gate block can only affect $\rho _ j$ and $\rho _ k$, so its effect on $\mathcal{S}$ is equal to its effect on
> $\mathcal{S}(\rho _ j) + \mathcal{S}(\rho _ k) = \mathcal{S}(\rho _ {\mathcal{I}})$, where $\rho _ {\mathcal{I}} \in \mathbb{C}^{4\times 4}$ is the two-qubit RDM of the state $|\boldsymbol{v} _ {t-1} \rangle \langle \boldsymbol{v} _ {t-1}|$ on qubits $(j,k)$.
> In the quantum data setting, the RDM $\rho_{\mathcal{I}}$ can be obtained by standard two-qubit tomography using a number of copies polynomial in $\mathcal{O}(1/\varepsilon^2)$. In the classical data setting, $\rho_{\mathcal{I}}$ can be exactly computed from the classical state vector: for example, suppose the state vector is $\boldsymbol{v}=\sum _ {i _ 1 = 0}^{1} \cdots \sum _ {i _ N = 0}^{1} v _ {i _ 1, \cdots, i _ N} \boldsymbol{e} _ {i _ 1, \cdots, i _ N}$  in the computational basis, then the RDM at $(i _ 1, i _ 2)$ is $\rho _ {j _ 1, j _ 2, k _ 1, k _ 2}=\sum _ {i _ 3, \cdots, i _ N} v _ { j _ 1, j _ 2, i _ 3, \cdots, i _ N} v _ { k _ 1, k _ 2, i _ 3, \cdots, i _ N}^*$. Given the matrix formulation of $\rho _ {\mathcal{I}}$, the remaining problem is to optimize the function $s _ {\mathcal{I}}(\boldsymbol{\alpha})=\mathcal{S}\bigl( V _ {\mathcal{I}}(\boldsymbol{\alpha}) \,\rho _ {\mathcal{I}}\, V_{\mathcal{I}}(\boldsymbol{\alpha})^\dagger\bigr)$ in classical ways, where $\boldsymbol{\alpha} \in \Omega$ and $\Omega = [0,2\pi]^5$.
>
> In practice, we use the Nelder–Mead optimizer to optimize the parameter $\boldsymbol{\alpha}$. We remark that we do not need to find the global minimum of $s _ {\mathcal{I}}(\boldsymbol{\alpha})$, since a reasonable decay in $\mathcal{S}$ could make AQER work well in practice. However, in principle, the global minimum of this 5-dimensional optimization can still be found in polynomial time. Since $s _ {\mathcal{I}}$ is globally Lipschitz with a Lipschitz constant $C _ 1$, a naive **grid search** over $\Omega = [0,2\pi]^5$ with grid $\delta = \mathcal{O}(\varepsilon)$ finds an $\mathcal{O}(\varepsilon)$-approximate minimum: any optimum point lies within a distance at most $\frac{\sqrt{5}}{2} \delta$ from some grid point, and due to the Lipschitz continuity, the value of this grid point is $\mathcal{O}(\varepsilon)$-close to the minimum. By considering the number of grid points $\mathcal{O}(1/\delta^5) = \mathcal{O}(1/\varepsilon^5)$, we obtain the polynomial complexity $\mathcal{O}(1/\varepsilon^5)$ to obtain the $\mathcal{O}(\varepsilon)$-approximate minimum of $s _ {\mathcal{I}}(\boldsymbol{\alpha})$.

---

> ### Author Response · Authors · 2025-11-22
> **Author Response to Reviewer LZZQ (3)**
>
> **W3.** The experimental comparison relies heavily on the two-qubit gate count, which is a good but incomplete metric. To make a fairer and more robust assessment of AQER's scalability, I suggest the authors consider the gate depth, the number of trainable parameters, measurement overhead, and training cost.
>
> **Response to W3.** We thank the reviewer for this constructive suggestion. In the revision, we have added a new figure (Figure 9 in Appendix F) that explicitly reports the four metrics raised by the reviewer. Specifically, for the state of random quantum circuit (S-RQC) task introduced in Section 4.1 in the original submission, we compare the proposed AQER with reference methods based on hardware-efficient circuit (HEC), automatic quantum circuit encoding (AQCE), and matrix product state (MPS). We plot the achieved infidelity as a function of (a) circuit depth, (b) number of trainable parameters, (c) total number of measurement shots, and (d)training time (seconds) in classical simulations.
>
> Overall, **AQER reaches lower infidelity than the other methods with comparable or lower circuit depth and parameter count**. For **measurement shots** and **training time**, the behavior is more nuanced. The MPS baseline, due to the simplicity of its underlying idea, requires fewer measurement shots and shorter training times. However, this comes at the cost of substantially worse performance in terms of circuit depth and parameter count. For circuit-based baselines (AQCE and HEC), **AQER achieves lower infidelity with comparable or lower measurement overhead and classical training time**.
>
> To provide a more quantitative comparison, for two representative accuracy regimes, we summarize the resource cost of each method in terms of these four metrics together with the achieved infidelity. The results are given in two tables below. The first table corresponds to a moderate-accuracy regime (infidelity around 0.4–0.5):
>
> | Method | Depth | #Params | #Shots | Training time (s) | Infidelity |
> | --- | --- | --- | --- | --- | --- |
> | AQER (Ours) | **6.2** | **70** | 1.41e+10 | 42.2 | 0.461 |
> | HEC | 8 | 160 | 3.2e+10 | 63.0 | **0.375** |
> | AQCE | 16.2 | 150 | 2.7e+10 | 252.0 | 0.472 |
> | MPS | 45 | 540 | **4.0e+06** | **0.0935** | 0.490 |
>
> In this regime, HEC attains slightly lower infidelity than AQER (0.375 vs. 0.461), but at noticeably higher resource cost in circuit depths (8 vs. 6.2), use more than twice as many trainable parameters (160 vs. 70), and require more than twice as many measurement shots (3.2e10 vs. 1.41e10) and longer training time (63 s vs. 42.2 s). AQCE is worse than AQER in both accuracy (0.472 vs. 0.461) and all resource metrics (depth, parameters, shots, and time). The MPS baseline, due to its simple structure and the absence of circuit/parameter optimization, uses orders-of-magnitude fewer measurement shots and negligible training time, but this comes at the cost of the largest circuit depth and parameter count and the worst infidelity (0.490) among all methods.
>
> The second Table focuses on a higher-accuracy regime (infidelity around 0.2–0.3):
>
> | Method | Depth | #Params | #Shots | Training time (s) | Infidelity |
> | --- | --- | --- | --- | --- | --- |
> | AQER (Ours) | **11.8** | **120** | 2.41e+10 | 67.8 | **0.196** |
> | HEC | 12 | 240 | 4.8e+10 | 92.6 | 0.259 |
> | AQCE | 30 | 300 | 9.01e+10 | 841.0 | 0.236 |
> | MPS | 51 | 675 | **5.0e+06** | **0.116** | 0.453 |
>
> In this more stringent regime, AQER achieves the lowest infidelity (0.196) among all methods while using the smallest circuit depth and parameter count: its depth is comparable to but slightly smaller than HEC (11.8 vs. 12) and significantly smaller than AQCE and MPS (30 and 51), and its number of trainable parameters (120) is about a factor of two less than HEC (240) and substantially less than AQCE and MPS (300 and 675). In terms of measurement overhead and classical training time, AQER again compares favorably with the other circuit-based baselines: relative to HEC, it reaches lower infidelity with roughly half the number of shots (2.41e10 vs. 4.8e10) and shorter training time (67.8 s vs. 92.6 s), and the gap is even larger compared to AQCE (which requires 9.01e10 shots and 841 s to achieve higher infidelity 0.236). The MPS baseline still enjoys the smallest shot count and training time, but its infidelity (0.453) remains much higher and does not enter the same low-error regime as the circuit-based methods under comparable circuit depth and parameter-count budgets.

---

> ### Author Response · Authors · 2025-11-22
> **Author Response to Reviewer LZZQ (4)**
>
> **W4.** The study is performed under ideal, noise-free conditions. Given that the practical utility of any near-term algorithm depends on its noise resilience, the paper would be substantially more impactful with a discussion on this aspect. For instance, how might hardware noise affect the measurement of entanglement and the subsequent greedy circuit construction? Even a qualitative discussion or a small-scale simulation under a simple noise model would add significant value.
>
> **Response to W4.** We are grateful to the reviewer for pointing out this weakness and for prompting a more detailed clarification. In the revision, we address this problem in three levels: (1) a theoretical extension of Theorem 3.1 to general noisy channels (Theorems C.1 and C.2), (2) a discussion on how noise affects entanglement estimation and the gate construction in Step I, and (3) small-scale simulations under depolarizing noise channels.
>
> *1. Theoretical extension to noisy channels (Appendix C).* We have extended Theorem 3.1 to **general CPTP maps** in Appendix C:
>
> - **Theorem C.1** shows that when the target state is passed through an arbitrary CPTP map $\mathcal{E}$, the entanglement measure $\mathcal{S}(\mathcal{E}(\rho_{\text{target}})) = S$ still controls the achievable infidelity with product states, with the **same functional form** as in Theorem 3.1. In other words, the entanglement-based characterization survives in the noisy setting.
> - **Theorem C.2** analyzes **layered noisy circuits** with interleaved noise channels $\mathcal{M}$ and $\mathcal{N}$ and shows that the original bounds are modified only by an **explicit additive noise term** that scales as $(L+1)\big( \| \mathcal{M} - \mathrm{id} \| _ \diamond + \| \mathcal{N} - \mathrm{id} \| _ \diamond\big)$, i.e., linearly in both the depth and the noise strength. For standard channels such as depolarizing channels, the diamond norm term scales linearly with the error rate $p$. Thus, as long as both the entanglement measure and the accumulated noise remain moderate, the entanglement-governed guarantees for AQER continue to hold with a controllable noise-dependent correction.
>
> *2. How do hardware noises affect the calculation of entanglement measurement?* In AQER, the entanglement measure $\mathcal{S}(\rho) = \sum_{n=1}^N - \log _ 2 \mathrm{Tr}[\rho _ n^2]$ depends only on single-qubit reduced density matrices (RDM) $\rho _ n$. On quantum hardware, we can estimate RDMs from local measurements of the noisy state $\tilde{\rho} _ n = \text{Tr} _ {[N]/{\{n\}}} \mathcal{E}(\rho _ {\text{target}})$, where $\mathcal{E}$ is the noisy quantum circuit produced by the device. In this case, hardware noise could lead to a biased estimation on the entanglement measure.
> For example, as provided in Theorem C.3 in Appendix C, for the case of a global depolarizing channel $\mathcal{D} _ p(\rho):=(1-p)\rho+pI/d$, $\left( 1 - \frac{p}{\ln4}\right) \mathcal{S}(\rho) + \frac{Np}{\ln4} \le \mathcal{S}(\mathcal{D} _ p(\rho)) \le \mathcal{S}(\rho) + N\log _ 2 \frac{2}{1+(1-p)^2}$. Combining this with the linear dependence between infidelity and the entanglement measure in Theorem C.1, we conclude that adding noise to the circuit negatively impacts the minimal achievable entanglement measure and infidelity of general AQLs, which include AQER. This result is also consistent with the conclusion of Theorem C.2. Nevertheless, as long as the noise strength and circuit depth remain in a moderate regime, this negative impact on the entanglement measure and infidelity remains well controlled.

---

> ### Author Response · Authors · 2025-11-22
> **Author Response to Reviewer LZZQ (5)**
>
> *3. Noisy simulations show the robustness of AQER in NISQ regimes.* Following the reviewer’s suggestion, we added **noisy channel simulations** in the revision. In particular, we apply AQER to the $N=10$ qubit transverse-field Ising model task under global depolarizing channels. For the single-qubit gate layer, we adopt the error rate at different levels $p_1 \in \{10^{-4}, 3\times 10^{-3}, 10^{-3}\}$, and the corresponding error rate for the two-qubit gate layer is $p_2=10p_1$. These values are chosen to be representative of gate error rates reported on current NISQ devices [1, 2], where single-qubit (two-qubit) gate errors lie in the $10^{-4}\text{–}10^{-3}$ ($10^{-3}\text{–}10^{-2}$) regime. In the experiment, we observe that, as the two-qubit count $T$ increases, the final infidelity reduces up to an optimal $T$ point. Beyond this point, the entanglement measure starts to increase again as accumulated noise dominates.
>
> | T | 5 | 7 | 10 | 12 | 15 | 20 | 24 | 28 | 32 | 36 | 40 | 50 | 60 | 80 | 100 |
> | --- | --- | --- | --- | --- | --- | --- | --- | --- | --- | --- | --- | --- | --- | --- | --- |
> | infidelity ($p_1=10^{-3},p_2=10^{-2}$ ) | 0.39 | 0.33 | 0.22 | 0.19 | **0.17** | **0.17** | 0.18 | 0.20 | 0.22 | 0.23 | 0.24 | 0.29 | 0.32 | 0.40 | 0.46 |
> | infidelity ($p_1=3\times10^{-4},p_2=3\times10^{-3}$ ) | 0.38 | 0.30 | 0.17 | 0.12 | 0.092 | 0.074 | **0.071** | **0.071** | 0.073 | 0.072 | 0.076 | 0.091 | 0.11 | 0.13 | 0.16 |
> | infidelity ( $p_1=10^{-4},p_2=10^{-3}$ ) | 0.39 | 0.31 | 0.16 | 0.11 | 0.086 | 0.057 | 0.050 | 0.037 | 0.035 | **0.033** | 0.035 | 0.040 | 0.042 | 0.051 | 0.060 |
>
> These results demonstrate the robustness of AQER in NISQ regimes:
>
> - For **all tested noise levels**, there exists a noise-dependent optimal $T$ at which AQER achieves a **lower infidelity** compared to shallow circuits, indicating that the entanglement-guided optimization remains beneficial before noise accumulation becomes dominant.
> - As the physical error rates decrease from $p_1 = 10^{-3}$ to $p_1 = 10^{-4}$, the **optimal infidelity improves gradually**.
>
> These observations are consistent with Theorem C.2: the entanglement-governed improvement persists, while noise introduces an additional error term that grows linearly with depth and noise strength. Taken together, there is an optimal circuit depth at which a noisy quantum implementation of AQER achieves its minimal error; beyond this depth, additional layers mainly amplify noise and degrade performance.
>
> [1] Dmitry A. Abanin, et al. Observation of constructive interference at the edge of quantum ergodicity. Nature 646, 825–830, 2025.
>
> [2] Anthony Ransford, et al. Helios: A 98-qubit trapped-ion quantum computer. arXiv:2511.05465.

---

> ### Author Response · Authors · 2025-11-26
> **Comment for Reviewer LZZQ**
>
> Dear Reviewer LZZQ:
>
> Thanks a lot for your helpful and thoughtful review of our paper! We have tried our best to address the mentioned concerns and have carefully revised the manuscript. Are there unclear explanations or remaining problems? We would be very grateful for further feedback and will do our best to address them.
>
> Best regards,
>
> The authors

---

> > ### Comment · Reviewer_LZZQ · 2025-11-27
> > **Question on the Author reply**
> >
> > Thank you for the detailed response. However, significant concerns regarding the scalability and generalizability of the proposed method remain unresolved.
> >
> > Regarding the $N=50$ experiment, you stated that Step I optimizes only one qubit pair (approx. 5 parameters) at a time. This implies that for a 50-qubit system, 48 qubits are effectively "frozen" during each optimization step. While this local strategy may mitigate Barren Plateaus, it is essentially a greedy, coordinate-descent-like approach. It is well-established that such strategies are prone to getting stuck in local minima, particularly when the target state possesses global, multi-partite entanglement. The ability to converge on $N=50$ strongly suggests that the optimization landscape was artificially simple due to the nature of the target states.
> >
> > I am unconvinced by the technical feasibility of the cost function evaluation for the claimed
> > $N=50$ experiments. Storing a general quantum state vector for $N=50$ requires ∼16 PB of memory, which is computationally intractable.
> > Could you explicitly clarify how the target state and the evolving state were represented in your classical simulations?
> >
> > These points reinforce the suspicion that AQER’s success on $N=50$ is conditional on the target datasets being structurally simple.
> > Have you tested AQER on states with long-range, volume-law entanglement (e.g., outputs of deep random quantum circuits) to strengthen the claim of "Scalability to 50 qubits"? Otherwise, it should be qualified to reflect that the method is likely scalable only for low-entanglement states, not for general quantum computing tasks.

---

> ### Author Response · Authors · 2025-12-03
> **Summary of Reviewer LZZQ’s Official Review and Our Response**
>
> First, we would like to thank the reviewer for investing substantial time in reviewing our work and for the constructive comments that helped improve the paper. In the original official review, the reviewer gave a positive score and a recommendation to accept. The review highlighted three main strengths:
>
> (1) “Theorem 3.1 establishes a formal information-theoretic bound between the approximation infidelity and the proposed measure. This provides a **solid theoretical justification for the algorithm's design, moving beyond purely heuristic approaches**.”
>
> (2) “The entanglement-reduction-guided strategy for building the circuit is **a novel and intelligent heuristic**. It offers a structured method for ansatz construction that aims to position the optimization in a favorable region, which is a key challenge in variational algorithms.”
>
> (3) “Extensive and compelling numerical experiments across a diverse set of benchmarks, including classical data and **quantum many-body states up to 50 qubits, strongly support the practical effectiveness and scalability** of the proposed method.”
>
> The reviewer also pointed out several weaknesses that helped identify directions to strengthen the paper:
>
> (1) the need for additional numerical evidence or discussion showing that AQER mitigates barren plateaus in the final optimization;
>
> (2) more discussion of why Step I is tractable;
>
> (3) more metrics, including circuit depth, number of trainable parameters, measurement overhead, and training cost;
>
> (4) more analysis or simulations in the presence of hardware noise.
>
> We carefully addressed these points in our detailed response. In particular,
>
> (1) we showed that the Step III optimization in AQER has non-vanishing initial gradients and effectively decaying loss values, i.e., it does not fall into the barren-plateau regime, and explained how Steps I and II reshape the landscape into a trainable regime.
>
> (2) We clarified that Step I decomposes into small classical optimizations that are tractable.
>
> (3) We provided additional comparisons and discussion on the four metrics raised by the reviewer.
>
> (4) We extended Theorem 3.1 to general noisy channels (Theorems C.1 and C.2) and conducted quantum noise–channel simulations showing that AQER is robust against such noise.

---

> ### Author Response · Authors · 2025-12-03
> **Response to the newly raised comments (1)**
>
> In the subsequent comment, the reviewer does not object to our responses to the original weaknesses, but instead raises additional concerns that go beyond the weaknesses in the initial review. In particular, the reviewer raised concerns mainly that the “**scalability and generalizability of the proposed method**” may not apply to “**general quantum computing tasks.**”
>
> This new comment is not a criticism of AQER itself, but rather a **misunderstanding of the scope of the approximate quantum data loading (AQL) problem considered in our paper**. Below, we first explain in three parts why the main concern is not appropriate as a critique of our work, and then address the specific questions raised in the new comment point by point.
>
> **Response** **to the main concern** on the scalability and generalizability of AQER for general quantum states.
>
> *1. General quantum states have exponential gate complexity.* Existing theoretical results show that preparing an approximate loading (encoding) circuit for general quantum states requires an exponential number of gates in the worst case. In particular, several works have established that states sampled from the Haar measure are almost surely highly entangled [1], and their gate complexity grows exponentially with the number of qubits [2]. As a consequence, **no algorithm** that is constrained to use only a sub-exponential number of gates **can achieve non-trivial fidelity** for such states in the worst case, and **no method can offer meaningful scalability guarantees** with sub-exponential gate complexity over general quantum states. This is a fundamental limitation, independent of the specific design of the algorithm.
>
> *2. The scope of AQL does not target general quantum states.* As explained in the general response, the AQL focuses on fixing a **polynomial** budget of quantum gates and then seeking an approximate encoding whose infidelity is as small as possible under this constraint. In other words, **AQL is a “gate-count–first, fidelity–second’’ regime**: we are not trying to prepare arbitrary states with unrestricted gate resources, but to achieve the best possible approximation within a realistic gate budget. This regime is well motivated in both near-term and fault-tolerant scenarios. On NISQ devices, a smaller gate count directly reduces noise accumulation. In the fault-tolerant case, it leads to lower logical gate complexity, which is critical for the overall efficiency and resource cost of quantum algorithms. Given the complexity barrier in (1) above, **AQL methods (both existing methods [3-7] and the proposed AQER) are not intended to be scalable for all general quantum states.**
>
> *3. AQER follows the scope of conventional AQL methods and targets structured data.* In line with existing AQL methods [3-7], AQER focuses on structured data, including practical classical datasets and physically meaningful quantum states such as ground states of local quantum many-body Hamiltonians. In these settings, we evaluate AQER under finite gate budgets and observe that, **under comparable gate constraints, AQER achieves lower infidelity than existing AQL baselines**. Methodologically, AQER improves over two main classes of AQL approaches. Compared to tensor-network (TN)–based methods [3,5,6], **AQER constructs encoding circuits more flexibly** by incrementally adding two-qubit gate blocks, rather than following a rigid TN-derived circuit structure. Compared to existing heuristic circuit-based methods [4,7] without theoretical guidance, AQER explicitly monitors the entanglement measure during gate generation. Together with Theorem 3.1, which links this entanglement measure to the approximation error (infidelity), this makes AQER, to our knowledge, **the first theoretically guided heuristic AQL algorithm** whose circuit construction is supported by a concrete theoretical performance guarantee.
>
> [1] Patrick Hayden, et al. Aspects of generic entanglement. Comm. Math. Phys. Vol. 265, No. 1, pp. 95-117, 2006.
>
> [2] Michał Oszmaniec, et al. Saturation and recurrence of quantum complexity in random local quantum dynamics. Phys. Rev. X, 2024.
>
> [3] Shi-Ju Ran. Encoding of matrix product states into quantum circuits of one- and two-qubit gates. Phys. Rev. A, 2020.
>
> [4] Manuel S Rudolph, et al. Synergistic pretraining of parametrized quantum circuits via tensor networks. Nat. Commun., 2023.
>
> [5] Bernhard Jobst, et al. Efficient MPS representations and quantum circuits from the Fourier modes of classical image data. Quantum, 2024.
>
> [6] Jason Iaconis, et al. Quantum state preparation of normal distributions using matrix product states. Npj Quantum Inf., 2024.
>
> [7] Tomonori Shirakawa, et al. Automatic quantum circuit encoding of a given arbitrary quantum state. Phys. Rev. Res., 2024.

---

> ### Author Response · Authors · 2025-12-03
> **Response to the newly raised comments (2)**
>
> We now respond to the specific points raised in the new comment.
>
> **Q1.** The reviewer suspects that $N = 50$ experiments use full state-vector simulation, which would be classically intractable.
>
> **Response to Q1.** *We did **not** conduct the $N = 50$ experiments using full state-vector simulations.* As stated at the beginning of Appendix E in the current revision (Appendix D in the original submission), for the transverse-field Ising model (TFIM) ground-state benchmarks with 20 or more qubits, we use PastaQ [1], a Julia-based package that employs tensor-network (TN) techniques to simulate large quantum states. In particular, in our experiments, the bond dimension is capped at $\chi = 100$. The memory cost of both the target and the evolved states scales as $\mathrm{poly}(N,\chi)$ instead of $\mathcal{O}(2^N)$, and **remains fully tractable in practice**. Using TN-based simulators to access larger-qubit regimes is standard practice in recent quantum machine learning and quantum many-body studies published in top journals and major machine learning conferences [2,3], and our setup follows this convention. We have made this implementation detail explicit in Section 4.2 of the revised main text.
>
> [1] Giacomo Torlai, et al. PastaQ: A package for simulation, tomography and analysis of quantum computers, 2020.
>
> [2] Ya-Dong Wu, et al. Learning quantum properties from short-range correlations using multi-task networks. Nat. Commun., 2024.
>
> [3] Yehui Tang, et al. QuaDiM: A Conditional Diffusion Model For Quantum State Property Estimation. ICLR 2025.
>
> **Q2.** On the concern that Step I is “greedy, coordinate-descent-like” and prone to local minima, particularly when the target state possesses global, multi-partite entanglement.
>
> **Response to Q2.** *Local minima are inherent to all variational, circuit-based AQL methods; AQER is designed to mitigate, not magically remove, this issue.* We emphasize that AQER (like other parameterized-circuit-based AQLs) solves a non-convex optimization problem during training the parameters. Local minima are **generally unavoidable** for these AQLs whose gate count does not scale exponentially [1]. With this in mind, we respond to the reviewer’s concern in four parts.
>
> *1. Step I does not permanently* *freeze 48 qubits.* In Step I, we do not optimize a single fixed pair while keeping the remaining qubits permanently frozen. Instead, AQER maintains a set of candidate qubit pairs and **sweeps over them**, so that each qubit participates in many local updates during Step I.
>
> *2. Scope of AQER: we do not aim to solve arbitrary globally entangled cases.* For general target states, eliminating local-minima issues in the parameter optimization requires overparameterized circuits with exponentially many gates [1], which is beyond the scope of AQL. As a method for AQL, AQER is therefore not intended as a universal solution for general states.
>
> *3. AQER **mitigates**, rather than eliminates, local-minima issues via entanglement-guided circuit design*. This mitigation effect is supported empirically across all datasets we considered. As shown in Table 1 and Figure 3(b) of the main text, the optimized infidelity after the parameter refinement in Step III consistently decreases as the circuit size generated by Step I increases. Thus, adding **more** entanglement-guided gates leads to **better** **local minima** rather than causing optimization to stagnate. Moreover, under comparable two-qubit gate counts, AQER consistently achieves **lower optimized infidelity** than the hardware-efficient circuit (HEC) baseline across all datasets. These observations suggest that AQER induces a more favorable optimization landscape, with **better local minima** than HEC.
>
> *4. For certain highly entangled target states, AQER has no local minima issue.* To illustrate this, we have added experiments for the GHZ state in the revision. The $N$-qubit GHZ state is globally entangled and has initial $\mathcal{S}=N$. Nevertheless, AQER can still reduce its entanglement efficiently without being trapped. For $N=10$, the entanglement measure $\mathcal{S}$ decays quickly and drops below 0.03 after only 9 iterations. Correspondingly, the infidelity is reduced to lower than 2e-4 with only $T=9$ two-qubit gates:
>
> | $T$ | 1 | 2 | 3 | 4 | 5 | 6 | 7 | 8 | 9 |
> | --- | --- | --- | --- | --- | --- | --- | --- | --- | --- |
> | $\mathcal{S}$ | 9.0 | 8.0 | 7.0 | 6.0 | 5.0 | 4.0 | 3.0 | 2.0 | 0.025 |
> | infidelity | 0.5 | 0.5 | 0.5 | 0.5 | 0.5 | 0.5 | 0.5 | 0.5 | 1.2e-4 |
>
> The evolution of $\mathcal{S}$ and the infidelity as a function of $T$ is plotted in Figure 12 in Appendix F. This example shows that a state can be **highly entangled** while AQER can still achieve a low entanglement measure and infidelity efficiently without being trapped in bad local minima.
>
> [1]. Martin Larocca, et al. Theory of overparametrization in quantum neural networks. Nat. Comput. Sci., 2023.

---

> ### Author Response · Authors · 2025-12-03
> **Response to the newly raised comments (3)**
>
> **Q3.** On the concern that convergence in the N = 50 experiment is “artificially simple”.
>
> **Response to Q3.** We respectfully disagree with the concern that successful convergence in the $N=50$ experiment is “artificially simple” for two reasons.
>
> *1. The $N = 50$ task is physically meaningful, not an artificial toy problem.* Our large-qubit experiments are performed on ground states of the transverse-field Ising model (TFIM) near the critical point, which is a standard benchmark in **quantum many-body** physics studies [1,2], instead of artificially simplified instances.
>
> *2. The benign optimization landscape is primarily due to the design of AQER.* The favorable behavior we observe arises from AQER’s three-step construction, in particular Step I (entanglement-guided circuit structure) and Step II (theory-driven parameter initialization). This is supported by the comparison with the hardware-efficient circuit (HEC) baseline in Table 1 of the main text. Under comparable or even fewer two-qubit gate counts, AQER consistently reaches **better optimized infidelity** than HEC across all datasets. These observations indicate that AQER induces a more favorable optimization landscape than the existing variational-circuit-based AQL for different tasks, rather than the tasks themselves being artificially simple.
>
> [1] Benedikt Fauseweh. Quantum many-body simulations on digital quantum computers: State-of-the-art and future challenges. Nat. Commun., 2024.
>
> [2] Riccardo Rende, Luciano Loris Viteritti, Federico Becca, Antonello Scardicchio, Alessandro Laio, Giuseppe Carleo. Foundation Neural-Networks Quantum States as a Unified Ansatz for Multiple Hamiltonians. Nat. Commun., 2025.
>
> **Q4.** On the concern that AQER only works for low-entanglement, structurally simple states, and no experiment for long-range, volume-law random-circuit states with large qubits.
>
> **Response to Q4.** We address this concern in two parts.
>
> *1. We do not expect and do not claim that large-qubit volume-law states from deep random quantum circuits fall within the regime of AQL and AQER.* Recent complexity-theoretic work [1] shows that learning the approximate output distributions of random quantum circuits with linear depth has **exponentially large** average-case complexity [1], while such linearly deep random circuits are known to generate volume-law entanglement [2]. We remark that an approximate encoding circuit for the target state naturally provides a procedure for generating its output distribution approximately. Thus, these hardness results suggest that, no AQL method can learn approximate encodings for large-qubit volume-law states generated from deep random circuits using sub-exponential time or resources. Therefore, large-qubit deep random circuit states are out of the scope of both AQL and AQER. We neither expect nor claim that AQER can scalably handle such states.
>
> *2. We have already conducted a long-range, volume-law random circuit benchmark at a moderate size.* As described in Section 4.1 in the main text, our experiments already include a benchmark on fully connected random circuits with $N=10$ qubits, 40 CZ gates with all-to-all full connectivity, and 120 single-qubit rotations, with all gates randomly shuffled. The average depth per qubit is $(120+40\times2)/N=20=2N$, which is linear in $N$. Thus, our 10-qubit random circuit benchmark is a small-$N$ instance drawn from the linear-depth volume-law random circuit family, rather than as a low-entanglement case. As shown in Table 1 in the main text, AQER consistently outperforms the reference methods on this benchmark. For example, with about 80 two-qubit gates, AQER already achieves infidelity below 0.1, whereas the other referenced AQL approaches have infidelity above 0.25. These results indicate that even in a regime at the boundary of AQL, AQER achieves better approximations than existing AQL methods under comparable gate budgets.
>
> [1] Alexander Nietner, Marios Ioannou, Ryan Sweke, Richard Kueng, Jens Eisert, Marcel Hinsche, Jonas Haferkamp. On the average-case complexity of learning output distributions of quantum circuits. Quantum, 2025.
>
> [2] Adam Nahum, Jonathan Ruhman, Sagar Vijay, Jeongwan Haah. Quantum Entanglement Growth Under Random Unitary Dynamics. Phys. Rev. X, 2017.

---

### Official Review · Reviewer_ctwW · 2025-11-01

**Soundness:** 3
**Presentation:** 3
**Contribution:** 2
**Rating:** 6
**Confidence:** 3

**Summary:**

This paper proposes a structured method for synthesizing circuits for quantum state preparation, with a primary focus on loading classical data onto a quantum computer as quantum states (e.g., encoding a classical vector into a quantum vector). The core insight behind the proposed method, AQER, is supported by a theoretical upper bound on the best-case infidelity (over initial product states), expressed in terms of the sum of single-qubit entanglement entropies of the target state evolved under a circuit.

Building on this insight, the method proceeds in three stages:
- Circuit search to reduce the entanglement of the target state.
- Approximation of the resulting low-entanglement state by a product state.
- Parameter refinement to optimize performance.

The authors perform simulation-based benchmarks on both classical and quantum data-loading tasks, demonstrating that their method consistently outperforms existing approaches.

**Strengths:**

- The approach is grounded in a clear and well-motivated theoretical insight, which lends credibility to the method and provides a room for future extensions. Given that quantum state preparation is, in general, an infeasible problem in terms of computational complexity, it is reasonable that the paper does not pursue purely theoretical guarantees.

- Numerical benchmarks across multiple datasets demonstrate consistent performance improvements over prior methods.

**Weaknesses:**

- Since the target hardware setting is the NISQ regime, the lack of experiments on real quantum devices, or even simulations under realistic noise models (e.g., depolarizing noise), makes it difficult to assess how the proposed method would perform in practice. This limitation is particularly relevant given that several prior works in this area include evaluations on real hardware.

**Questions:**

Can the theoretical characterization of infidelity be extended to general noise models, where the unitary evolution is replaced by a generic quantum channel (CPTP map)?

---

> ### Author Response · Authors · 2025-11-22
> **Author Response to Reviewer ctwW (1)**
>
> **Q1.** Can the theoretical characterization of infidelity be extended to general noise models, where the unitary evolution is replaced by a generic quantum channel (CPTP map)?
>
> **Response to Q1.** We thank the reviewer for raising this important question about extending our theoretical results to noisy quantum channels. **The answer is yes: Theorem 3.1 can be generalized to general CPTP maps.** In the revision, we make this explicit with two additional theorems in Appendix C.
>
> 1. **Theorem C.1:** We consider an arbitrary CPTP map $\mathcal{E}$ acting on $\rho_{\text{target}}$ and consider the entanglement measure $\mathcal{S}(\mathcal{E}(\rho_{\text{target}})) = S$. Then the infidelity between $\mathcal{E}(\rho_{\text{target}})$ and any product state is lower bounded by $f_1(S)$. Moreover, given access to $\mathcal{E}(\rho_{\text{target}})$, we can construct a product state, such that the infidelity is upper bounded by $f_2(S)$. Functions $f_1$ and $f_2$ are defined in Theorem 3.1 in the revision. **Both lower and upper bounds have the same form as Theorem 3.1.**
>
>     Technically, bounds in Theorems 3.1 and C.1 are the same because in the proof of Theorem 3.1 we already treat the state for the entanglement measure as a density matrix and perform all subsequent steps at the level of density matrices. Replacing the pure state with a generally mixed state does not change the structure of the argument.
>
> 2. **Theorem C.2:** We further analyze a **layered noisy circuit** with interleaved CPTP noise channels $\mathcal{M}$ and $\mathcal{N}$. We show that both bounds differ from that in Theorem 3.1 with an explicit additive noise term that scales as $(L+1)\big( \| \mathcal{M} - \text{id} \| _ \diamond + \| \mathcal{N} - \text{id} \| _ \diamond\big)$, where $L$ is the circuit depth, $\text{id}$ is the identity channel, and  $\| \cdot \|_\diamond$ is the diamond norm.
>
>     For common noise channels, the correction (diamond norm terms) has a clear formulation. For example, for a depolarizing channel $\mathcal{D}_p(\rho) = (1-p)\rho + p I/d$ with error rate $p$, the diamond norm term is $\mathcal{O}(p)$, so the additive term is of order $(L+1)p$. Similar linear relationships between the diamond norm and the error rate hold for other noise models such as dephasing and amplitude-damping channels. Thus, the entanglement-governed terms from Theorem 3.1 persist in the noisy setting, and the effect of noise enters as a correction that scales linearly with both the depth and the error rate for each layer.

---

> ### Author Response · Authors · 2025-11-22
> **Author Response to Reviewer ctwW (2)**
>
> **W1.** Since the target hardware setting is the NISQ regime, the lack of experiments on real quantum devices, or even simulations under realistic noise models (e.g., depolarizing noise), makes it difficult to assess how the proposed method would perform in practice. This limitation is particularly relevant given that several prior works in this area include evaluations on real hardware.
>
> **Response to W1.** We thank the reviewer for this comment and respond in two parts.
>
> *Why no experiments on real quantum computers.* AQER is proposed as a general framework that applies both to classical simulation and to quantum hardware (both fault-tolerant and near-term). In the current work, our focus is on developing and understanding the algorithmic framework. Due to limited access to real quantum devices, we follow the standard practice in quantum computing works on top machine learning conferences like ICLR/ICML/NeurIPS [1-11] and test AQER via classical simulations.
>
> *Noisy simulations show the robustness of AQER in NISQ regimes.* Following the reviewer’s suggestion, we add **noisy channel simulations** in the revision. In particular, we apply AQER to $N=10$ qubit TFIM tasks under global depolarizing channels. For the single-qubit gate layer, we adopt the error rate in different levels $p_1 \in \{10^{-4}, 3\times 10^{-3}, 10^{-3}\}$, and the corresponding error rate for the two-qubit gate layer is $p_2=10p_1$. These values are chosen to be representative of gate error rates reported on current NISQ devices [12, 13], where single-qubit (two-qubit) gate errors lie in the $10^{-4}\text{–}10^{-3}$ ($10^{-3}\text{–}10^{-2}$) regime. In the experiment, we observe that, as the two-qubit count $T$ increases, the final infidelity reduces up to an optimal $T$ point. Beyond this point, the entanglement measure starts to increase again as accumulated noise dominates.
>
> | T | 5 | 7 | 10 | 12 | 15 | 20 | 24 | 28 | 32 | 36 | 40 | 50 | 60 | 80 | 100 |
> | --- | --- | --- | --- | --- | --- | --- | --- | --- | --- | --- | --- | --- | --- | --- | --- |
> | infidelity ($p_1=10^{-3},p_2=10^{-2}$ ) | 0.39 | 0.33 | 0.22 | 0.19 | **0.17** | **0.17** | 0.18 | 0.20 | 0.22 | 0.23 | 0.24 | 0.29 | 0.32 | 0.40 | 0.46 |
> | infidelity ($p_1=3\times10^{-4},p_2=3\times10^{-3}$ ) | 0.38 | 0.30 | 0.17 | 0.12 | 0.092 | 0.074 | **0.071** | **0.071** | 0.073 | 0.072 | 0.076 | 0.091 | 0.11 | 0.13 | 0.16 |
> | infidelity ( $p_1=10^{-4},p_2=10^{-3}$ ) | 0.39 | 0.31 | 0.16 | 0.11 | 0.086 | 0.057 | 0.050 | 0.037 | 0.035 | **0.033** | 0.035 | 0.040 | 0.042 | 0.051 | 0.060 |
>
> These results demonstrate the robustness of AQER in NISQ regimes:
>
> - For **all tested noise levels**, there exists a noise-dependent optimal $T$ at which AQER achieves a **lower infidelity** compared to shallow circuits, indicating that the entanglement-guided optimization remains beneficial before noise accumulation becomes dominant.
> - As the physical error rates decrease from $p_1 = 10^{-3}$ to $p_1 = 10^{-4}$, the **best achievable infidelity improves gradually**.
>
> [1] Xin Wang, Hanxiao Tao, Rebing Wu. Predictive Performance of Deep Quantum Data Re-uploading Models. ICML 2025.
>
> [2] Jiaqi Leng, Bin Shi. Quantum Optimization via Gradient-Based Hamiltonian Descent. ICML 2025.
>
> [3] Lucas Fabian Naumann, Jannik Irmai, Bjoern Andres. A Sub-Problem Quantum Alternating Operator Ansatz for Correlation Clustering. ICML 2025.
>
> [4] Tak Hur, Daniel K. Park. Understanding Generalization in Quantum Machine Learning with Margins. ICML 2025.
>
> [5] Sabrina Herbst, Sandeep Cranganore, Vincenzo De Maio, Ivona Brandić. Exploring channel distinguishability in local neighborhoods of the model space in quantum neural networks. ICLR 2025.
>
> [6] Yusen Wu, Bujiao Wu, Yanqi Song, Xiao Yuan, Jingbo Wang. Learning the Complexity of Weakly Noisy Quantum States. ICLR 2025.
>
> [7] Chen-Yu Liu, Chao-Han Huck Yang, Hsi-Sheng Goan, Min-Hsiu Hsieh. A Quantum Circuit-Based Compression Perspective for Parameter-Efficient Learning. ICLR 2025.
>
> [8] Philipp Schleich, Marta Skreta, Lasse B. Kristensen, Rodrigo A. Vargas-Hernández, Alán Aspuru-Guzik. Quantum Deep Equilibrium Models. NeurIPS 2024.
>
> [9] Leo Zhou, Joao Basso, Song Mei. Statistical Estimation in the Spiked Tensor Model via the Quantum Approximate Optimization Algorithm. NeurIPS 2024.
>
> [10] Zhan Yu, Qiuhao Chen, Yuling Jiao, Yinan Li, Xiliang Lu, Xin Wang, Jerry Zhijian Yang. Non-asymptotic Approximation Error Bounds of Parameterized Quantum Circuits. NeurIPS 2024.
>
> [11] Connor Clayton, Jiaqi Leng, Gengzhi Yang, Yi-Ling Qiao, Ming C. Lin, Xiaodi Wu. Differentiable Quantum Computing for Large-scale Linear Control. NeurIPS 2024.
>
> [12] Dmitry A. Abanin, et al. Observation of constructive interference at the edge of quantum ergodicity. Nature 646, 825–830, 2025.
>
> [13] Anthony Ransford, et al. Helios: A 98-qubit trapped-ion quantum computer. arXiv:2511.05465.

---

> ### Author Response · Authors · 2025-11-26
> **Comment for Reviewer ctwW**
>
> Dear Reviewer ctwW:
>
> Thanks a lot for your helpful and thoughtful review of our paper! We have tried our best to address the mentioned concerns and have carefully revised the manuscript. Are there unclear explanations or remaining problems? We would be very grateful for further feedback and will do our best to address them.
>
> Best regards,
>
> The authors

---

> > ### Comment · Reviewer_ctwW · 2025-11-26
> >
> > Thank you for the detailed response and revision. I’m glad to see that the analysis has been extended to generic quantum channels beyond noise-free unitary evolution, and that the method has been evaluated under idealized noise models and demonstrated to be efficient. I have revised my score accordingly.

---

> ### Author Response · Authors · 2025-12-03
> **Summary of Reviewer ctwW’s Official Review and Our Response**
>
> First, we would like to thank the reviewer for carefully evaluating our work and for the constructive comments. In the original official review, the reviewer gave a positive score and a recommendation to accept. The reviewer highlighted two key strengths:
>
> (1) AQER is “grounded in a clear and well-motivated theoretical insight,” namely an information-theoretic bound that relates approximation error to a locally measurable entanglement proxy, which “lends credibility to the method and provides room for future extensions.”
>
> (2) numerical benchmarks across multiple datasets “demonstrate consistent performance improvements over prior methods.” The reviewer also noted that, given the inherent hardness of quantum state preparation, it is reasonable that the paper does not attempt fully general theoretical guarantees.
>
> The reviewer’s main concern was that,
>
> (W1) Since the target hardware setting is the NISQ regime, the lack of experiments on real quantum devices or simulations under realistic noise models made it difficult to assess practical performance.
>
> (Q1) The reviewer also asked whether the theoretical characterization of infidelity can be extended to general noise models.
>
> We addressed these points as follows. For Q1, we **extended Theorem 3.1 to general CPTP maps** by adding Theorems C.1 and C.2 in Appendix C, which generalized the original noiseless case to general CPTP noises.
>
> For W1, we explained that our focus in this work is on developing and analyzing the algorithmic framework, and we rely on classical simulations due to limited hardware access, which is consistent with standard practice in recent ICLR/ICML/NeurIPS quantum ML papers. In addition, following the reviewer’s suggestion, we **added noisy-channel simulations** on the $N=10$ TFIM task under global depolarizing noise with gate error rates chosen to match current NISQ devices. These experiments show that AQER remains effective up to a noise-dependent optimal two-qubit gate count, and that the best achievable infidelity improves as the physical error rates decrease.
>
> In the subsequent comment, the reviewer explicitly noted that the reviewer is “glad to see that the analysis has been extended to generic quantum channels beyond noise-free unitary evolution, and that the method has been evaluated under idealized noise models and demonstrated to be efficient,” and stated the **revision to a higher score (8)**.

---

### Meta-Review · Area_Chair_X4QE · 2026-01-01

**Summary:**

This paper proposed AQER, a scalable and efficient data loader for digital quantum computers. In the initial version of the paper, the following concerns were raised by reviewers:
- Computational complexity, more theory analysis
- Lack of experiments on real quantum devices, or classical simulations under realistic noise models
- More comprehensive metrics, such as gate depth, the number of trainable parameters, measurement overhead, and training cost.

During the rebuttal, the authors carefully addressed these points, by adding more theory analysis as well as detailed numerical experiments covering various metrics. Given the difficult circumstance this year at ICLR, reviewers Ki6d and eQiP weren't able to double check and confirm the rebuttal from authors, but according to the reading from the AC, the changes are sufficient and make the high-level picture clearer.

In all, given that the proposed AQER framework is a scalable and efficient data loader satisfying:
- Theoretical link between entanglement and quantum data loader performance;
- Theory-driven initialization;
- Global refinement of the true loss, with overall good performance in numerical experiments;

It is a timely result to be highlighted in ICLR 2026. The decision is to accept as a poster.

**Reviewer Concerns:**

The authors did a very careful rebuttal to address reviewer concerns, and the issues listed above are all addressed from my perspective. No outstanding issue.

**Reviewer Scores:**

During the limited discussion period this year, Reviewer ctwW had agreed to increase score from 6 to 8. Reviewer LZZQ gave an initial positive score of 6. Reviewers Ki6d and eQiP were involved into the discussion but interrupted by the unforeseen circumstance. If more discussions are allowed, there might be a chance that the scores were increased given that the authors detailed address of their concerns.

---

### Decision · Program_Chairs · 2026-01-26

Accept (Poster)